# A vision transformer for decoding surgeon activity from surgical videos

**Dani Kiyasseh** [1] ✉, **Runzhuo Ma**[2], **Taseen F. Haque** [2], **Brian J. Miles**[3], **Christian Wagner**[4], **Daniel A. Donoho**[5], **Animashree Anandkumar**[1] & **Andrew J. Hung** [2] ✉

The intraoperative activity of a surgeon has substantial impact on postoperative outcomes. However, for most surgical procedures, the details of intraoperative surgical actions, which can vary widely, are not well understood. Here we report a machine learning system leveraging a vision transformer and supervised contrastive learning for the decoding of elements of intraoperative surgical activity from videos commonly collected during robotic surgeries. The system accurately identified surgical steps, actions performed by the surgeon, the quality of these actions and the relative contribution of individual video frames to the decoding of the actions. Through extensive testing on data from three different hospitals located in two different continents, we show that the system generalizes across videos, surgeons, hospitals and surgical procedures, and that it can provide information on surgical gestures and skills from unannotated videos. Decoding intraoperative activity via accurate machine learning systems could be used to provide surgeons with feedback on their operating skills, and may allow for the identification of optimal surgical behaviour and for the study of relationships between intraoperative factors and postoperative outcomes.

The overarching goal of surgery is to improve postoperative patient outcomes[1,2]. It was recently demonstrated that such outcomes are strongly influenced by intraoperative surgical activity[3], that is, what actions are performed by a surgeon during a surgical procedure and how well those actions are executed. For the vast majority of surgical procedures, however, a detailed understanding of intraoperative surgical activity remains elusive. This scenario is all too common in other domains of medicine, where the drivers of certain patient outcomes either have yet to be discovered or manifest differently. The status quo within surgery is that intraoperative surgical activity is simply not measured. Such lack of measurement makes it challenging to capture the variability in the way surgical procedures are performed across time,

surgeons and hospitals, to test hypotheses associating intraoperative activity with patient outcomes, and to provide surgeons with feedback on their operating technique.

Intraoperative surgical activity can be decoded from videos commonly collected during robot-assisted surgical procedures. Such decoding provides insight into what procedural steps (such as tissue dissection and suturing) are performed over time, how those steps are executed (for example, through a set of discrete actions or gestures) by the operating surgeon, and the quality with which they are executed (that is, mastery of a skill; Fig. 1). Currently, if a video were to be decoded, it would be through a manual retrospective analysis by an expert surgeon. However, this human-driven approach is subjective, as

[1]Department of Computing and Mathematical Sciences, California Institute of Technology, Pasadena, CA, USA. [2]Center for Robotic Simulation and Education, Catherine & Joseph Aresty Department of Urology, University of Southern California, Los Angeles, CA, USA. [3]Department of Urology, Houston Methodist Hospital, Houston, TX, USA. [4]Department of Urology, Pediatric Urology and Uro-Oncology, Prostate Center Northwest, St. Antonius-Hospital, Gronau, Germany. [5]Division of Neurosurgery, Center for Neuroscience, Children's National Hospital, Washington, DC, USA. ✉e-mail: danikiy@hotmail.com; ajhung@gmail.com

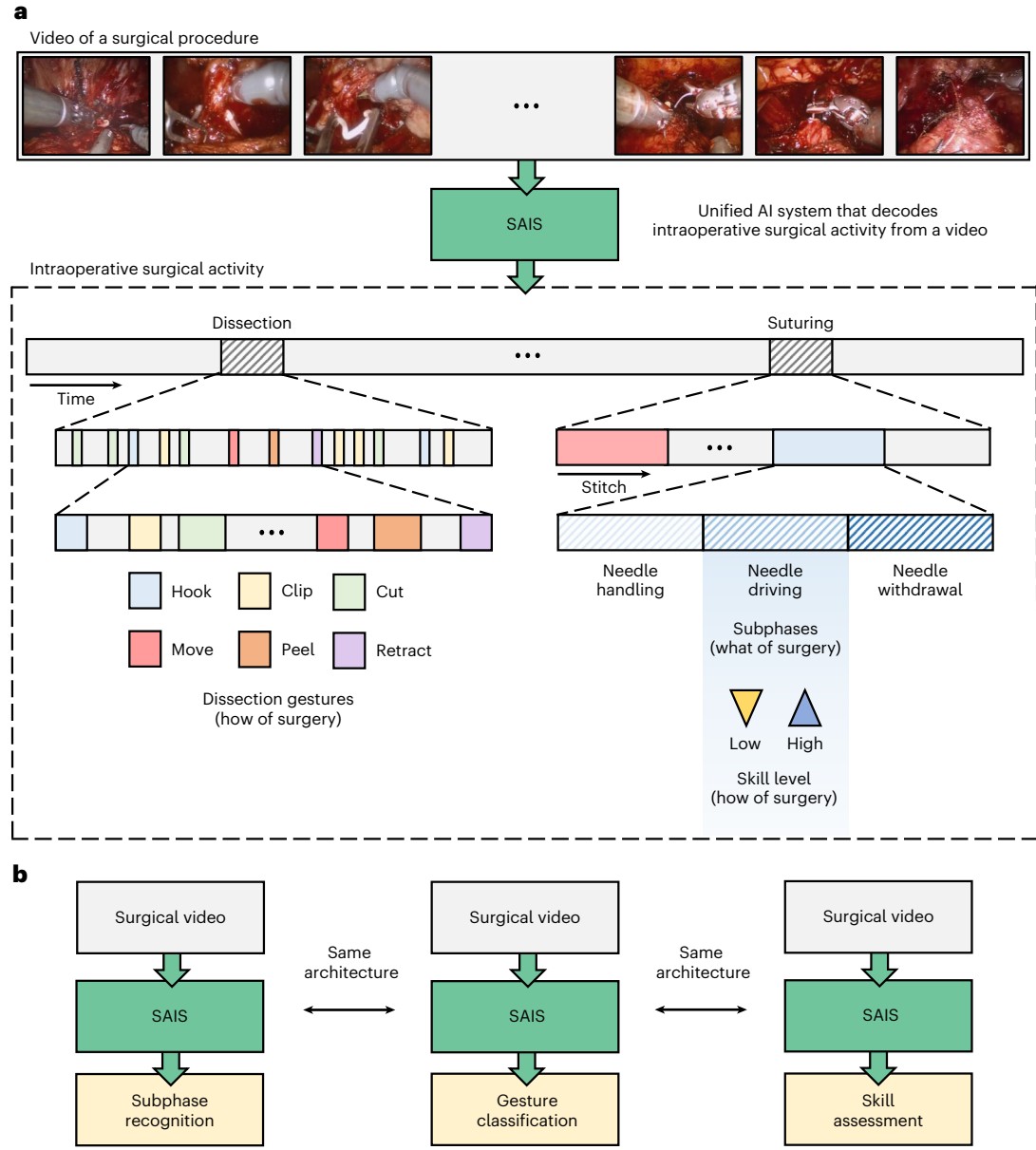

**Fig. 1 | An AI system that decodes intraoperative surgical activity from videos. a**, Surgical videos commonly collected during robotic surgeries are decoded via SAIS into multiple elements of intraoperative surgical activity: what is performed by a surgeon, such as the suturing subphases of needle handling, needle driving and needle withdrawal, and how that activity is executed by a surgeon, such as through discrete gestures and at different levels of skill. **b**, SAIS is a unified system since the same architecture can be used to independently decode different elements of surgical activity, from subphase recognition to gesture classification and skill assessment.

it depends on the interpretation of activity by the reviewing surgeon; unreliable, as it assumes that a surgeon is aware of all intraoperative activity; and unscalable, as it requires the presence of an expert surgeon and an extensive amount of time and effort. These assumptions are particularly unreasonable where expert surgeons are unavailable (as in low-resource settings) and already pressed for time. As such, there is a pressing need to decode intraoperative surgical activity in an objective, reliable and scalable manner.

Given these limitations, emerging technologies such as artificial intelligence (AI) have been used to identify surgical activity[4], gestures[5], surgeon skill levels[6,7] and instrument movements[8] exclusively from videos. However, these technologies are limited to decoding only a single element of intraoperative surgical activity at a time (such as only gestures), limiting their utility. These technologies are also seldom rigorously evaluated, where it remains an open question whether they

generalize to, or perform well in, new settings, such as with unseen videos from different surgeons, surgical procedures and hospitals. Such a rigorous evaluation is critical to ensuring the development of safe and trustworthy AI systems.

In this study, we propose a unified surgical AI system (SAIS) that decodes multiple elements of intraoperative surgical activity from videos collected during surgery. Through rigorous evaluation on data from three hospitals, we show that SAIS reliably decodes multiple elements of intraoperative activity, from the surgical steps performed to the gestures that are executed and the quality with which they are executed by a surgeon. This reliable decoding holds irrespective of whether videos are of different surgical procedures and from different surgeons across hospitals. We also show that SAIS decodes such elements more reliably than state-of-the-art AI systems, such as Inception3D (I3D; ref. 6), which have been developed to decode only a single element

**Table 1 | Total number of videos and video samples associated with each of the hospitals and tasks**

| Task | Activity | Details | Hospital | Videos | Video samples | Surgeons | Generalization to |
|---|---|---|---|---|---|---|---|
| Subphase recognition | Suturing | VUA | **USC** | 78 | 4,774 | 19 | Videos |
| | | | SAH | 60 | 2,115 | 8 | Hospitals |
| | | | HMH | 20 | 1,122 | 5 | Hospitals |
| | | | *USC* | 48 | Inference on entire videos | | |
| Gesture classification | Suturing | VUA | **USC** | 78 | 1,241 | 19 | Videos |
| | | Laboratory | JIGSAWS | 39 | 793 | 8 | Users |
| | | DVC | UCL | 36 | 1,378 | 8 | Videos |
| | Dissection | NS | **USC** | 86 | 1,542 | 15 | Videos |
| | | | SAH | 60 | 540 | 8 | Hospitals |
| | | | *USC* | 154 | Inference on entire unlabelled videos | | |
| | | RAPN | **USC** | 27 | 339 | 16 | Procedures |
| Skill assessment | Suturing | Needle handling | USC | 78 | 912 | 19 | Videos |
| | | | SAH | 60 | 240 | 18 | Hospitals |
| | | | HMH | 20 | 184 | 5 | Hospitals |
| | | Needle driving | **USC** | 78 | 530 | 19 | Videos |
| | | | SAH | 60 | 280 | 18 | Hospitals |
| | | | HMH | 20 | 220 | 5 | Hospitals |

Note that we train our model, SAIS, exclusively on data from hospitals whose names are shown in **bold** following a ten-fold Monte Carlo cross-validation setup. For an exact breakdown of the number of video samples in each fold and training, validation and test split, please refer to Supplementary Tables 1–5. The data from the remaining hospitals are exclusively used for inference. We perform inference on entire videos from hospitals whose names are shown in italics. Except for the task of subphase recognition, SAIS is always trained and evaluated on a class-balanced set of data whereby each category (low skill and high skill) contains the same number of samples. This prevents SAIS from being negatively affected by a sampling bias during training, and allows for a more intuitive appreciation of the evaluation results.

(such as surgeon skill). We also show that SAIS, through deployment on surgical videos without any human-driven annotations, provides information about intraoperative surgical activity, such as its quality over time, that otherwise would not have been available to a surgeon. Through a qualitative assessment, we demonstrate that SAIS provides accurate reasoning behind its decoding of intraoperative activity. With these capabilities, we illustrate how SAIS can be used to provide surgeons with actionable feedback on how to modulate their intraoperative surgical behaviour.

## Results

### SAIS reliably decodes surgical subphases

We decoded the 'what' of surgery by tasking SAIS to distinguish between three surgical subphases: needle handling, needle driving and needle withdrawal (Fig. 1). For all experiments, we trained SAIS on video samples exclusively from the University of Southern California (USC) (Table 1). A description of the surgical procedures and subphases is provided in Methods.

**Generalizing across videos.** We deployed SAIS on the test set of video samples from USC, and present the receiver operating characteristic (ROC) curves stratified according to the three subphases (Fig. 2a). We observed that SAIS reliably decodes surgical subphases with area under the receiver operating characteristic curve (AUC) of 0.925, 0.945 and 0.951, for needle driving, needle handling and needle withdrawal, respectively. We also found that SAIS can comfortably decode the high-level steps of surgery, such as suturing and dissection (Supplementary Note 3 and Supplementary Fig. 2).

**Generalizing across hospitals.** To determine whether SAIS can generalize to unseen surgeons at distinct hospitals, we deployed it on video samples from St. Antonius Hospital (SAH) (Fig. 2b) and Houston Methodist Hospital (HMH) (Fig. 2c). We found that SAIS continued to excel with AUC ≥0.857 for all subphases and across hospitals.

**Benchmarking against baseline models.** We deployed SAIS to decode subphases from entire videos of the vesico-urethral anastomosis (VUA) suturing step (20 min long) without any human supervision (inference section in Methods). We present the $F1_{10}$ score (Fig. 2e), a commonly reported metric[9], and contextualize its performance relative to that of a state-of-the-art I3D network[6]. We found that SAIS decodes surgical subphases more reliably than I3D, with these models achieving $F1_{10}$ of 50 and 40, respectively.

### The performance of SAIS stems from attention mechanism and multiple data modalities

To better appreciate the degree to which the components of SAIS contributed to its overall performance, we trained variants of SAIS, after having removed or modified these components (ablation section in Methods), and report their positive predictive value (PPV) when decoding the surgical subphases (Fig. 2d).

We found that the self-attention (SA) mechanism was the largest contributor to the performance of SAIS, where its absence resulted in ΔPPV of approximately −20. This finding implies that capturing the relationship between, and temporal ordering of, frames is critical for the decoding of intraoperative surgical activity. We also observed that the dual-modality input (red–green–blue, or RGB, frames and flow) has a greater contribution to performance than using either modality of data alone. By removing RGB frames ('without RGB') or optical flow ('without flow'), the model exhibited an average ΔPPV of approximately −3 relative to the baseline implementation. Such a finding suggests that these two modalities are complementary to one another. We therefore used the baseline model (SAIS) for all subsequent experiments.

### SAIS reliably decodes surgical gestures

In the previous section, we showed the ability of SAIS to decode surgical subphases (the 'what' of surgery) and to generalize to video samples from unseen surgeons at distinct hospitals, and also quantified the marginal benefit of its components via an ablation study. In this section, we

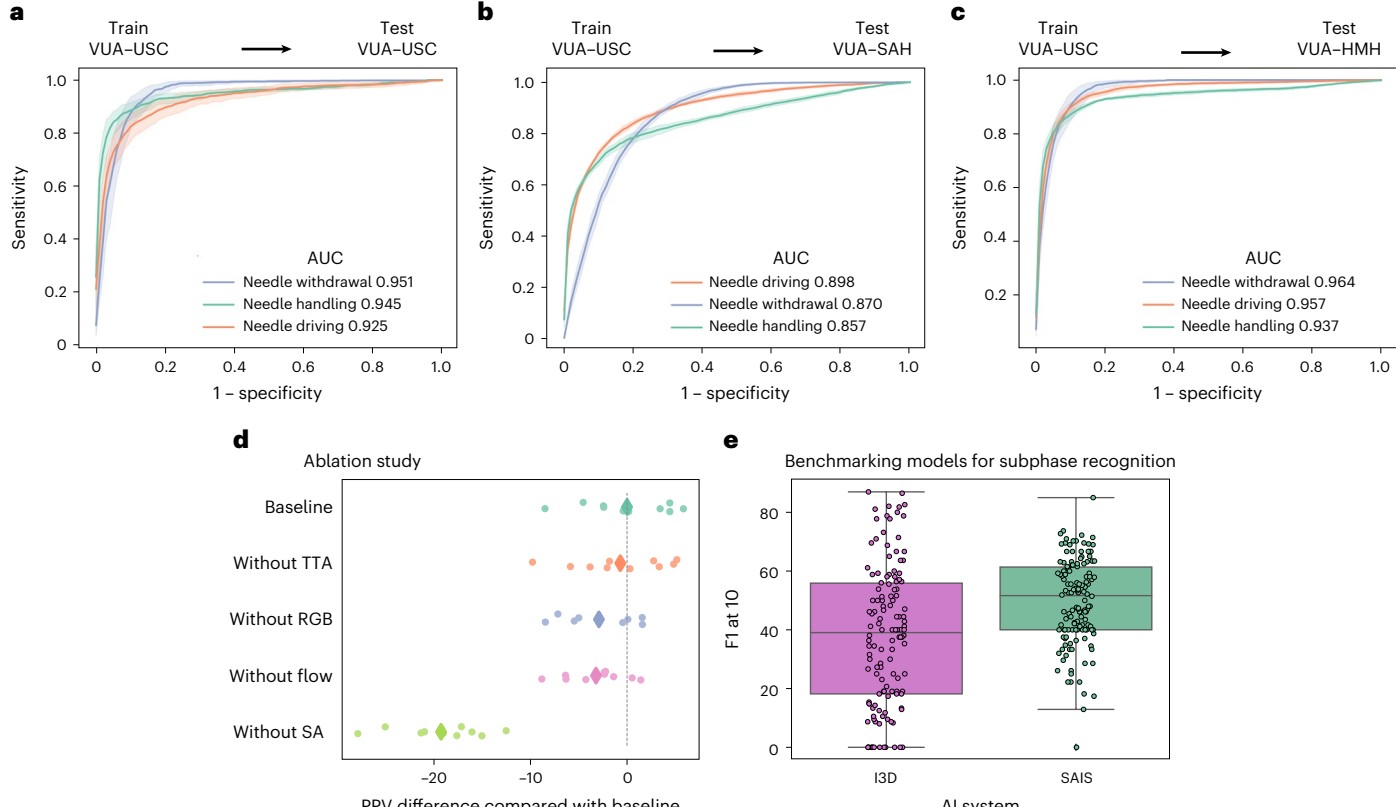

**Fig. 2 | Decoding surgical subphases from videos. a–c,** SAIS is trained on video samples exclusively from USC and evaluated on those from USC (**a**), SAH (**b**) and HMH (**c**). Results are shown as an average (±1 standard deviation) of ten Monte Carlo cross-validation steps. **d,** We trained variants of SAIS to quantify the marginal benefit of its components on its PPV. We removed test-time augmentation ('without TTA'), RGB frames ('without RGB'), flow maps ('without

flow') and the self-attention mechanism ('without SA'). We found that the attention mechanism and the multiple modality input (RGB and flow) are the greatest contributors to PPV. **e,** We benchmarked SAIS against an I3D model when decoding subphases from entire VUA videos without human supervision. Each box reflects the quartiles of the results, and the whiskers extend to 1.5× the interquartile range.

examine the ability of SAIS to decode surgical gestures (the 'how' of surgery) performed during both tissue suturing and dissection activities (the description of gestures and activities is provided in Methods). For the suturing activity (VUA), we trained SAIS to distinguish between four discrete suturing gestures: right forehand under (R1), right forehand over (R2), left forehand under (L1) and combined forehand over (C1). For the dissection activity, known as nerve sparing (NS), we trained SAIS to distinguish between six discrete dissection gestures: cold cut (c), hook (h), clip (k), camera move (m), peel (p) and retraction (r). We note that training was performed on video samples exclusively from USC.

**Generalizing across videos.** We deployed SAIS on the test set of video samples from USC, and present the ROC curves stratified according to the discrete suturing gestures (Fig. 3a) and dissection gestures (Fig. 3b). There are two main takeaways here. First, we observed that SAIS can generalize well to both suturing and dissection gestures in unseen videos. This is exhibited by the high AUC achieved by SAIS across the gestures. For example, in the suturing activity, AUC was 0.837 and 0.763 for the right forehand under (R1) and combined forehand over (C1) gestures, respectively. In the dissection activity, AUC was 0.974 and 0.909 for the clip (k) and camera move (m) gestures, respectively. These findings bode well for the potential deployment of SAIS on unseen videos for which ground-truth gesture annotations are unavailable, an avenue we explore in a subsequent section. Second, we found that the performance of SAIS differs across the gestures. For example, in the dissection activity, AUC was 0.701 and 0.974 for the retraction (r) and clip (k) gestures, respectively. We hypothesize that

the strong performance of SAIS for the latter stems from the clear visual presence of a clip in the surgical field of view. On the other hand, the ubiquity of retraction gestures in the surgical field of view could be a source of the relatively lower ability of SAIS in decoding retractions, as explained next. Retraction is often annotated as such when it is actively performed by a surgeon's dominant hand. However, as a core gesture that is used to, for example, improve a surgeon's visualization of the surgical field, a retraction often complements other gestures. As such, it can occur simultaneously with, and thus be confused for, other gestures by the model.

**Generalizing across hospitals.** To measure the degree to which SAIS can generalize to unseen surgeons at a distinct hospital, we deployed it on video samples from SAH (Fig. 3c and video sample count in Table 1). We found that SAIS continues to perform well in such a setting. For example, AUC was 0.899 and 0.831 for the camera move (m) and clip (k) gestures, respectively. Importantly, such a finding suggests that SAIS can be reliably deployed on data with several sources of variability (surgeon, hospital and so on). We expected, and indeed observed, a slight degradation in performance in this setting relative to when SAIS was deployed on video samples from USC. For example, AUC was 0.823 → 0.702 for the cold cut (c) gesture in the USC and SAH data, respectively. This was expected due to the potential shift in the distribution of data collected across the two hospitals, which has been documented to negatively affect network performance[10]. Potential sources of distribution shift include variability in how surgeons perform the same set of gestures (for instance, different techniques) and

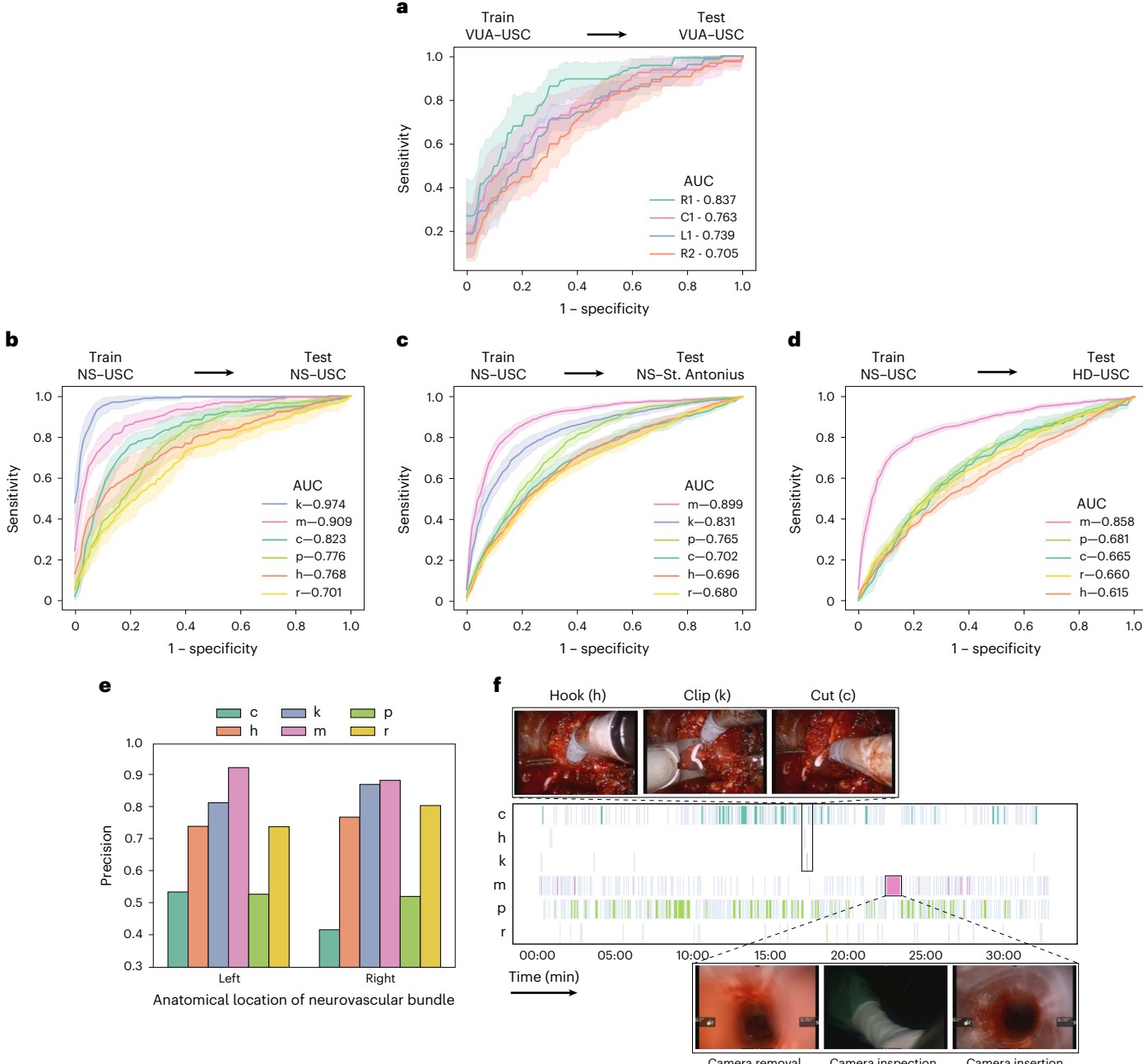

**Fig. 3 | Decoding surgical gestures from videos. a**, SAIS is trained and evaluated on the VUA data exclusively from USC. The suturing gestures are right forehand under (R1), right forehand over (R2), left forehand under (L1) and combined forehand over (C1). **b**–**d**, SAIS is trained on the NS data exclusively from USC and evaluated on the NS data from USC (**b**), NS data from SAH (**c**) and HD data from USC (**d**). The dissection gestures are cold cut (c), hook (h), clip (k), camera move (m), peel (p) and retraction (r). Note that clips (k) are not used during the HD step. Results are shown as an average (±1 standard deviation) of ten Monte Carlo cross-validation steps. **e**, Proportion of predicted gestures identified as correct (precision) stratified on the basis of the anatomical location of the neurovascular bundle in which the gesture is performed. **f**, Gesture profile where each row represents a distinct gesture and each vertical line represents the occurrence of that gesture at a particular time. SAIS identified a sequence of gestures (hook, clip and cold cut) that is expected in the NS step of RARP procedures, and discovered outlier behaviour of a longer-than-normal camera move gesture corresponding to the removal, inspection and re-insertion of the camera into the patient's body.

in the surgical field of view (for example, clear view with less blood). Furthermore, our hypothesis for why this degradation affects certain gestures (such as cold cuts) more than others (such as clips) is that the latter exhibits less variability than the former, and is thus easier to classify by the model.

**Generalizing across surgical procedures.** While videos of different surgical procedures (such as nephrectomy versus prostatectomy) may exhibit variability in, for example, anatomical landmarks (such as

kidney versus prostate), they are still likely to reflect the same tissue dissection gestures. We explored the degree to which such variability affects the ability of SAIS to decode dissection gestures. Specifically, we deployed SAIS on video samples of a different surgical step: renal hilar dissection (HD), from a different surgical procedure: robot-assisted partial nephrectomy (RAPN) (Fig. 3d and Table 1 for video sample count). We observed that SAIS manages to adequately generalize to an unseen surgical procedure, albeit exhibiting degraded performance, as expected (0.615 < AUC < 0.858 across the gestures). Interestingly,

the hook (h) gesture experienced the largest degradation in performance (AUC 0.768 → 0.615). We hypothesized that this was due to the difference in the tissue in which a hook is performed. Whereas in the NS dissection step, a hook is typically performed around the prostatic pedicles (a region of blood vessels), in the renal HD step, it is performed in the connective tissue around the renal artery and vein, delivering blood to and from the kidney, respectively.

**Validating on external video datasets.** To contextualize our work with previous methods, we also trained SAIS to distinguish between suturing gestures on two publicly available datasets: JHU-ISI gesture and skill assessment working set (JIGSAWS)[11] and dorsal vascular complex University College London (DVC UCL)[12] (Methods). While the former contains videos of participants in a laboratory setting, the latter contains videos of surgeons in a particular step (dorsal vascular complex) of the live robot-assisted radical prostatectomy (RARP) procedure. We compare the accuracy of SAIS with that of the best-performing methods on JIGSAWS (Supplementary Table 6) and DVC UCL (Supplementary Table 7).

We found that SAIS, despite not being purposefully designed for the JIGSAWS dataset, performs competitively with the baseline methods (Supplementary Table 6). For example, the best-performing video-based method achieved accuracy of 90.1, whereas SAIS achieved accuracy of 87.5. It is conceivable that incorporating additional modalities and dataset-specific modifications into SAIS could further improve its performance. As for the DVC UCL dataset, we followed a different evaluation protocol from the one originally reported[12] (see Implementation details of training SAIS on external video datasets in Methods) since only a subset of the dataset has been made public. To fairly compare the models in this setting, we quantify their improvement relative to a naive system that always predicts the majority gesture (Random) (Supplementary Table 7). We found that SAIS leads to a greater improvement in performance relative to the state-of-the-art method (MA-TCN) on the DVC UCL dataset. This is evident by the three-fold and four-fold increase in accuracy achieved by MA-TCN and SAIS, respectively, relative to a naive system.

### SAIS provides surgical gesture information otherwise unavailable to surgeons

One of the ultimate, yet ambitious, goals of SAIS is to decode surgeon activity from an entire surgical video without annotations and with minimal human oversight. Doing so would provide surgeons with information otherwise less readily available to them. In pursuit of this goal, and as an exemplar, we deployed SAIS to decode the dissection gestures from entire NS videos from USC (20–30 min in duration) to which it has never been exposed (Methods).

**Quantitative evaluation.** To evaluate this decoding, we randomly selected a prediction made by SAIS for each dissection gesture category in each video ($n = 800$ gesture predictions in total). This ensured we retrieved predictions from a more representative and diverse set of videos, thus improving the generalizability of our findings. We report the precision of these predictions after manually confirming whether or not the corresponding video samples reflected the correct gesture (Fig. 3e). We further stratified this precision on the basis of the anatomical location of the neurovascular bundle relative to the prostate gland. This allowed us to determine whether SAIS was (a) learning an

unreliable shortcut to decoding gestures by associating anatomical landmarks with certain gestures, which is undesirable, and (b) robust to changes in the camera angle and direction of motion of the gesture. For the latter, note that operating on the left neurovascular bundle often involves using the right-hand instrument and moving it towards the left of the field of view (Fig. 3f, top row of images). The opposite is true when operating on the right neurovascular bundle.

We found that SAIS is unlikely to be learning an anatomy-specific shortcut to decoding gestures and is robust to the direction of motion of the gesture. This is evident by its similar performance when deployed on video samples of gestures performed in the left and right neurovascular bundles. For example, hook (h) gesture predictions exhibited precision of ~0.75 in both anatomical locations. We also observed that SAIS was able to identify an additional gesture category beyond those it was originally trained on. Manually inspecting the video samples in the cold cut (c) gesture category with a seemingly low precision, we found that SAIS was identifying a distinct cutting gesture, also known as a hot cut, which, in contrast to a cold cut, involves applying heat/energy to cut tissue.

**Qualitative evaluation.** To qualitatively evaluate the performance of SAIS, we present its gesture predictions for a single 30-min NS video (Fig. 3f). Each row represents a distinct gesture, and each vertical line represents the occurrence of this gesture at a particular time. We observed that, although SAIS was not explicitly informed about the relationship between gestures, it nonetheless correctly identified a pattern of gestures over time which is typical of the NS step within RARP surgical procedures. This pattern constitutes a (a) hook, (b) clip and (c) cold cut and is performed to separate the neurovascular bundle from the prostate while minimizing the degree of bleeding that the patient incurs.

We also found that SAIS can discover outlier behaviour, despite not being explicitly trained to do so. Specifically, SAIS identified a contiguous 60-s interval during which a camera move (m) was performed, and which is 60× longer than the average duration (1 s) of a camera move. Suspecting outlier behaviour, we inspected this interval and discovered that it coincided with the removal of the camera from the patient's body, its inspection by the operating surgeon, and its re-insertion into the patient's body.

### SAIS reliably decodes surgical skills

At this point, we have demonstrated that SAIS, as a unified AI system, can independently achieve surgical subphase recognition (the what of surgery) and gesture classification (the how of surgery), and generalize to samples from unseen videos in the process. In this section, we examine the ability of SAIS to decode skill assessments from surgical videos. In doing so, we also address the how of surgery, however through the lens of surgeon skill. We evaluated the quality with which two suturing subphases were executed by surgeons: needle handling and needle driving (Fig. 1a, right column). We trained SAIS to decode the skill level of these activities using video samples exclusively from USC.

**Generalizing across videos.** We deployed SAIS on the test set of video samples from USC, and present the ROC curves associated with the skills of needle handling (Fig. 4a) and needle driving (Fig. 4b). We found that SAIS can reliably decode the skill level of surgical activity, achieving AUC of 0.849 and 0.821 for the needle handling and driving activity, respectively.

---

**Fig. 4 | Decoding surgical skills from videos and simultaneous provision of reasoning. a,b,** We train SAIS on video samples exclusively from USC to decode the skill-level of needle handling (**a**) and needle driving (**b**), and deploy it on video samples from USC, SAH and HMH. Results are an average (±1 standard deviation) of ten Monte Carlo cross-validation steps. **c,d,** We also present the attention placed on frames by SAIS for a video sample of low-skill needle handling (**c**) and needle driving (**d**). Images with an orange bounding box indicate that SAIS

places the highest attention on frames depicting visual states consistent with the respective skill assessment criteria. These criteria correspond to needle repositions and needle adjustments, respectively. **e,** Surgical skills profile depicting the skill assessment of needle handling and needle driving from a single surgical case at SAH. **f,g,** Ratio of low-skill needle handling (**f**) and needle driving (**g**) in each of the 30 surgical cases at SAH. The horizontal dashed lines represent the average ratio of low-skill activity at USC.

**Generalizing across hospitals.** We also deployed SAIS on video samples from unseen surgeons at two hospitals: SAH and HMH (Fig. 4a,b and Table 1 for video sample count). This is a challenging task that requires SAIS to adapt to the potentially different ways in which surgical activities are executed by surgeons with different preferences. We found that SAIS continued to reliably decode the skill level of needle

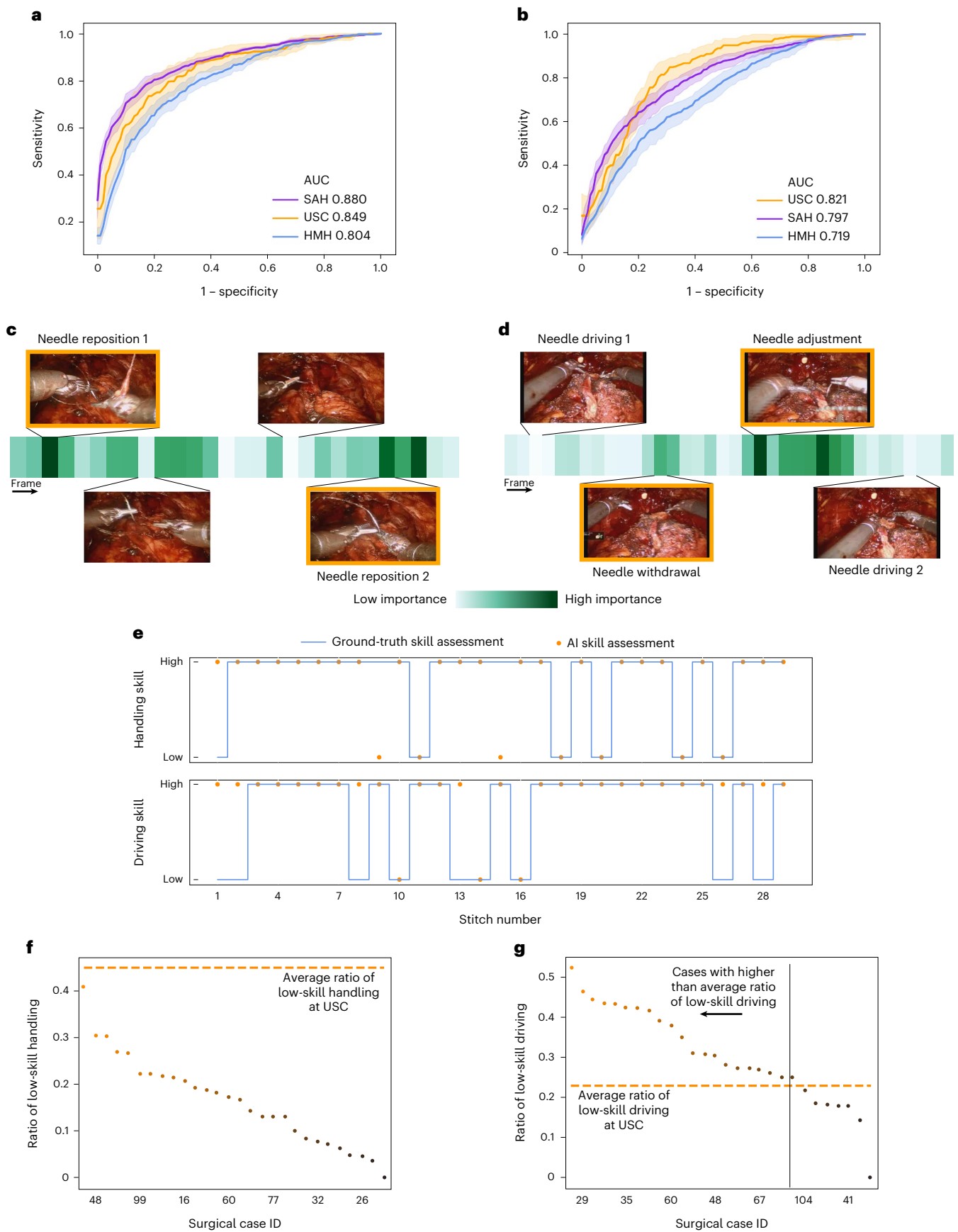

**Table 2 | SAIS outperforms a state-of-the-art model when decoding the skill level of surgical activity. SAIS is trained on video samples exclusively from USC. We report the average AUC (±1 standard deviation) on the test-set across all of the ten folds**

| Activity | Hospital | I3D (ref. 6) | SAIS |
|---|---|---|---|
| Needle handling | USC | 0.681 (0.07) | 0.849 (0.06) |
| | SAH | 0.730 (0.04) | 0.880 (0.02) |
| | HMH | 0.680 (0.04) | 0.804 (0.03) |
| Needle driving | USC | 0.630 (0.12) | 0.821 (0.05) |
| | SAH | 0.656 (0.08) | 0.797 (0.04) |
| | HMH | 0.571 (0.07) | 0.719 (0.06) |

handling (SAH: AUC 0.880, HMH: AUC 0.804) and needle driving (SAH: AUC 0.821, HMH: AUC 0.719). The ability of SAIS to detect consistent patterns across hospitals points to its potential utility for the objective assessment of surgical skills.

**Benchmarking against baseline models.** Variants of the 3D convolutional neural network (3D-CNN) have achieved state-of-the-art results in decoding surgical skills on the basis of videos of either a laboratory trial[6] or a live procedure[13]. As such, to contextualize the utility of SAIS, we fine-tuned a pre-trained I3D model (see Implementation details of I3D experiments in Methods) to decode the skill level of needle handling and needle driving (Table 2). We found that SAIS consistently outperforms this state-of-the-art model when decoding the skill level of surgical activities across hospitals. For example, when decoding the skill level of needle handling, SAIS and I3D achieved AUC of 0.849 and 0.681, respectively. When decoding the skill level of needle driving, they achieved AUC of 0.821 and 0.630, respectively. We also found that I3D was more sensitive to the video samples it was trained on and the initialization of its parameters. This is evident by the higher standard deviation of its performance relative to that of SAIS across the folds (0.12 versus 0.05 for needle driving at USC). Such sensitivity is undesirable as it points to the lack of robustness and unpredictable behaviour of the model.

## SAIS provides accurate reasoning behind decoding of surgical skills

The safe deployment of clinical AI systems often requires that they are interpetable[14]. We therefore wanted to explore whether or not SAIS was identifying relevant visual cues while decoding the skill level of surgeons. This would instill machine learning practitioners with confidence that SAIS is indeed latching onto appropriate features, and can thus be trusted in the event of future deployment within a clinical setting. We first retrieved a video sample depicting a low-skill activity (needle handling or needle driving) that was correctly classified by SAIS. By inspecting the attention placed on such frames by the attention mechanism (architecture in Fig. 5), we were able to quantify the importance of each frame. Ideally, high attention is placed on frames of relevance, where relevance is defined on the basis of the skill being assessed.

We present the attention (darker is more important) placed on frames of a video sample of needle handling (Fig. 4c) and needle driving (Fig. 4d) and that was correctly classified by SAIS as depicting low skill. We found that SAIS places the most attention on frames that are consistent with the skill assessment criteria. For example, with the low-skill needle handling activity based on the number of times a needle is re-grasped by a surgeon, we see that the most important frames highlight the time when both robotic arms simultaneously hold onto the needle, which is characteristic of a needle reposition manoeuvre (Fig. 4c). Multiple repetitions of this behaviour thus align well with the low-skill assessment of needle handling. Additionally, with needle driving assessed as low-skill based

on the smoothness of its trajectory, we see that the needle was initially driven through the tissue, adjusted, and then completely withdrawn (opposite to direction of motion) before being re-driven through the tissue seconds later (Fig. 4d). SAIS placed a high level of attention on the withdrawal of the needle and its adjustment and was thus in alignment with the low-skill assessment of needle driving. More broadly, these explainable findings suggest that SAIS is not only capable of providing surgeons with a reliable, objective, and scalable assessment of skill but can also pinpoint the important frames in the video sample. This capability addresses why a low-skill assessment was made and bodes well for when SAIS is deployed to provide surgeons with targeted feedback on how to improve their execution of surgical skills.

## SAIS provides surgical skill information otherwise unavailable to surgeons

We wanted to demonstrate that SAIS can also provide surgeons with information about surgical skills that otherwise would not have been available to them. To that end, we tasked SAIS with assessing the skill of all needle handling and needle driving video samples collected from SAH.

With needle handling (and needle driving) viewed as a subphase of a single stitch and knowing that a sequence of stitches over time makes up a suturing activity (such as VUA) in a surgical case, SAIS can generate a surgical skills profile for a single case (Fig. 4e) for needle handling and needle driving. We would like to emphasize that this profile, when generated for surgical cases that are not annotated with ground-truth skill assessments, provides surgeons with actionable information that otherwise would not have been available to them. For example, a training surgeon can now identify temporal regions of low-skill stitch activity, relate that to anatomical locations perhaps, and learn to focus on such regions in the future. By decoding profiles for different skills within the same surgical case, a surgeon can now identify whether subpar performance for one skill (such as needle handling) correlates with that for another skill (such as needle driving). This insight will help guide how a surgeon practises such skills.

SAIS can also provide actionable information beyond the individual surgical case level. To illustrate this, we present the proportion of needle handling (Fig. 4f) and needle driving (Fig. 4g) actions in a surgical case that were deemed low-skill, for all 30 surgical cases from SAH. We also present the average low-skill ratio observed in surgical videos from USC. With this information, the subset of cases with the lowest rate of low-skill actions can be identified and presented to training surgeons for educational purposes. By comparing case-level ratios to the average ratio at different hospitals (Fig. 4g), surgeons can identify cases that may benefit from further surgeon training.

## SAIS can provide surgeons with actionable feedback

We initially claimed that the decoding of intraoperative surgical activity can pave the way for multiple downstream applications, one of which is the provision of postoperative feedback to surgeons on their operating technique. Here we provide a template of how SAIS, based on the findings we have presented thus far, can deliver on this goal. In reliably decoding surgical subphases and surgical skills while simultaneously providing its reasoning for doing so, SAIS can provide feedback of the following form: 'when completing stitch number three of the suturing step, your needle handling (what−subphase) was executed poorly (how−skill). This is probably due to your activity in the first and final quarters of the needle handling subphase (why−attention)'. Such granular and temporally localized feedback now allows a surgeon to better focus on the element of intraoperative surgical activity that requires improvement, a capability that was not previously available.

## The skill assessments of SAIS are associated with patient outcomes

While useful for mastering a surgical technical skill itself, surgeon feedback becomes more clinically meaningful when grounded in

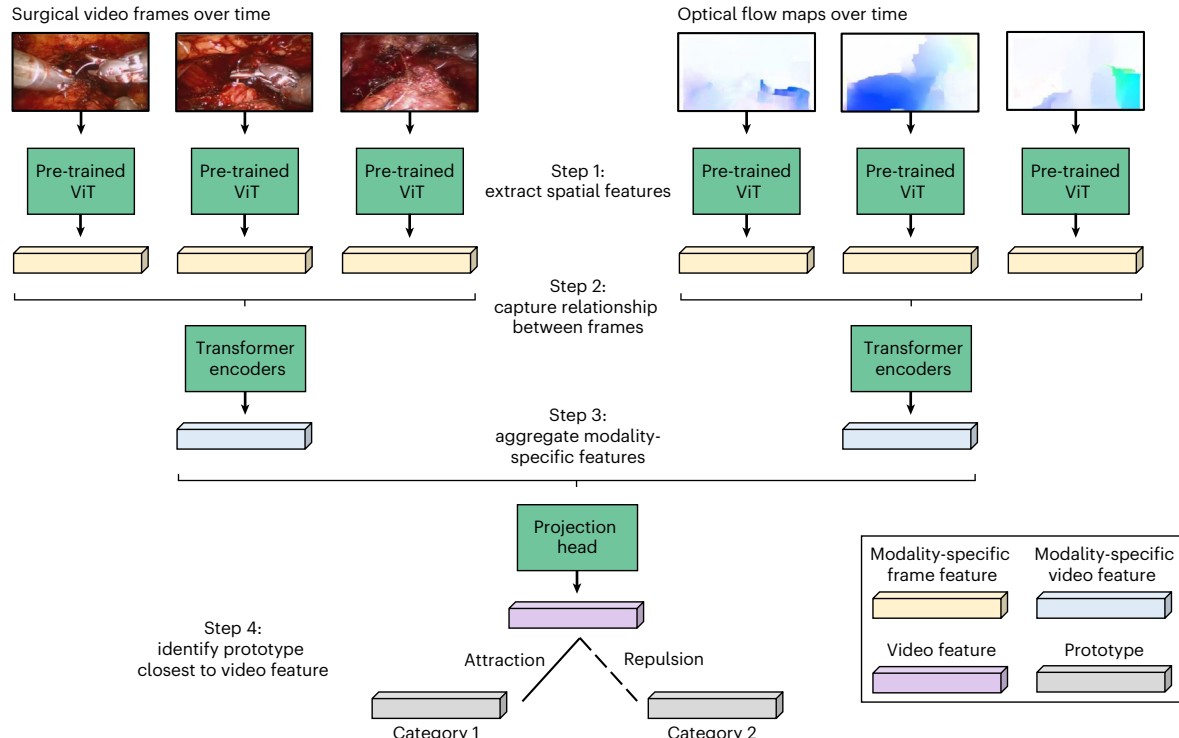

**Fig. 5 | A vision-and-attention-based AI system.** SAIS consists of two parallel streams that process distinct input data modalities: RGB surgical videos and optical flow. Irrespective of the data modality, features are extracted from each frame via a ViT pre-trained in a self-supervised manner on ImageNet. Features of video frames are then input into a stack of transformer encoders to obtain a modality-specific video feature. These modality-specific features are aggregated and passed into a projection head to obtain a single video feature, which is either attracted to, or repelled from, the relevant prototype. Although we illustrate two prototypes to reflect binary categories (high-skill activity versus low-skill activity), we would have C prototypes in a setting with C categories.

patient outcomes. For example, if low-skill assessments are associated with poor outcomes, then a surgeon can begin to modulate specific behaviour to improve such outcomes. To that end, we conducted a preliminary analysis regressing the surgeon skill assessments of SAIS at USC onto a patient's binary recovery of urinary continence (ability to voluntarily control urination) 3 months after surgery (Methods). When considering all video samples (multiple per surgical case), and controlling for surgeon caseload and patient age, we found that urinary continence recovery was 1.31× (odds ratio (OR), confidence interval (CI) 1.08–1.58, $P = 0.005$) more likely when needle driving was assessed as high skill than as low skill by SAIS. When aggregating the skill assessments of video samples within a surgical case, that relationship is further strengthened (OR 1.89, CI 0.95–3.76, $P = 0.071$). These preliminary findings are consistent with those based on manual skill assessments from recent studies[15,16].

## Discussion

Only in the past decade or so has it been empirically demonstrated that intraoperative surgical activity can have a direct influence on postoperative patient outcomes. However, discovering and acting upon this relationship to improve outcomes is challenging when the details of intraoperative surgical activity remain elusive. By combining emerging technologies such as AI with videos commonly collected during robotic surgeries, we can begin to decode multiple elements of intraoperative surgical activity.

We have shown that SAIS can decode surgical subphases, gestures and skills, on the basis of surgical video samples, in a reliable, objective and scalable manner. Although we have presented SAIS as decoding these specific elements in robotic surgeries, it can conceivably be applied to decode any other element of intraoperative activity from different surgical procedures. Decoding additional elements of surgery will simply require curating a dataset annotated with the surgical element of interest. To facilitate this, we release our code such that others can extract insight from their own surgical videos with SAIS. In fact, SAIS and the methods that we have presented in this study apply to any field in which information can be decoded on the basis of visual and motion cues.

Compared with previous studies, our study offers both translational and methodological contributions. From a translational standpoint, we demonstrated the ability of SAIS to generalize across videos, surgeons, surgical procedures and hospitals. Such a finding is likely to instil surgeons with greater confidence in the trustworthiness of SAIS, and therefore increases their likelihood of adopting it. This is in contrast to previous work that has evaluated AI systems on videos captured in either a controlled laboratory environment or a single hospital, thereby demonstrating limited generalization capabilities.

From a methodological standpoint, SAIS has much to offer compared with AI systems previously developed for decoding surgical activity. First, SAIS is unified in that it is capable of decoding multiple elements of intraoperative surgical activity without any changes to its underlying architecture. By acting as a dependable core architecture around which future developments are made, SAIS is likely to reduce the amount of resources and cognitive burden associated with developing AI systems to decode additional elements of surgical activity. This is in contrast to the status quo in which the burdensome process of developing specialized AI systems must be undertaken to decode just a single element. Second, SAIS provides explainable findings in that it can highlight the relative importance of individual video frames in contributing to the decoding. Such explainability, which we systematically investigate in a concurrent study[17] is critical to gaining the trust of surgeons and ensuring the safe deployment of AI systems for high-stakes decision making such as skill-based surgeon credentialing. This is in contrast to previous AI systems such as MA-TCN[12], which is only capable of highlighting the relative importance of data modalities

**a**

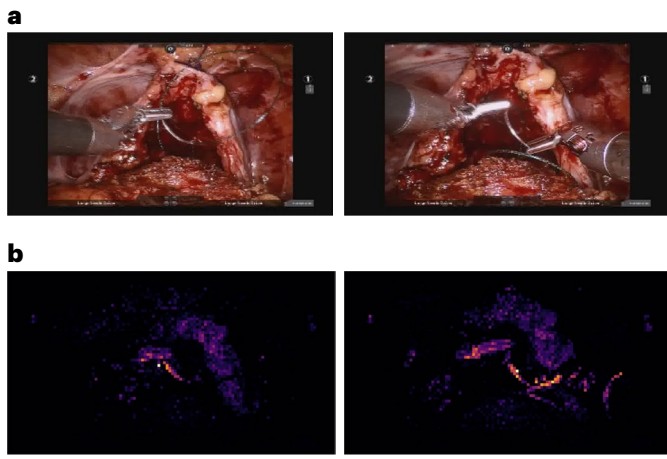

**b**

**Fig. 6 | ViT feature extractor places the highest importance on instrument tips, needles and anatomical edges.** We present two sample RGB video frames of the needle handling activity and the corresponding spatial attention placed by ViT on patches of these frames.

(for example, images versus kinematics), and therefore lacks the finer level of explainability of SAIS.

SAIS is also flexible in that it can accept video samples with an arbitrary number of video frames as input, primarily due to its transformer architecture. Such flexibility, which is absent from previous commonly used models such as 3D-CNNs, confers benefits to training, fine-tuning and performing inference. During training, SAIS can accept a mini-batch of videos each with a different number of frames. This can be achieved by padding videos in the mini-batch (with zeros) that have fewer frames, and appropriately masking the attention mechanism in the transformer encoder (see Implementation details and hyperparameters in Methods). This is in contrast to existing AI systems, which must often be presented with a mini-batch of equally sized videos. Similarly, during fine-tuning or inference, SAIS can be presented with an arbitrary number of video frames, thus expanding the spectrum of videos that it can be presented with. This is in contrast to existing setups that leverage a 3D-CNN that has been pre-trained on the Kinetics dataset[18], whereby video samples must contain either 16 frames or multiples thereof[6,13]. Abiding by this constraint can be suboptimal for achieving certain tasks, and departing from it implies the inability to leverage the pre-trained parameters that have proven critical to the success of previous methods. Furthermore, SAIS is architecturally different from previous models in that it learns prototypes via supervised contrastive learning to decode surgical activity, an approach that has yet to be explored with surgical videos. Such prototypes pave the way for multiple downstream applications from detecting out-of-distribution video samples, to identifying clusters of intraoperative activity, and retrieving samples from a large surgical database[19].

We also showed that SAIS can provide information that otherwise would not have been readily available to surgeons. This includes surgical gesture and skill profiles, which reflect how surgical activity is executed by a surgeon over time for a single surgical case and across different cases. Such capabilities pave the way for multiple downstream applications that otherwise would have been difficult to achieve. For example, from a scientific perspective, we can now capture the variability of surgical activity across time, surgeons and hospitals. From a clinical perspective, we can now test hypotheses associating intraoperative surgical activity with long-term patient outcomes. This brings the medical community one step closer to identifying, and eventually modulating, causal factors responsible for poor outcomes. Finally, from an educational perspective, we can now monitor and provide surgeons with feedback on their operating technique. Such feedback

can help surgeons master necessary skills and contribute to improved patient outcomes.

There are important challenges our work does not yet address. First, our framework, akin to others in the field, is limited to only decoding the elements of surgical activity that have been previously outlined in some taxonomy (such as gestures). In other words, it cannot decode what it does not know. Although many of these taxonomies have been rigorously developed by teams of surgeons and through clinical experience, they may fail to shed light on other intricate aspects of surgical activity. This, in turn, limits the degree to which automated systems can discover novel activity that falls beyond the realm of existing protocol. Such discovery can lend insight into, for example, optimal but as-of-yet undiscovered surgical behaviour. In a similar vein, SAIS is currently incapable of decoding new elements of surgical activity beyond those initially presented to it. Such continual learning capabilities[10] are critical to adapting to an evolving taxonomy of surgical activity over time.

The goal of surgery is to improve patient outcomes. However, it remains an open question whether or not the decoded elements of intraoperative surgical activity: subphases, gestures and skills, are the factors most predictive of postoperative patient outcomes. Although we have presented preliminary evidence in this direction for the case of surgical skills, large-scale studies are required to unearth these relationships. To further explore these relationships and more reliably inform future surgical practice, we encourage the public release of large-scale surgical video datasets from different hospitals and surgical specialties. Equipped with such videos and SAIS, researchers can begin to decode the various elements of surgery at scale.

Moving forward, we look to investigate whether SAIS has the intended effect on clinical stakeholders. For example, we aim to deploy SAIS in a controlled laboratory environment to assess the skill level of activity performed by medical students and provide them with feedback based on such assessments. This will lend practical insight into the utility of AI-based skill assessments and its perception by surgical trainees. We also intend to explore the interdependency of the elements of intraoperative surgical activity (subphase recognition, gesture classification and skill assessment). This can be achieved, for example, by training a multi-task variant of SAIS in which all elements are simultaneously decoded from a video. In such a setting, positive interference between the tasks could result in an even more reliable decoding. Alternatively, SAIS can be trained to first perform subphase recognition (a relatively easy task) before transferring its parameters to perform skill assessment (a relatively harder task). This is akin to curriculum learning[20], whereby an AI system is presented with increasingly difficult tasks during the learning process in order to improve its overall performance. In a concurrent study[21], we also investigate whether SAIS exhibits algorithmic bias against various surgeon subcohorts[22]. Such a bias analysis is particularly critical if SAIS is to be used for the provision of feedback to surgeons. For example, it may disadvantage certain surgeon subcohorts (such as novices with minimal experience) and thus affect their ability to develop professionally.

## Methods

### Ethics approval

All datasets (data from USC, SAH, and HMH) were collected under institutional review board approval in which informed consent was obtained (HS-17-00113). These datasets were de-identified before model development.

### Previous work

**Computational methods.** Previous work has used computational methods, such as AI, to decode surgery[23,24]. One line of research has focused on exploiting robot-derived sensor data, such as the displacement and velocity of the robotic arms (kinematics), to predict clinical outcomes[25–28]. For example, researchers have used automated performance metrics to predict a patient's postoperative length of stay

within a hospital[26]. Another line of research has instead focused on exclusively exploiting live surgical videos from endoscopic cameras to classify surgical activity[4,29], gestures[5,30–33] and skills[6,7,13,34,35], among other tasks[36,37]. For information on additional studies, we refer readers to a recent review[9]. Most recently, attention-based neural networks such as transformers[38] have been used to distinguish between distinct surgical steps within a procedure[39–42].

**Evaluation setups.** Previous work often splits their data in a way that has the potential for information 'leakage' across training and test sets. For example, it is believed that the commonly adopted leave-one-user-out evaluation setup on the JIGSAWS dataset[11] is rigorous. Although it lends insight into the generalizability of a model to a video from an unseen participant, this setup involves reporting a cross-validation score, which is often directly optimized by previous methods (for example, through hyperparameter tuning), therefore producing an overly optimistic estimate of performance. As another example, consider the data split used for the CholecT50 dataset[43]. Here there is minimal information about whether videos in the training and test sets belong to the same surgeon. Lastly, the most recent DVC UCL dataset[12] consists of 36 publicly available videos for training and 9 private videos for testing. After manual inspection, we found that these nine videos come from six surgeons whose data are also in the training set. This is a concrete example of surgeon data leakage, and as such, we caution the use of such datasets for benchmarking purposes. It is therefore critical to more rigorously evaluate the performance of SAIS, and in accordance with how it is likely to be deployed in a clinical setting.

## Description of surgical procedures and activities
We focused on surgical videos depicting two types of surgical activity commonly performed within almost any surgery: tissue dissection and suturing, which we next outline in detail.

**Tissue dissection.** Tissue dissection is a fundamental activity in almost any surgical procedure and involves separating pieces of tissue from one another. For example, the RARP surgical procedure, where a cancerous prostate gland is removed from a patient's body, entails several tissue dissection steps, one of which is referred to as nerve-sparing, or NS. NS involves preserving the neurovascular bundle, a mesh of vasculature and nerves to the left and right of the prostate, and is essential for a patient's postoperative recovery of erectile function for sexual intercourse. Moreover, an RAPN surgical procedure, where a part of a cancerous kidney is removed from a patient's body, entails a dissection step referred to as hilar dissection, or HD. HD involves removing the connective tissue around the renal artery and vein to control any potential bleeding from these blood vessels.

These dissection steps (NS and HD), although procedure specific (RARP and RAPN), are performed by a surgeon through a common vocabulary of discrete dissection gestures. In our previous work, we developed a taxonomy[44] enabling us to annotate any tissue dissection step with a sequence of discrete dissection gestures over time.

**Tissue suturing.** Suturing is also a fundamental component of surgery[45] and involves bringing tissue together. For example, the RARP procedure entails a suturing step referred to as vesico-urethral anastomosis, or VUA. VUA follows the removal of the cancerous prostate gland and involves connecting, via stitches, the bladder neck (a spherical structure) to the urethra (a cylindrical structure), and is essential for postoperative normal flow of urine. The VUA step typically consists of an average of 24 stitches where each stitch can be performed by a surgeon through a common vocabulary of suturing gestures. In our previous work, we developed a taxonomy[5] enabling us to annotate any suturing activity with a sequence of discrete suturing gestures. We note that suturing gestures are different to, and more subtle than, dissection gestures.

Each stitch can also be deconstructed into the three recurring subphases of (1) needle handling, where the needle is held in preparation for the stitch, (2) needle driving, where the needle is driven through tissue (such as the urethra), and (3) needle withdrawal, where the needle is withdrawn from tissue to complete a single stitch. The needle handling and needle driving subphases can also be evaluated on the basis of the skill level with which they are executed. In our previous work, we developed a taxonomy[46] enabling us to annotate any suturing subphase with a binary skill level (low skill versus high skill).

## Surgical video samples and annotations
We collected videos of entire robotic surgical procedures from three hospitals: USC, SAH and HMH. Each video of the RARP procedure, for example, was on the order of 2 h. A medical fellow (R.M.) manually identified the NS tissue dissection step and VUA tissue suturing step in each RARP video. We outline the total number of videos and video samples from each hospital in Table 1. We next outline how these steps were annotated with surgical subphases, gestures and skill levels.

It is important to note that human raters underwent a training phase whereby they were asked to annotate the same set of surgical videos, allowing for the calculation of the inter-rater reliability (between 0 and 1) of their annotations. Once this reliability exceeded 0.8, we deemed the training phase complete[47].

**Surgical gesture annotations.** Each video of the NS dissection step (on the order of 20 min) was retrospectively annotated by a team of trained human raters (R.M., T.H. and others) with tissue dissection gestures. This annotation followed the strict guidelines of our previously developed taxonomy of dissection gestures[44]. We focused on the six most commonly used dissection gestures: cold cut (c), hook (h), clip (k), camera move (m), peel (p) and retraction (r). Specifically, upon observing a gesture, a human rater recorded the start time and end time of its execution by the surgeon. Therefore, each NS step resulted in a sequence of $n \approx 400$ video samples of gestures (from six distinct categories) with each video sample on the order of 0–10 s in duration. Moreover, each video sample mapped to one and only one gesture. The same strategy was followed for annotating the VUA suturing step with suturing gestures. This annotation followed the strict guidelines of our previously developed taxonomy of suturing gestures[5]. We focused on the four most commonly used suturing gestures: right forehand under (R1), right forehand over (R2), left forehand under (L1) and combined forehand over (C1).

**Surgical subphase and skill annotations.** Each video of the VUA suturing step (on the order of 20 min) was retrospectively annotated by a team of trained human raters (D.K., T.H. and others) with surgical subphases and skills. This annotation followed the strict guidelines of our previously developed taxonomy referred to as the end-to-end assessment of suturing expertise or EASE[46]. Since the VUA step is a reconstructive one in which the bladder and urethra are joined together, it often requires a series of stitches (on the order of 24 stitches: 12 on the bladder side and another 12 on the urethral side).

With a single stitch consisting of the three subphases of needle handling, needle driving and needle withdrawal (always in that order), a human rater would first identify the start time and end time of each of these subphases. Therefore, each VUA step may have $n = 24$ video samples of the needle handling, needle driving and needle withdrawal subphases with each video sample on the order of 10–30 s. The distribution of the duration of such video samples is provided in Supplementary Note 2.

Human raters were also asked to annotate the quality of the needle handling or needle driving activity (0 for low skill and 1 for high skill). For needle handling, a high-skill assessment is based on the number of times the surgeon must reposition their grip on the needle in preparation for driving it through the tissue (the fewer the better).

For needle driving, a high-skill assessment is based on the smoothness and number of adjustments required to drive the needle through the tissue (the smoother and fewer number of adjustments the better). Since each video sample was assigned to multiple raters, it had multiple skill assessment labels. In the event of potential disagreements in annotations, we considered the lowest (worst) score. Our motivation for doing so was based on the assumption that if a human rater penalized the quality of the surgeon's activity, then it must have been due to one of the objective criteria outlined in the scoring system, and is thus suboptimal. We, in turn, wanted to capture and encode this suboptimal behaviour.

### Motivation behind evaluating SAIS with Monte Carlo cross-validation

In all experiments, we trained SAIS on a training set of video samples and evaluated it using ten-fold Monte Carlo cross- validation where each fold's test set consisted of subphases from videos unseen during training. Such an approach contributes to our goal of rigorous evaluation by allowing us to evaluate the ability of SAIS to generalize to unseen videos (hereon referred to as across videos). This setup is also more challenging and representative of real-world deployment than one in which an AI system generalizes to unseen samples within the same video. As such, we adopted this evaluation setup for all experiments outlined in this study, unless otherwise noted. A detailed breakdown of the number of video samples used for training, validation and testing can be found in Supplementary Note 1.

**Data splits.** For all the experiments conducted, unless otherwise noted, we split the data at the case video level into a training (90%) and test set (10%). We used 10% of the videos in the training set to form a validation set with which we performed hyperparameter tuning. By splitting at the video level, whereby data from the same video do not appear across the sets, we are rigorously evaluating whether the model generalizes across unseen videos. Note that, while it is possible for data from the same video to appear in both the training and test sets, we also experiment with even more rigorous setups: across hospitals—where videos are from entirely different hospitals and surgeons—and across surgical procedures—where videos are from entirely different surgical procedures (such as nephrectomy versus prostatectomy). While there are various ways to rigorously evaluate SAIS, we do believe that demonstrating its generalizability across surgeons, hospitals and surgical procedures, as we have done, is a step in the right direction. We report the performance of models as an average, with a standard deviation, across the folds.

### Leveraging both RGB frames and optical flow

To capture both visual and motion cues in surgical videos, SAIS operated on two distinct modalities: live surgical videos in the form of RGB frames and the corresponding optical flow of such frames. Surgical videos can be recorded at various sampling rates, which have the units of frames per second (fps).

Knowledge of the sampling rate alongside the natural rate with which activity occurs in a surgical setting is essential to multiple decisions. These can range from the number of frames to present to a deep learning network, and the appropriate rate with which to downsample videos, to the temporal step size used to derive optical flow maps, as outlined next. Including too many frames where there is very little change in the visual scene leads to a computational burden and may result in over-fitting due to the inclusion of highly similar frames (low visual diversity). On the other hand, including too few frames might result in missing visual information pertinent to the task at hand. Similarly, deriving reasonable optical flow maps, which is a function of a pair of images which are temporally spaced, is contingent upon the time that has lapsed between such images. Too short of a timespan could result in minimal motion in the visual scene, thus resulting in uninformative

optical flow maps. Analogously, too long of a timespan could mean missing out on informative intermediate motion in the visual scene. We refer to these decisions as hyperparameters (see Implementation details and hyperparameters section in Methods). Throughout this paper, we derived optical flow maps by deploying a RAFT model[48], which we found to provide reasonable maps.

### SAIS is a model for decoding activity from surgical videos

Our AI system—SAIS—is vision based and unified (Fig. 5). It is vision based as it operates exclusively on surgical videos routinely collected as part of robotic surgical procedures. It is unified as the same architecture, without any modifications, can be used to decode multiple elements of intraoperative surgical activity (Fig. 1b). We outline the benefits of such a system in Discussion.

### Single forward pass through SAIS

**Extracting spatial features.** We extract a sequence of $D$-dimensional representations, $\{v_t \in \mathbb{R}^D\}_{t=1}^T$, from $T$ temporally ordered frames via a (frozen) vision transformer (ViT) pre-trained on the ImageNet dataset in a self-supervised manner[49]. In short, this pre-training setup, entitled DINO, involved optimizing a contrastive objective function whereby representations of the same image, augmented in different ways (such as random cropping), are encouraged to be similar to one another. For more details, please refer to the original paper[50].

ViTs convert each input frame into a set of square image patches of dimension $H \times H$ and introduce a self-attention mechanism that attempts to capture the relationship between image patches (that is, spatial information). We found that this spatial attention picks up on instrument tips, needles, and anatomical edges (Fig. 6). We chose this feature extractor on the basis of (a) recent evidence favouring self-supervised pre-trained models relative to their supervised counterparts and (b) the desire to reduce the computational burden associated with training a feature extractor in an end-to-end manner.

**Extracting temporal features.** We append a learnable $D$-dimensional classification embedding, $e_{cls} \in \mathbb{R}^D$, to the beginning of the sequence of frame representations, $\{v_t\}_{t=1}^T$. To capture the temporal ordering of the frames of the images, we add $D$-dimensional temporal positional embeddings, $\{e_t \in \mathbb{R}^D\}_{t=1}^T$, to the sequence of frame representations before inputting the sequence into four Transformer encoder layers. Such an encoder has a self-attention mechanism whereby each frame attends to every other frame in the sequence. As such, both short- and long-range dependencies between frames are captured. We summarize the modality-specific video through a modality-specific video representation, $h_{cls} \in \mathbb{R}^D$, of the classification embedding, $e_{cls}$, at the final layer of the Transformer encoder, as is typically done. This process is repeated for the optical flow modality stream.

**Aggregating modality-specific features.** The two modality-specific video representations, $h_{RGB}$ and $h_{Flow}$, are aggregated as follows:

$$h_{agg} = h_{RGB} + h_{Flow} \tag{1}$$

The aggregated representation, $h_{agg}$, is passed through two projection heads, in the form of linear layers with a non-linear activation function (ReLU), to obtain an $E$-dimensional video representation, $h_{Video} \in \mathbb{R}^E$.

**Training protocol for SAIS.** To achieve the task of interest, the video-specific representation, $h_{Video}$, undergoes a series of attractions and repulsions with learnable embeddings, which we refer to as prototypes. Each prototype, $p$, reflects a single category of interest and is of the same dimensionality as $h_{Video}$. The representation, $h_{Video} \in \mathbb{R}^E$, of a video from a particular category, $c$, is attracted to the single prototype, $p_c \in \mathbb{R}^E$, associated with the same category, and repelled from all other prototypes, $\{p_j\}_{j=1}^C$, $j \neq c$, where $C$ is the total number of categories.

We achieve this by leveraging contrastive learning and minimizing the InfoNCE loss, $\mathcal{L}_{\mathrm{NCE}}$:

$$\mathcal{L}_{\mathrm{NCE}} = -\sum_{i=1}^{B} \log \frac{e^{s(h_{\mathrm{Video}},\, p_c)}}{\sum_{j} e^{s(h_{\mathrm{Video}},\, p_j)}} \tag{2}$$

$$s(h_{\mathrm{Video}}, p_j) = \frac{h_{\mathrm{Video}} \cdot p_j}{|h_{\mathrm{Video}}||p_j|}$$

During training, we share the parameters of the Transformer encoder across modalities to avoid over-fitting. As such, we learn, in an end-to-end manner, the parameters of the Transformer encoder, the classification token embedding, the temporal positional embeddings, the parameters of the projection head and the category-specific prototypes.

**Evaluation protocol for SAIS.** To classify a video sample into one of the categories, we calculate the similarity (that is, cosine similarity) between the video representation, $h_{\mathrm{Video}}$, and each of the prototypes, $\{p_j\}_{j=1}^{C}$. We apply the softmax function to these similarity values in order to obtain a probability mass function over the categories. By identifying the category with the highest probability mass (argmax), we can make a classification.

The video representation, $h_{\mathrm{Video}}$, can be dependent on the choice of frames (both RGB and optical flow) which are initially input into the model. Therefore, to account for this dependence and avoid missing potentially informative frames during inference, we deploy what is known as test-time augmentation (TTA). This involves augmenting the same input multiple times during inference, which, in turn, outputs multiple probability mass functions. We can then average these probability mass functions, analogous to an ensemble model, to make a single classification. In our context, we used three test-time inputs; the original set of frames at a fixed sampling rate, and those perturbed by offsetting the start frame by $K$ frames at the same sampling rate. Doing so ensures that there is minimal frame overlap across the augmented inputs, thus capturing different information, while continuing to span the most relevant aspects of the video.

**Implementation details and hyperparameters**
During training and inference, we use the start time and end time of each video sample to guide the selection of video frames from that sample. For gesture classification, we select ten equally spaced frames from the video sample. For example, for a video sample with a frame rate of 30 Hz and that is 3 s long, then from the original $30 \times 3 = 90$ frames, we would only retrieve frames $\in [0, 9, 18, \ldots]$. In contrast, for subphase recognition and skill assessment, we select every other tenth frame. For example, for the same video sample above, we would only retrieve frames $\in [0, 10, 20, \ldots]$. We found that these strategies resulted in a good trade-off between computational complexity and capturing sufficiently informative signals in the video to complete the task. Similarly, optical flow maps were based on pairs of images that were 0.5 s apart. Shorter timespans resulted in frames that exhibited minimal motion and thus uninformative flow maps. During training, to ensure that the RGB and optical flow maps were associated with the same timespan, we retrieved maps that overlapped in time with the RGB frames. During inference, and for TTA, we offset both RGB and optical flow frames by $K = 3$ and $K = 6$ frames.

We conduct our experiments in PyTorch[51] using a V100 GPU on a DGX machine. Each RGB frame and optical flow map was resized to $224 \times 224$ (from $960 \times 540$ at USC and SAH and $1,920 \times 1,080$ at SAH) before being input into the ViT feature extractor. The ViT feature extractor pre-processed each frame into a set of square patches of dimension $H = 16$ and generated a frame representation of dimension $D = 384$. All video representations and prototypes are of dimension $E = 256$. In practice, we froze the parameters of the ViT, extracted all such representations offline (that is, before training), and stored them as h5py files. We followed the same strategy for extracting representations

of optical flow maps. This substantially reduced the typical bottleneck associated with loading videos and streamlined our training and inference process. This also facilitates inference performed on future videos. Once a new video is recorded, its features can immediately be extracted in an offline manner, and stored for future use.

Unless otherwise stated, we trained SAIS using a mini-batch size of eight video samples and a learning rate of $1e^{-1}$, and optimize its parameters via stochastic gradient descent. Mini-batch samples are often required to have the same dimensionality ($B \times T \times D$) where $B$ is the batch size, $T$ is the number of frames and $D$ is the dimension of the stored representation. Therefore, when we encountered video samples in the same mini-batch with a different number of temporal frames (such as $T = 10$ versus $T = 11$), we first appended placeholder representations (tensors filled with zeros) to the end of the shorter video samples. This ensured all video samples in the mini-batch had the same dimension. To avoid incorporating these padded representations into downstream processing, we used a masking matrix (matrix with binary entries) indicating which representations the attention mechanism should attend to. Importantly, padded representations are not attended to during a forward pass through SAIS.

**Description of ablation study**
We trained several variants of SAIS to pinpoint the contribution of each its components on overall performance. Specifically, the model variants are trained using SAIS (baseline), evaluated without test-time augmentation ('without TTA'), and exposed to only optical flow ('without RGB') or RGB frames ('without flow') as inputs. We also removed the self-attention mechanism which captured the relationship between, and temporal ordering of, frames ('without SA'). In this setting, we simply averaged the frame features. Although we present the PPV in Results, we arrived at similar findings when using other evaluation metrics.

**Implementation details of inference on entire videos**
After we trained and evaluated a model on video samples (on the order of 10–30 s), we deployed it on entire videos (on the order of 10–30 min) to decode an element of surgical activity without human supervision. We refer to this process as inference. As we outline next, a suitable implementation of inference is often dependent on the element of surgical activity being decoded.

**Suturing subphase recognition.** Video samples used for training and evaluating SAIS to decode the three suturing subphases of needle handling, needle driving and needle withdrawal spanned, on average, 10–30 s (Supplementary Note 2). This guided our design choices for inference.

**Curating video samples for inference.** During inference, we adopted two complementary approaches, as outlined next. Approach 1: we presented SAIS with 10-s video samples from an entire VUA video with 5-s overlaps between subsequent video samples, with the latter ensuring we capture boundary activity. As such, each 10-s video sample was associated with a single probabilistic output, $\{s_{\mathrm{NH}}, s_{\mathrm{ND}}, s_{\mathrm{NW}}\}$, reflecting the probability, $s$, of needle handling (NH), needle driving (ND) and needle withdrawal (NW). Approach 2: we presented SAIS with 5-s non-overlapping video samples from the same video. The motivation for choosing a shorter video sample is to capture a brief subphase that otherwise would have bled into another subphase when using a longer video sample. As such, each 5-s video sample was associated with a single probabilistic output. Note that we followed the same approach for selecting frames from each video sample as we did during the original training and evaluation setup (see Implementation details and hyperparameters).

As an example of these approaches, the first video sample presented to SAIS in approach 1 spans 0–10 s whereas the first two video samples presented to SAIS in approach 2 span 0–5 s and 5–10 s,

respectively. When considering both approaches, the timespan 0–10 s is thus associated with three unique probabilistic outputs (as is every other 10-s timespan).

**Using ensemble models.** Recall that we trained SAIS using ten-fold Monte Carlo cross-validation, resulting in ten unique models. To increase our confidence in the inference process, we performed inference following the two aforementioned approaches with each of the ten models. As such, each 10-s timespan was associated with 3 probabilistic outputs ($P$) × 10 folds ($F$) × 3 TTAs = 90 probabilistic outputs in total. As is done with ensemble models, we then averaged these probabilistic outputs (a.k.a. bagging) to obtain a single probabilistic output, $\{\bar{s}_{NH}, \bar{s}_{ND}, \bar{s}_{NW}\}$, where the $j$th probability value for $j \in [1, C]$ ($C$ categories) is obtained as follows:

$$\bar{s}_i = \sum_{P=1}^{3} \sum_{F=1}^{10} \sum_{TTA=1}^{3} s_j^{TTA,F,P} \forall j \in [1, C] \tag{3}$$

**Abstaining from prediction.** In addition to ensemble models often outperforming their single model counterparts, they can also provide an estimate of the uncertainty about a classification. Such uncertainty quantification can be useful for identifying out-of-distribution video samples[52] such as those the model has never seen before or for highlighting video samples where the classification is ambiguous and thus potentially inaccurate. To quantify uncertainty, we took inspiration from recent work[53] and calculated the entropy, $S$, of the resultant probabilistic output post bagging. With high entropy implying high uncertainty, we can choose to abstain from considering classifications whose entropy exceeds some threshold, $S_{thresh}$:

$$S = -\sum_{j=1}^{c} \bar{s}_j \log \bar{s}_j > S_{thresh} \tag{4}$$

**Aggregating predictions over time.** Once we have filtered out the predictions which are uncertain (that is, exhibit high entropy), we were left with individual predictions for each subphase spanning at most 10 s (because of how we earlier identified video samples). However, we know from observation that certain subphases can be longer than 10 s (Supplementary Note 2). To account for this, we aggregated subphase predictions that were close to one another over time. Specifically, we aggregated multiple predictions of the same subphase into a single prediction if they were less than 3 s apart, in effect chaining the predictions. Although this value is likely to be dependent on other choices in the inference process, we found it to produce reasonable results.

**NS dissection gesture classification.** Video samples used for training and evaluating SAIS to decode the six dissection gestures spanned, on average, 1–5 s. This also guided our design choices for inference.

**Identifying video samples for inference.** During inference, we found it sufficient to adopt only one of the two approaches for inference described earlier (inference for subphase recognition). Specifically, we presented SAIS with 1-s non-overlapping video samples of an entire NS video. As such, each 1-s video sample was associated with a single probabilistic output, $\{s_j\}_{j=1}^{6}$ reflecting the probability, $s$, of each of the six gestures.

**Using ensemble models.** As with inference for suturing subphase recognition, we deployed the ten SAIS models (from the ten Monte Carlo folds) and three TTAs on the same video samples. As such, each 1-s video sample was associated with 10 × 3 = 30 probabilistic outputs. These are then averaged to obtain a single probabilistic output, $\{\bar{s}_j\}_{j=1}^{6}$.

**Abstaining from prediction.** We also leveraged the entropy of gesture classifications as a way to quantify uncertainty and thus abstain

from making highly uncertain gesture classifications. We found that $S_{thresh} = 1.74$ led to reasonable results.

**Aggregating predictions over time.** To account for the observation that gestures can span multiple seconds, we aggregated individual 1-s predictions that were close to one another over time. Specifically, we aggregated multiple predictions of the same gesture into a single prediction if they were less than 2 s apart. For example, if a retraction gesture (r) is predicted at intervals 10–11 s, 11–12 s and 15–16 s, we treated this as two distinct retraction gestures. The first one spans 2 s (10–12 s) while the second one spans 1 s (15–16 s). This avoids us tagging spurious and incomplete gestures (for example, the beginning or end of a gesture) as an entirely distinct gesture over time. Our 2-s interval introduced some tolerance for a potential misclassification between gestures of the same type and allowed for the temporal continuity of the gestures.

**Implementation details of training SAIS on external video datasets**
We trained SAIS on two publicly available datasets: JIGSAWS[11] and DVC UCL[12]. In short, these datasets contain video samples of individuals performing suturing gestures either in a controlled laboratory setting or during the dorsal vascular complex step of the RARP surgical procedure. For further details on these datasets, we refer readers to the original respective publications.

**JIGSAWS dataset.** We followed the commonly adopted leave-one-user-out cross-validation setup[11]. This involves training on video samples from all but one user and evaluating on those from the remaining user. These details can be found in a recent review[9].

**DVC UCL dataset.** This dataset, recently released as part of the Endoscopic Vision Challenge 2022 at MICCAI, consists of 45 videos from a total of eight surgeons performing suturing gestures during the dorsal vascular complex step of the RARP surgical procedure[12]. The publicly available dataset, at the time of writing, is composed of 36 such videos (Table 1). Similar to the private datasets we used, each video (on the order of 2–3 min) is annotated with a sequence of eight unique suturing gestures alongside their start time and end time. Note that these annotations do not follow the taxonomy we have developed and are therefore distinct from those we outlined in the Surgical video samples and annotations section. The sole previous method to evaluate on this dataset does so on a private test set. As this test set is not publicly available, we adopted a leave-one-video-out setup and reported the ten-fold cross-validation performance of SAIS (Supplementary Table 3 for the number of video samples in each fold). Such a setup provides insight into how well SAIS can generalize to unseen videos. Furthermore, in light of the few samples from one of the gesture categories (G5), we distinguished between only seven of the gestures. To facilitate the reproducibility of our findings, we will release the exact data splits used for training and testing.

**Implementation details of I3D experiments**
We trained the I3D model to decode the binary skill level of needle handling and needle driving on the basis of video samples of the VUA step. For a fair comparison, we presented the I3D model with the same exact data otherwise presented to SAIS (our model). In training the I3D model, we followed the core strategy proposed in ref. 6. For example, we loaded the parameters pre-trained on the Kinetics dataset and froze all but the last three layers (referred to as Mixed_{5b}, Mixed_{5c} and logits).

However, having observed that the I3D model was quite sensitive to the choice of hyperparameters, we found it necessary to conduct an extensive number of experiments to identify the optimal setup and hyperparameters for decoding surgical skill, the details of which are outlined next. First, we kept the logits layer as is, resulting in a 400-dimensional representation, and followed it with a non-linear classification head to output the probability of, for example, a high-skill

activity. We also leveraged both data modalities (RGB and flow) which we found to improve upon the original implementation that had used only a single modality. Specifically, we added the two 400-dimensional representations (one for each modality) to one another and passed the resultant representation through the aforementioned classification head. With the pre-trained I3D expecting an input with 16 frames or multiples thereof, we provided it with a video sample composed of 16 equally spaced frames between the start time and end time of that sample. While we also experimented with a different number of frames, we found that to yield suboptimal results. To train I3D, we used a batch-size of 16 video samples and a learning rate of $1e^{-3}$.

## Association between the skill assessments of SAIS and patient outcomes

To determine whether the skill assessments of SAIS are associated with patient outcomes, we conducted an experiment with two variants. We first deployed SAIS on the test set of video samples in each fold of the Monte Carlo cross-validation setup. This resulted in an output, $Z_1 \in [0, 1]$, for each video sample reflecting the probability of a high-skill assessment. In the first variant of this experiment, we assigned each video sample, linked to a surgical case, a urinary continence recovery (3 months after surgery) outcome, $Y$. To account for the fact that a single outcome, $Y$, is linked to an entire surgical case, in the second variant of this experiment, we averaged the outputs, $Z$, for all video samples within the same surgical case. This, naturally, reduced the total number of samples available.

In both experiments, we controlled for the total number of robotic surgeries performed by the surgeon (caseload, $Z_2$) and the age of the patient being operated on ($Z_3$), and regressed the probabilistic outputs of SAIS to the urinary continence recovery outcome using a logistic regression model (SPSS), as shown below ($\sigma$ is the sigmoid function). After training this model, we extracted the coefficient, $b_1$, and report the odds ratio (OR) and the 95% confidence interval (CI).

$$Y = \sigma\left(b_0 + b_1 Z_1 + b_2 Z_2 + b_3 Z_3\right)$$
$$\text{OR} = \frac{\text{odds}_{\text{high-skill}}}{\text{odds}_{\text{low-skill}}} = e^{b_1} \tag{5}$$

## Reporting summary
Further information on research design is available in the Nature Portfolio Reporting Summary linked to this article.

## Data availability
Data supporting the results in this study involve surgeon and patient data. As such, while the data from SAH and HMH are not publicly available, de-identified data from USC can be made available upon reasonable request from the authors.

## Code availability
Code is made available at https://github.com/danikiyasseh/SAIS.

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

## Acknowledgements

We are grateful to T. Chu for the annotation of videos with gestures. We also thank J. Laca and J. Nguyen for early feedback on the presentation of the manuscript. A.J.H. discloses support for the research described in this study from the National Cancer Institute under award no. R01CA251579-01A1 and a multi-year Intuitive Surgical Clinical Research Grant.

## Author contributions

D.K. and A.J.H. contributed to the conception of the study. D.K. contributed to the study design, developed the deep learning models and wrote the manuscript. R.M. and T.H. provided annotations for the video samples. D.A.D. provided extensive feedback on the manuscript. B.J.M. provided data for the study. C.W. collected data from SAH and provided feedback on the manuscript. A.J.H. and A.A. provided supervision and contributed to edits of the manuscript.

## Competing interests

D.K. is a paid employee of Vicarious Surgical and a consultant of Flatiron Health. C.W. is a paid consultant of Intuitive Surgical. A.A. is an employee of Nvidia. A.J.H. is a consultant of Intuitive Surgical. The other authors declare no competing interests.

## Additional information

**Correspondence and requests for materials** should be addressed to Dani Kiyasseh or Andrew J. Hung.

# nature research

# Reporting Summary

Nature Research wishes to improve the reproducibility of the work that we publish. This form provides structure for consistency and transparency in reporting. For further information on Nature Research policies, see our Editorial Policies and the Editorial Policy Checklist.

## Statistics

For all statistical analyses, confirm that the following items are present in the figure legend, table legend, main text, or Methods section.

| n/a | Confirmed | |
|---|---|---|
| ☐ | ☒ | The exact sample size (*n*) for each experimental group/condition, given as a discrete number and unit of measurement |
| ☐ | ☒ | A statement on whether measurements were taken from distinct samples or whether the same sample was measured repeatedly |
| ☒ | ☐ | The statistical test(s) used AND whether they are one- or two-sided<br>*Only common tests should be described solely by name; describe more complex techniques in the Methods section.* |
| ☐ | ☒ | A description of all covariates tested |
| ☒ | ☐ | A description of any assumptions or corrections, such as tests of normality and adjustment for multiple comparisons |
| ☐ | ☒ | A full description of the statistical parameters including central tendency (e.g. means) or other basic estimates (e.g. regression coefficient) AND variation (e.g. standard deviation) or associated estimates of uncertainty (e.g. confidence intervals) |
| ☒ | ☐ | For null hypothesis testing, the test statistic (e.g. *F*, *t*, *r*) with confidence intervals, effect sizes, degrees of freedom and *P* value noted<br>*Give P values as exact values whenever suitable.* |
| ☒ | ☐ | For Bayesian analysis, information on the choice of priors and Markov chain Monte Carlo settings |
| ☒ | ☐ | For hierarchical and complex designs, identification of the appropriate level for tests and full reporting of outcomes |
| ☒ | ☐ | Estimates of effect sizes (e.g. Cohen's *d*, Pearson's *r*), indicating how they were calculated |

*Our web collection on statistics for biologists contains articles on many of the points above.*

## Software and code

Policy information about availability of computer code

| Data collection | No software was used to collect data. |
|---|---|
| Data analysis | Custom code, Python 3, PyTorch 1.8.0. Code is available at https://github.com/danikiyasseh/SAIS. |

For manuscripts utilizing custom algorithms or software that are central to the research but not yet described in published literature, software must be made available to editors and reviewers. We strongly encourage code deposition in a community repository (e.g. GitHub). See the Nature Research guidelines for submitting code & software for further information.

## Data

Policy information about availability of data

All manuscripts must include a data availability statement. This statement should provide the following information, where applicable:
- Accession codes, unique identifiers, or web links for publicly available datasets
- A list of figures that have associated raw data
- A description of any restrictions on data availability

The data supporting the results in this study involve surgeon and patient data. The data from St. Antonius Hospital and Houston Methodist Hospital are not publicly available, yet de-identified data from the University of Southern California can be made available by the corresponding authors on reasonable request.

April 2020

# Field-specific reporting

Please select the one below that is the best fit for your research. If you are not sure, read the appropriate sections before making your selection.

☒ Life sciences  ☐ Behavioural & social sciences  ☐ Ecological, evolutionary & environmental sciences

For a reference copy of the document with all sections, see nature.com/documents/nr-reporting-summary-flat.pdf

# Life sciences study design

All studies must disclose on these points even when the disclosure is negative.

| | |
|---|---|
| Sample size | All experiments were conducted using 10-fold Monte Carlo cross-validation. Details of the number of samples and surgical videos used during training, validation and testing are provided in the Supplementary Information. |
| Data exclusions | We did not exclude any patients for training, validation or testing. |
| Replication | The protocol is described in the paper, and in-depth implementation details are provided in the Supplementary Information. |
| Randomization | The experiments were conducted across 10 folds. Each fold consisted of a different subset of data on which to train and evaluate the model. The initialization of the parameters of the network was also different across folds. Each fold was set based on a seed (from 0 to 10). |
| Blinding | Blinding was not applicable. When comparing different methods, we maintained the same experimental settings to allow for a fair comparison. |

# Reporting for specific materials, systems and methods

We require information from authors about some types of materials, experimental systems and methods used in many studies. Here, indicate whether each material, system or method listed is relevant to your study. If you are not sure if a list item applies to your research, read the appropriate section before selecting a response.

## Materials & experimental systems

| n/a | Involved in the study |
|---|---|
| ☒ | Antibodies |
| ☒ | Eukaryotic cell lines |
| ☒ | Palaeontology and archaeology |
| ☒ | Animals and other organisms |
| ☐ ☒ | Human research participants |
| ☒ | Clinical data |
| ☒ | Dual use research of concern |

## Methods

| n/a | Involved in the study |
|---|---|
| ☒ | ChIP-seq |
| ☒ | Flow cytometry |
| ☒ | MRI-based neuroimaging |

## Human research participants

Policy information about studies involving human research participants

| | |
|---|---|
| Population characteristics | Table 1 provides a summary of the number of video samples used for the different experimental tasks (sub-phase recognition, gesture classification, and skill assessment). We also include the total number of surgeons from each hospital who were associated with these samples. |
| Recruitment | Surgeons from different hospitals were recruited as part of Award No. R01CA251579-01A1 by the National Cancer Institute. Videos from hospitals exclusively reflected two different types of surgical procedures: robot-assisted radical prostatectomy and robot-assisted partial nephrectomy. Details of these procedures can be found in Methods. |
| Ethics oversight | All datasets (data from the University of Southern California, St. Antonius Hospital, and Houston Methodist Hospital) were collected under Institutional Review Board (IRB) approval, in which informed consent was obtained (HS-17-00113). These datasets were de-identifed prior to model development. |

Note that full information on the approval of the study protocol must also be provided in the manuscript.

