## [Peer Review File · Nature Biomedical Engineering]

A vision transformer for decoding surgeon activity from surgical videos

Corresponding author: Dani Kiyasseh

Editorial note

This document includes relevant written communications between the manuscript's corresponding author and the editor and reviewers of the manuscript during peer review. It includes decision letters relaying any editorial points and peer-review reports, and the authors' replies to these (under 'Rebuttal' headings). The editorial decisions are signed by the manuscript's handling editor, yet the editorial team and ultimately the journal's Chief Editor share responsibility for all decisions.

Any relevant documents attached to the decision letters are referred to as **Appendix #**, and can be found appended to this document. Any information deemed confidential has been redacted or removed. Earlier versions of the manuscript are not published, yet the originally submitted version may be available as a preprint. Because of editorial edits and changes during peer review, the published title of the paper and the title mentioned in below correspondence may differ.

Correspondence

Wed 08 Jun 2022

Decision on Presubmission Enquiry nBME-22-1333-PE

Dear Dr Kiyasseh,

Thank you for submitting to *Nature Biomedical Engineering* your Presubmission Enquiry, "Decoding surgeon activity from surgical videos with a unified artificial intelligence system".

As you may know, we screen Presubmission Enquiries against our editorial criteria. These editorial judgements are based on considerations of fit to the journal's scope and, when enough information is provided, of the degree of advance, broad implications, and breadth and depth of the work.

In this case, the topic of the Presubmission Enquiry is within the remit of the journal, and we would like to invite you to submit a full manuscript so that we can carry out a full editorial assessment.

I should also ask you to please fill in our reporting summary and policy checklist. (Please note that these forms are dynamic PDF files that can only be properly visualized and filled in by using Acrobat Reader.)

Both documents are aimed at ensuring good reporting standards and at easing the interpretation of results, and will be available to any reviewers. Should the manuscript be eventually published, the reporting summary will be attached to the published PDF of the paper and will also be available as supplementary information. More information is available on the editorial policies page.

Moreover, we highly recommend that you use our manuscript template. This will help you ensure that the manuscript complies with our data-presentation recommendations, that it includes all the necessary sections, and that it is structured to facilitate the assessment of peer reviewers and editors. In particular, please make sure that the manuscript provides thorough information on statistics and methods, and that the images comply with our guidelines for image integrity.When you are ready to submit the manuscript, please upload the manuscript files as well as the reporting summary and policy checklist.

Best wishes,

Pep

Pep Pàmies
Chief Editor, Nature Biomedical Engineering

Tue 26 Jul 2022

Decision on Article NBME-22-1333A

Dear Dr Kiyasseh,

Thank you again for submitting to *Nature Biomedical Engineering* your manuscript, "Decoding surgeon activity from surgical videos with a unified artificial intelligence system". The manuscript has been seen by four experts, whose reports you will find at the end of this message.

You will see that the reviewers appreciate the work, and in particular its translational value. However, they express concerns about the degree of technical innovation and raise queries about the performance claims. They also provide many useful suggestions for improvement, also with regards to the reporting of the methodology. We hope that with significant further work you can address the criticisms and convince the reviewers of the merits of the study. In particular, we would expect that a revised version of the manuscript provides:

- * Benchmarking of the model against state-of-the-art surgeon-skill recognition models.
- * Validation of the model on public datasets.
- * Thorough and clear methodological details.

When you are ready to resubmit your manuscript, please upload the revised files, a point-by-point rebuttal to the comments from all reviewers, the reporting summary, and a cover letter that explains the main improvements included in the revision and responds to any points highlighted in this decision.

Please follow the following recommendations:

- * Clearly highlight any amendments to the text and figures to help the reviewers and editors find and understand the changes (yet keep in mind that excessive marking can hinder readability).
- * If you and your co-authors disagree with a criticism, provide the arguments to the reviewer (optionally, indicate the relevant points in the cover letter).
- * If a criticism or suggestion is not addressed, please indicate so in the rebuttal to the reviewer comments and explain the reason(s).
- * Consider including responses to any criticisms raised by more than one reviewer at the beginning of the rebuttal, in a section addressed to all reviewers.
- * The rebuttal should include the reviewer comments in point-by-point format (please note that we provide all reviewers will the reports as they appear at the end of this message).
- * Provide the rebuttal to the reviewer comments and the cover letter as separate files.

We hope that you will be able to resubmit the manuscript within 15 weeks from the receipt of this message. If this is the case, you will be protected against potential scooping. Otherwise, we will be happy to consider a revised manuscript as long as the significance of the work is not compromised by work published elsewhere or accepted for publication at *Nature Biomedical Engineering*.

We hope that you will find the referee reports helpful when revising the work. Please do not hesitate to contact me should you have any questions.

Best wishes,

Pep

Pep Pàmies
Chief Editor, Nature Biomedical Engineering

Reviewer #1 (Report for the authors (Required)):

//A brief summary of the results.//

The authors propose a unified surgical AI system that can decode surgical activities. This work developed a system named SAIS which can identify the procedural steps of surgery, the actions performed by a surgeon, and the quality of such actions by utilizing two data modalities (RGB frames and optical flow maps). The proposed system can be used to decode surgical steps (phase), gestures and the surgeon's skill. Extensive experiments are conducted to show models' performance and robustness in generalizing across videos, hospitals and surgical produces in performing those tasks. The author's writing in the introduction section is commendable as it adequately stresses the need and advantage of such application. This work has merits in terms of application. However, it needs major revision in the results and methodology section to help readers understand it easily. The details on the dataset, dataset preparation and proposed model are scattered between the results and methodology section making it hard to fully comprehend. Furthermore, the proposed system lacks significant novelty and benchmarks against the existing system. The workload of this work in preparing the dataset is heavy. However, the work lacks comparative experiments and technical novelty.

//Your reasoned opinion on the degree of advance (fundamental, mechanistic, methodological, technological, therapeutic, translational and/or clinical) of the work with respect to the state of the art. If the results or conclusions are not original, please provide relevant references.//

This work showcases application novelty, where similar architectures (the last layer of the architecture changes depending on the number of classes for each task) for phase/gesture/skill recognition. While the results provided are original and interesting, the degree of advancement in terms of the model is unclear due to the following:

- Based on my understanding, the proposed SAIS model is a single-task (pseudo-tri-task) model, which can be trained to recognise phase/gesture/surgeon skills. However, to perform all three tasks at once requires three SAIS models in parallel. In such a case, how different is the SAIS model compared to existing gesture recognition models, which can also be technically used for the remaining tasks since they also warrant only changing the last layer?
- What is the advantage of using multiple SAIS models in parallel vs using the existing state-of-the-art models for phase/gesture/skill recognition in parallel? Additional comparison of computation requirements will also provide additional insights.
- Are different models trained for different surgical tasks (tissue dissection/suturing)? If so, it raises questions on full automation of the system as it would warrant manual video segmentation of surgical tasks before being fed to the SAIS model.
- While results on model generalization to videos and hospital is shown, the datasets appear to be severely controlled to meet the study requirement (same surgical tools). For instance, even for the same surgical task, the number and type of tools may vary depending on the surgical system used. While an out-of-distribution approach is used to handle unknown data, the work doesn't provide insights or acknowledge its limitation in handling new surgical tools for the same tasks.
- The degree of advancement in terms of the SAIS model is unclear due to a lack of fair comparison. The use of two non-public self-generated datasets also raises questions on model biases. To fairly benchmark the SAIS model, it must be evaluated against existing state-of-the-art task/gesture/surgeon skill recognition models on a public dataset.

//As numbered lists:

* Any major technical criticisms or questions.//

- As stated above, application merit is clear but technical novelty is unclear.

- The work presented as it has value in terms of application. Providing clear details on the dataset and SAIS model by rewriting the results and methodology section is necessary and will improve the paper. Additionally, benchmarking against state-of-the-art models on the public datasets in terms of performance, computational cost and ease of training and deployment will further validate the SAIS model and add value to the paper.

//Any minor technical criticisms or questions.//

This work uses the same ViT model to achieve three different tasks (phase recognition, gesture recognition, and skill assessment). Simply pursuing the number of tasks does not seem to justify more novelties. In this work, is the learning among these three tasks isolated and independent of each other (The proof is that the input video segments are different for surgical phase tasks and surgical gestures)? However, there is actually a connection between these tasks, so can the paper give insights into the progressive relationships?

Furthermore, based on the assumption that “the tissue dissection and suturing are commonly performed within almost any surgery”, when decoding suturing gestures (78 videos from USC) and dissection gestures (86 videos from USC), why did the authors choose different surgical videos for training?

The development of such a system is to be encouraged. It's good to see the integration of these tasks. The experimental design and methodology can be further improved to enrich the work.

//Any missing or unclear details about statistics, protocols or materials.//

- As the information on details on details and model is scattered between results and methodology section, it's difficult to comprehend fully on how the dataset is prepared and inputted to the model. Please clarify the following:

- * For phase/gesture/skill recognition tasks, If the full surgical videos are auto segmented into 1 second with 2 FPS, are each classification done using two frames? How does test-time augmentation contribute here?

- * It is actually unclear how many temporal frames are used for each classification task? In case more than 2 frames are used for each forward propagation, what is the computational requirement from start to end (extracting features from each frame to end classification)? Is the computational load justified compared to using existing state-of-the-art models?

- * I recommend rewriting the results and methodology. Simply results to just results and observation. Move the information in the results sections related to the dataset/methodology into the methodology section. In the methodology section, have a dataset and dataset preparation subsections. Under dataset, state the total number of videos generated and used for each task. Under dataset preparation, state how the video was segmented and augmented for each task. Under methodology, clearly define step-by-step forward propagation. How many frames were used, and how was it combined. Move the architecture figure to the methodology section for easy reference.

//Any missing citations to relevant literature (please keep in mind that the suggested maximum number of references is 50).//

1-3 Provides state-of-the-art models and public datasets. 4 is an attention-based model that is worth benchmarking.

- 1) CholecTriplet2021: A benchmark challenge for surgical action triplet recognition
- 2) PEG TRANSfer Workflow recognition challenge report: Does multi-modal data improve recognition?
- 3) Learning and Reasoning with the Graph Structure Representation in Robotic Surgery
- 4) Rendezvous: Attention mechanisms for the recognition of surgical action triplets in endoscopic videos

Many related and recent AI-engineering references on “Surgical Scenes understanding”, “Surgical Interaction Recognition”, “surgical report generation” (searching these keywords will yield the references) are missing.

//Any optional suggestions for improvement.//

- (1) may use the same set of surgical videos for these tasks and consider dependencies between tasks, rather than isolating them.

- (2) In addition to those mentioned in the paper, surgical evaluation can be evaluated from more perspectives. Such as: Whether the task can be completed within the expected time?

//Any stylistic issues or recommendations.//

- (1) can add some Tables to the paper
- (2) can polish Figure 2 by filling the text in green blocks.
- (3) Formulas can be centered

Reviewer #2 (Report for the authors (Required)):

I appreciate the opportunity to review your manuscript. I understand this is the experimental study to develop SAIS which can automatically recognize surgical phases (dissection or suturing), dissection gestures, sub-phases, and the skill level of sub-phases. SAIS might be able to provide surgeons with optimal feedback intraoperatively, and it might contribute to improving surgical skills. Ultimately, it might contribute to improving patients' postoperative outcomes. Several comments and questions are listed below.

The methodology of sub-phase recognition is clear; however, phase recognition is unclear. How many surgical phases did you define and annotate? I understand SAIS can recognize and distinguish each sub-phase with high accuracy but how high is the accuracy of the phase recognition task and which figure does show it? According to Supplementary Table 1, each fold had approximately 3800 samples, but what do samples mean and include? In my understanding, samples mean the target video scene, but if so, does it mean that each intraoperative video includes approximately 50 target scene? If each sample includes only the scene of NS, HD, and VUS with various time duration, you can extract only 3 scenes from each video, right? Besides, when you input the entire video into SAIS, can SAIS extract only the target scene from the entire video with high accuracy?

SAIS uses a ViT pre-trained in a self-supervised manner on the ImageNet dataset. In my understanding, the strong point of using a self-supervised pre-training manner is that you can use a big dataset with the same domain information for pre-training. In other words, ImageNet is just a general annotation dataset including various types of information such as foods, animals, vehicles, etc. However, by adopting a self-supervised manner for pre-training, you can use the big dataset of robotic surgical images in a self-supervised manner. Please explain the motivation and reason why you combined self-supervised pre-training manner and ImageNet in this study.

According to Figure 2, this network architecture is similar to I3D (3D-CNN). I strongly agree with your policy of selecting a spatiotemporal model for surgical video analysis tasks instead of the standard CNN model which can only analyze a static image. A previous study used I3D for surgical skill assessment (JAMA Netw Open. 2021 Aug 2;4(8):e2120786.), so you should lightly mention it in your text.

In the section on Qualitative evaluation, you just introduce only one case. I wonder if this tendency you realized as the outlier can adopt and apply to the other case. You set a 60-second interval as the threshold, but this threshold should be optimized based on the result of more cases. The rationale for the number 60 is ambiguous.

In the section on Generalizing across videos, you mentioned that for needle handling, skill assessment is based on the number of grip reposition times, and the fewer the better. If the manual skill assessment is performed strictly following this rule, you should develop a model which can recognize the grip reposition action and count the number of target actions. I think that skill assessment based on the results is highly reliable and interpretable compared to the proposed SAIS output.

In the section on Generalizing across hospitals, you mentioned the potential source of distribution shift includes variety in the camera recording devices between the surgical robots. However, I assume all surgical robot used in this study is DaVinci with the same camera. If not, please show the list of the type of surgical robots.

In the section on SAIS that can provide surgeons with actionable feedback, you mentioned that SAIS can allow surgeons to better focus on the element of intraoperative surgical activity that requires improvement. Please clarify whether your goal is intraoperative feedback or postoperative feedback. If yours is the former one, inference speed should be mentioned. I assume the network architecture which focuses on temporal information tends to take a bit longer inference time.

As you also mentioned in your discussion, the most crucial theme of this research field is how this

contributes to improving patient outcomes, and as for this point, I strongly agree with your opinion. I would like you to show the data about the correlation between SAIS output and erectile function or urinary incontinence even a preliminary one. The correlation between them will be truly observed or not so far.

Reviewer #3 (Report for the authors (Required)):

This paper presents a framework for multi-task surgical action classification. It uses the same neural network to estimate the phase of the surgery, the gesture being performed, and the skill at which the gesture is performed. Several datasets are curated, including two types of procedures in different anatomy and data from two hospitals. The proposed network uses a pretrained feature extractor to get frame-by-frame information from both RGB images and optical flow. The features are then stacked and passed through a transformer network, which uses self-attention to selectively process the video information. Lastly, the video features are aggregated between the RGB and optical branches and the resulting feature vector is used to classify 1) the surgical gesture, 2) the surgical phase, and 3) the skill at which this gesture is performed. The authors showed that not only can the same architecture be used across tasks, but it also generalizes across datasets collected at different institutions, with different practices. This is a good step in translational advance. The thorough evaluations of the proposed model make a good case that neural networks can obtain consistent and generalizable results in clinical settings. The results from this paper could be useful in introducing more machine learning-based analysis into surgery training programs, though it would have been more convincing if there was a user study showing how the analysis actually benefitted surgeon training.

Major technical criticisms or questions:

1) While there are interesting ablations of what features were important, the technical and performance-wise advance is somewhat lacking. The idea of using video analysis to assess surgical skill and gesture is not new, nor is it new to do both simultaneously, using a complex neural network followed by a simpler classifier (Khalid et al, 2020, Wu et al, 2021). The previous works were limited translationally since the publicly available datasets are on benchtop setups. Nevertheless, it would have been interesting to see a comparison to the previous methods on a consistent dataset to understand the technical contributions of this work. Comparisons with previous methods on the proposed dataset would be helpful in placing the proposed technical framework in the context of the existing body of work. Alternatively, the authors could provide results by using their method on the benchmark dataset in this field, JIGSAWS (Gao et al. 2014). Although JIGSAWS is a benchtop dataset rather than in vivo, evaluating on it would demonstrate the technical innovation better than solely evaluating on a proprietary dataset.

2) What is the benefit of using the same network for multiple tasks? There did not seem to be evidence suggesting that multi-task learning is actually more human-interpretable as the evaluation was mostly based on ROC curves. While it is hard to compare surgical gesture accuracy across different datasets, as different gestures are labelled at different granularity, state-of-the-art methods for surgical skill have accuracies above 90% (Funke et al. 2019, and Wang and Fey 2018, using video and robot kinematics respectively). The dataset for the previous works is simpler as it used data from phantoms rather than actual surgeries, but it does sort skills into three classes instead of two. There should be a discussion about what multi-task learning brings that is worth the trade-off for lower accuracy.

3) The paper is missing discussion on how camera views affect what the network learns. Shifting the camera should shift both the tool's position in the RGB image and the optical flow. Does the network see the same gesture in the two views as different examples of the gesture?

Minor technical criticisms or questions:

1) Natural images often place importance on corners and edges, both of which tend to be less distinct in endoscope images so it is interesting that this works. This is particularly true for the optical flow images, which should have very different characteristics compared to ImageNet. Since the stated goal of this manuscript is to build an explainable pipeline, it seems important to know what features a network trained on natural images picks up from endoscope images and optical flow. How was the ViT architecture determined? What was the self-supervised task on ImageNet?

2) One of the drawbacks of this method appears to be poor generalization to new anatomy (ex. from NS in RARP to HD in RAPN) despite both being dissection and sharing a common set of gestures. This suggests that the network focuses too much on the underlying image of the anatomy. While optical flow should provide some generalization over different backgrounds so it is unclear whether the drop is caused by image features or by true differences in how gestures are performed in the different procedures. It would be interesting to see whether incorporating robot motions into the analysis would benefit generalization. Since both are robot-assisted procedures, it should be possible to capture anatomy-independent motion cues from the joint information rather than obtaining it from optical flow.

3) It is unclear from the description whether videos from one surgeon can be in both the testing and training set (if two videos were performed by the same surgeon). If it could, there might be a leak of information between the two sets that is specific to how one surgeon performs a gesture (ex. if the network sees the surgeon performing a gesture in training and saw the label for the skill level, and then it sees the same gesture performed the same way during testing, the skill level is leaked).

4) At what time scale is skill level assessed? Gestures are decoded at 1 s intervals, at 2 fps. But from Fig. 6, more than 2 frames appear to be used for skill assessment, which makes sense since it should pick up on repetitions of a behaviour.

5) The authors suggest that differences in performance between the two sites may be partially due to different cameras. In that case, what are the characteristics of the cameras (ex. the dimensions of the images captured could have an effect on the artifacts created by scaling)?

Missing or unclear details about statistics:

How many actions were low skill level vs. high skill level and how consistent was this label for one surgeon?

Additional citations to relevant literature:

(Funke et al. 2019) Funke, I., Mees, S.T., Weitz, J. et al. Video-based surgical skill assessment using 3D convolutional neural networks. *Int J CARS* 14, 1217–1225 (2019). <https://doi.org/10.1007/s11548-019-01995-1>

(Gao et al. 2014) Gao, Y., Vedula, S. S., Reiley, C. E., Ahmidi, N., Varadarajan, B., Lin, H. C., ... & Hager, G. D. (2014, September). Jhu-isi gesture and skill assessment working set (jigsaws): A surgical activity dataset for human motion modeling. In *MICCAI workshop: M2cai* (Vol. 3, No. 3).

(Khalid et al. 2020) Khalid S, Goldenberg M, Grantcharov T, Taati B, Rudzicz F. Evaluation of Deep Learning Models for Identifying Surgical Actions and Measuring Performance. *JAMA Netw Open*. 2020;3(3):e201664. doi:10.1001/jamanetworkopen.2020.1664

(Wang and Fey 2018) Wang, Z., Majewicz Fey, A. Deep learning with convolutional neural network for objective skill evaluation in robot-assisted surgery. *Int J CARS* 13, 1959–1970 (2018). <https://doi.org/10.1007/s11548-018-1860-1>

(Wu et al. 2021) Wu, J., Tamhane, A., Kazanzides, P., & Unberath, M. (2021). Cross-modal self-supervised representation learning for gesture and skill recognition in robotic surgery. *International Journal of Computer Assisted Radiology and Surgery*, 16(5), 779-787.

Stylistic issues or recommendations:

1) It would be helpful to have a section describing the datasets.

Reviewer #4 (Report for the authors (Required)):

Summary:

This work proposes a machine learning model to classify multiple aspects of surgical activities. The authors

train and test their model on the classification of surgical phases, gestures and skill level. The authors evaluate the potential generalization capability of the model on videos from different surgeons, two hospitals and two different surgical procedures.

Degree of advance:

The authors demonstrate a method with some generalization capability by collecting data from different surgeons, two hospitals, and two different surgical procedures with their corresponding anatomical sites. The proposed architecture is relatively simple and can perform reasonably well on multiple classification tasks on multiple frames of a surgical video.

However, there is no comparison to other established machine learning models, so the difficulty of the proposed task and dataset vs. the capability of the model remain unknown.

The authors provide detailed demonstrations of the clinical utility their model could bring by decoding surgical activity. While interesting from a clinical translational point of view, the work however does not aim at demonstrating high technical novelty.

Implications:

The implications of this work are that the proposed machine learning model based on transformers is able to perform well and generalize on the classification of multiple classification tasks in isolated surgical activities. The performance on the proposed tasks seems satisfactory. However, the clinical applicability needs further validation.

Major technical criticisms:

1. Metrics like mAP and F1 Score are well established in the domain of surgical phase and action recognition and they should be reported here as well.

2. The demonstrated model is not evaluated on other datasets.

The dataset is unfortunately not public, therefore the results are not reproducible even though the model and code are to be published.

3. The authors claim that previous work commonly allows for the leakage of data from the same video into training and test set without any specific reference or example. This is incorrect. Avoiding data leakage is standard practice in the community and most of the referenced papers are doing so. If the authors are aware of any exceptions, they need to reference them explicitly and discuss the issues. Describing not allowing data leakage in this work as especially rigorous evaluation is therefore misleading.

4. Simplification of surgical events might not represent the true complexity of surgical interventions e.g. Skill level: High and Low.

Minor technical criticisms:

1. The relevance of steps, gestures and skill level on suturing and dissection activities to postoperative patient outcomes remains unknown until further studies in this direction are published.

2. It can be assumed that the impact of skill difference in the suturing step on patient outcome is limited. The impact of the quality of the dissection step can be assumed to be higher, but it is not analyzed here.

3. The authors proclaim that their model can already be reliably deployed on surgical videos of the nerve-sparing dissection step, but the performance on some of the classes e.g. in Fig. 5a does not seem sufficient for deployment in a clinical setting. (Recall ~ 0.5 and lower)

4. Parts of the surgical procedure (clips of suturing and dissection) are isolated manually, limiting automated analysis capability as proposed in this paper.

Stylistic Issues:

1. In figure 7. the caption and the paragraph below state that SAIS can provide skill information that otherwise would not be available - but this is also true for all surgical assessment methods.

2. In Discussion the first sentence says: "Only in the past decade or so has it been demonstrated that intraoperative surgical activity can have a direct influence on postoperative patient outcomes". This has been a known fact of surgery, in the last decades only the automatic analysis has been discussed.

Thu 27 Oct 2022

Decision on Article NBME-22-1333B

Dear Dr Kiyasseh,

Thank you for your patience in waiting for the feedback on your revised manuscript, "Decoding surgeon activity from surgical videos with a unified artificial intelligence system", which has been seen by the original reviewers. As noted in previous e-mail correspondence, despite our chasing efforts Reviewer #4 has not yet provided their feedback, and we feel that it is unlikely that they will.

The reports of the other three reviewers that I had already forward to you are included at the end of this message. The reviewers acknowledge the improvements to the work, and question the inherent technological novelty of the work. As I noted earlier, in particular with respect to the comments of Reviewer #1, for this manuscript editorially we have placed emphasis on the extended benchmarking and validation across datasets rather than on raw technical novelty or innovation. However, I hope that the next version of the manuscript can better frame the background of the story to take the reviewers' points into account.

As before, when you are ready to resubmit your manuscript, please upload the revised files, a point-by-point rebuttal to the comments from Reviewers #1 and #3, the reporting summary, and a cover letter that explains the main improvements included in the revision and responds to any points highlighted in this decision.

We look forward to receive a further revised version of the work. Please do not hesitate to contact me should you have any questions.

Best wishes,

Pep

Pep Pàmies
Chief Editor, Nature Biomedical Engineering

Reviewer #1 (Report for the authors (Required)):

I appreciate the efforts of the authors to revise the manuscript.

The key issues are still there regarding the lack of a significant degree of advance (fundamental, mechanistic, methodological, technological, therapeutic) in the work with respect to state of the art.

The authors themselves also agreed that "application merit is clear but technical novelty is unclear" and admitted that "the primary benefit of SAIS is translational in nature as it lends confidence into its use for live surgical videos stemming from distinct settings". However, more novel fundamental/mechanistic/methodological/technological/therapeutic contributions are typically expected for a publication appears in the top journal of "Nature Biomedical Engineering".

This work uses the same ViT model to achieve three different tasks (phase recognition, gesture recognition, and skill assessment). Simply pursuing the number of tasks does not seem to justify more novelties.

The learning among these three tasks (sub-phase recognition, gesture classification, and skill assessment) are independent of one another) are independent of each other. However, there are actually inherent connections between these tasks. The authors responded and would like to leave these discoveries to future work, which left many important questions unanswered.

Still, many related and recent AI-engineering references on Surgical Workflow Recognition, Surgical State Estimation, Surgical Scenes understanding, Surgical Interaction Recognition are missing.

Related ideas with original methodological algorithms have been published in the literature. For example:

Qin, Y., Allan, M., Burdick, J. W., & Azizian, M. (2021). Autonomous hierarchical surgical state estimation during robot-assisted surgery through deep neural networks. *IEEE Robotics and Automation Letters*, 6(4), 6220-6227. This paper proposed Hierarchical Estimation of Surgical States to estimate the associated super- and fine-grained states concurrently.

Shi, X., Jin, Y., Dou, Q., & Heng, P. A. (2021). Semi-supervised learning with progressive unlabeled data excavation for label-efficient surgical workflow recognition. *Medical Image Analysis*, 73, 102158.

Wagner, M., Müller-Stich, B. P., Kisilenko, A., Tran, D., Heger, P., Mündermann, L., ... & Bodenstedt, S. (2021). Comparative validation of machine learning algorithms for surgical workflow and skill analysis with the HeiChole benchmark. *arXiv preprint arXiv:2109.14956*.

Soleymani, A., Asl, A. A. S., Yeganejou, M., Dick, S., Tavakoli, M., & Li, X. (2021, November). Surgical skill evaluation from robot-assisted surgery recordings. In *2021 International Symposium on Medical Robotics (ISMR)* (pp. 1-6). IEEE.

Shi, X., Jin, Y., Dou, Q., & Heng, P. A. (2020). LRTD: long-range temporal dependency based active learning for surgical workflow recognition. *International Journal of Computer Assisted Radiology and Surgery*, 15(9), 1573-1584.

Seenivasan, L., Mitheran, S., Islam, M., & Ren, H. (2022). Global-Reasoned Multi-Task Learning Model for Surgical Scene Understanding. *IEEE Robotics and Automation Letters*, 7(2), 3858-3865.

van Amsterdam, B., Clarkson, M. J., & Stoyanov, D. (2020, May). Multi-task recurrent neural network for surgical gesture recognition and progress prediction. In *2020 IEEE International Conference on Robotics and Automation (ICRA)* (pp. 1380-1386). IEEE.

Zia, A., Guo, L., Zhou, L., Essa, I., & Jarc, A. (2019). Novel evaluation of surgical activity recognition models using task-based efficiency metrics. *International journal of computer assisted radiology and surgery*, 14(12), 2155-2163.

Gao, X., Jin, Y., Dou, Q., & Heng, P. A. (2020, May). Automatic gesture recognition in robot-assisted surgery with reinforcement learning and tree search. In *2020 IEEE International Conference on Robotics and Automation (ICRA)* (pp. 8440-8446). IEEE.

Reviewer #2 (Report for the authors (Required)):

Thank you for the opportunity to review the manuscript again.
My questions were answered clearly and appropriately by the authors.
I have no further comments.

Reviewer #3 (Report for the authors (Required)):

The authors have strengthened the clinical contribution of the work by clarifying the dataset description, adding another hospital, and adding the section correlating skill to clinical outcome. This is a valuable contribution to the field since curating datasets and long term studies to track patient outcome is often not possible depending on the institution. The translational contribution is useful as it does show that AI systems can help with targeting surgeon training, and potentially improving patient outcome. The authors have done a thorough job in addressing the concerns raised in the initial review.

The technical contribution of the work is still not obvious though. As mentioned by multiple reviewers, there is a lot of existing work on this topic and from Tables 1 and 2, it is unclear that the proposed method gets better performance than existing methods (even with other modalities that only use video). It's not clear to me that a percentage improvement over majority class is a convincing metric since the improvement seems to mostly

derive from Random doing much worse. This is especially concerning if G5 was discarded from the analysis for SAIS but included in MA-TCN. The architecture proposed does not seem fundamentally different than other transformer-based architecture (such as MA-TCN), combined with established ensemble methods. The value was that it was validated on data from multiple hospitals.

Tue 13 Dec 2022

Decision on Article NBME-22-1333C

Dear Dr Kiyasseh,

Thank you for your revised manuscript, "Decoding surgeon activity from surgical videos with a unified artificial intelligence system". Having consulted with Reviewers #1 and #3 (whose comments you will find at the end of this message), I am pleased to write that we shall be happy to publish the manuscript in *Nature Biomedical Engineering*.

The reviewers do not have any additional technical concerns, yet they do not feel that the manuscript provides sufficient technical innovation. We have taken their opinion into consideration. Yet, as outlined in our earlier decision e-mail, for this manuscript editorially we have placed emphasis on the extended benchmarking and validation across datasets rather than on raw technical novelty or innovation.

We will be performing detailed checks on your manuscript, and in due course will send you a checklist detailing our editorial and formatting requirements. You will need to follow these instructions before you upload the final manuscript files.

Best wishes,

Pep

Pep Pàmies
Chief Editor, Nature Biomedical Engineering

Reviewer #1 (Report for the authors (Required)):

I appreciate the efforts of the authors to revise the manuscript.

The responses and revisions still have the key issues: the lack of a significant degree of advance (fundamental, mechanistic, methodological, technological, therapeutic) in the work with respect to state of the art.

For the claim "Methodological Contribution 1 – SAIS is a unified AI system", the SOTA multi-task learning framework in the AI community and their adoption in decoding multiple elements of intraoperative surgical activity have been reported.

For the claim "Methodological Contribution 2 – SAIS provides explainable findings", the paper claimed "SAIS' has finer level of explainability, but such a claim lacks evidence or supporting data, nor details on this claim of "explainability" throughout the paper.

For the claim "Methodological Contribution 3 – SAIS is a flexible AI system", "primarily due to its transformer architecture," – this work just utilized the transformer architecture and inherited the related benefits rather than a significant degree of advance (fundamental, mechanistic, methodological, technological).

For the claim "Methodological Contribution 4 – SAIS is architecturally different from baseline methods," – the "architecture" here is still referring to "transformer architecture", so this work just utilized the transformer architecture and inherited the related benefits rather than a significant degree of advance (fundamental, mechanistic, methodological, technological).

Still, though directing readers towards more comprehensive review papers can make the manuscript more concise, the key and highly related recent references are worth discussing.

The inherent connections between multiple tasks are just briefly discussed in the Discussion section, rather than in-depth analysis.

Reviewer #3 (Report for the authors (Required)):

The updated manuscript's method has not changed from the previous revisions'. Therefore, there are still the same concerns about technical innovation. The discussion introduced in this revision largely highlights the translational advantages of the method, not technical innovation.

Rebuttal 1

We thank the reviewers for taking the time and effort to read our manuscript and provide us with valuable feedback.

In addition to addressing each of your comments, we have grouped those with a common theme and addressed them first. The three high-level themes include a:

- 1) better description of the datasets and methods
- 2) comparison of our model to baseline models
- 3) deployment of our model on publicly-available datasets

THEMES

Theme 1a – better description of the datasets

For ease of access and readability, we have now moved the description of the datasets used for training and evaluating models from the Results section to the Methods section. Specifically, the Methods section now contains the following modified sub-sections:

- *Description of the surgical procedures and activities* – this provides a high-level overview to the lay reader of the surgical procedures the specific surgical activities we are focusing on. By providing such context, readers can better appreciate the tasks that we plan to achieve: sub-phase recognition, gesture classification, and skill assessment. Methods → Description of surgical procedures and activities (page 11)
- *Surgical video samples and annotations* – this describes, in detail, our definition of a video sample and the process of annotating such video samples by trained human raters. Importantly, we have included a new table (**Table 4, page 12**) which summarizes all the datasets we have used in the manuscript. In doing so, readers can easily refer to the table when reading the appropriate parts of the Results section. For further clarification on the duration of the video samples, we present, in **Supplementary Note 2**, a distribution of such durations for video samples across hospitals and suturing sub-phases: needle handling, needle driving, and needle withdrawal. Methods → Surgical video samples and annotations (page 11)

Task	Activity	Details	Hospital	Videos	Video Samples	Surgeons	Generalizing to
Sub-phase recognition	Suturing	VUA	USC	78	4774	19	videos
			SAH	60	2115	8	hospitals
			HMH	20	1122	5	hospitals
			USC	48	inference on entire videos		
Gesture classification	Suturing	VUA	USC	78	1241	19	videos
		Laboratory	JIGSAWS	39	793	8	users
		DVC	UCL	36	1378	9	videos
	Dissection	NS	USC	86	1542	15	videos
			SAH	60	540	8	hospitals
			USC	154	inference on entire unlabelled videos		
	RAPN	USC	27	339	16	procedures	
Skill assessment	Suturing	Needle handling	USC	78	912	19	videos
			SAH	60	240	18	hospitals
			HMH	20	184	5	hospitals
	Needle driving	USC	78	530	19	videos	
		SAH	60	280	18	hospitals	
		HMH	20	220	5	hospitals	

Table 4. Total number of videos and video samples associated with each of the hospitals and tasks. Note that we train our model, SAIS, on data exclusively shown in **green** following a 10-fold Monte Carlo cross-validation setup. For an exact breakdown of the number of video samples in each fold and training, validation, and test split, please refer to Supplementary Tables 1-6. The data from the remaining hospitals are exclusively used for inference. We perform inference on entire videos shown in **yellow**. Except for the task of sub-phase recognition, SAIS is always trained and evaluated on a class-balanced set of data whereby each category (e.g., low skill and high skill) contains the same number of samples. This prevents SAIS from being negatively affected by a sampling bias during training, and allows for a more intuitive appreciation of the evaluation results.

Supplementary Note 2 - Duration of video samples

In this section, we present the distribution of the duration of video samples used for training and evaluating SAIS' ability to decode surgical sub-phases and the skill-level of surgeons (Fig. 1). These are shown for the three suturing sub-phases of needle handling, needle driving, and needle withdrawal (columns) for the different hospitals: USC, SAH, and HMH (rows). As we can see, the video samples can span 5 – 100 seconds.

Supplementary Figure 1. Distribution of the duration of video samples for the three sub-phases and across hospitals. Each row reflects a different hospital. Each column reflects a different suturing sub-phase: needle handling, needle driving, and needle withdrawal. We see that video samples can span 5 – 100 seconds.

Theme 1b – better description of the methods

We have now moved the description of our model (SAIS) from the Results section to the Methods section. Considering the reader's convenience, we have also moved the corresponding figure of the model to the Methods section. Specifically, the Methods section now contains the following modified sub-sections:

- *Single forward pass through SAIS* – this describes, through a series of steps, the mechanism of a single forward pass of data through the SAIS model, from extracting spatial features, to aggregating such features over time, and making the final prediction. These descriptions are also more consistent with the summary figure (**Figure 7, page 14**) enabling readers to more easily map the descriptive text to the figure. **Methods → SAIS is a model for decoding surgeon activity from surgical videos → Single forward pass through SAIS (page 13)**
- *Implementation details of inference on entire videos* – this describes, also through a series of steps, how we went about performing inference with SAIS on entire surgical videos with minimal human annotations. We provide such descriptions for the sub-phase recognition task and the gesture classification task. These details should allow a machine learning practitioner to replicate our approach. **Methods → Implementation details of inference on entire videos (page 16)**

Theme 2 – comparison of our model to baseline models

We now compare our model to baseline models in the following ways:

- *Sub-phase recognition* – we take the suggestion of several reviewers to compare against the Inception3D (I3D) model and thus train I3D to perform sub-phase recognition. We then deploy it on entire videos to decode surgical sub-phases and compare its performance to that of SAIS using the F1@10 metric (see **Figure 2e**), a commonly-used metric for predicting temporal segments. **Results → SAIS reliably decodes surgical sub-phases → Benchmarking against baseline models (page 3)**

- *Skill assessment* – given the success of I3D in surgical skill assessment, we also train it to distinguish between low and high-skill surgical activity. We compare its performance to that of SAIS in **Table 3** (page 8). **Results → SAIS reliably decodes surgical skills → Benchmarking against baseline models (page 7)**

Activity	Hospital	I3D ⁹	SAIS (ours)
needle handling	USC	0.681 (0.07)	0.849 (0.06)
	SAH	0.730 (0.04)	0.880 (0.02)
	HMH	0.680 (0.04)	0.804 (0.03)
needle driving	USC	0.630 (0.12)	0.821 (0.05)
	SAH	0.656 (0.08)	0.797 (0.04)
	HMH	0.571 (0.07)	0.719 (0.06)

Table 3. SAIS outperforms state-of-the-art model when decoding the skill-level of surgical activity. SAIS is trained on video samples exclusively from USC. We report the average AUC (± 1 standard deviation) on the test-set of each of the 10 folds. Bold indicates the better-performing method.

Theme 3 – deployment of our model on publicly-available datasets

We now train our model on publicly-available datasets and compare its performance to state-of-the-art models on those datasets. Specifically, we *demonstrate competitive performance* with baseline models when SAIS is tasked with distinguishing between suturing gestures on the:

- *JIGSAWS dataset* – this dataset of laboratory-based videos has long been the benchmark dataset of sorts in the realm of surgery. These results are presented in **Table 1** and compared to the best-performing methods that use different data modalities (kinematics, video, etc.). **Results → SAIS reliably decodes surgical gestures → Validating on external video datasets (page 5)**
- *DVC UCL dataset* – this is a recently-released dataset of live surgical videos depicting suturing gestures. These results are presented in **Table 2** and compared to the best-performing reported method. **Results → SAIS reliably decodes surgical skills → Benchmarking against baseline models (page 7)**

Method	Accuracy	Modalities
Fusion-KV ¹¹	86.3	Video + Kinematics
MS-RNN ¹²	90.2	Kinematics
Sym. Dilation ¹³	90.1	Video
SAIS (ours)	87.5 (13.0)	Video

Table 1. Accuracy of gesture classification on the JIGSAWS suturing dataset. We report the accuracy of the best-performing methods⁹ evaluated using leave-one-user-out (LOUO) cross-validation and in each modality category.

Method	Accuracy (%)		
	Random	Reported	Improved
MA-TCN ¹⁵	25.9	80.9	3.1×
SAIS (ours)	14.3	59.8 (1.0)	4.2×

Table 2. Accuracy of gesture classification on the DVC UCL dataset. MA-TCN reports accuracy on a private test-set with gesture imbalance. We report the average cross-validation accuracy on the publicly-available training set with balanced categories. Bold indicates the better-performing method.

Additions to the manuscript beyond the above themes

We have since added results, beyond those related to the above themes, to the manuscript. These include results:

- 1) With data from a new hospital (Houston Methodist) to further demonstrate the generalizability of our findings
- 2) For distinguishing between the dissection and suturing steps to demonstrate that the full workflow for automation of surgeon performance assessment is achievable
- 3) For a preliminary analysis associated SAIS' outputs with patient outcomes to further emphasize the clinical significance of our work

ADDITIONS**Addition 1**

Specifically, to further demonstrate the generalizability of our model to videos from unseen surgeons at distinct hospitals, we now include data from a third and entirely new hospital: Houston Methodist Hospital (HMH). These data are used for:

- *Sub-phase recognition* – **Figure 2c** now depicts the performance of SAIS when decoding surgical sub-phases on video samples from HMH. We demonstrate consistently strong performance across hospitals. **Results → SAIS reliably decodes surgical sub-phases → Generalizing across hospitals (page 3)**

- *Skill assessment* – **Figure 5a and c** (page 7) now depict the performance of SAIS when decoding the skill-level of surgical activity on video samples from HMH. We demonstrate

consistently strong performance across hospitals. **Results → SAIS reliably decodes surgical skills → Generalizing across hospitals (page 7)**

Addition 2

We provide evidence that SAIS can comfortably distinguish between the tissue dissection and tissue suturing steps of a surgery. Specifically, we demonstrate that SAIS achieves an AUC = 1 when distinguishing between the nerve-sparing (NS) dissection step and the vesico-urethral anastomosis (VUA) step (**Supplementary Note 3 → Supplementary Figure 2**). We had hypothesized, prior to conducting these experiments, that such an initial task (phase recognition) compared to downstream tasks (e.g., sub-phase recognition or gesture classification) would be relatively straightforward as the various steps of surgery are often more distinguishable based on visual cues compared to the more subtle elements of surgery (e.g., sub-phases, gestures, etc.).

Supplementary Figure 2. SAIS reliably decodes surgical phases across videos. SAIS is trained on video samples exclusively from USC and also evaluated on video samples from USC. Results are shown as an average (± 1 standard deviation) of 10 Monte-Carlo cross-validation steps.

Addition 3

We have also included a preliminary analysis associating SAIS' skill assessments with postoperative patient outcomes. Specifically, we demonstrate a statistically-significant relationship between the surgeons' needle driving skill assessments provided by SAIS and patients' 3-month urinary continence recovery. Further large-scale studies are required to cement this relationship. **Results → SAIS' skills assessments are associated with patient outcomes (page 9)**

SAIS' skill assessments are associated with patient outcomes

While useful for mastering a surgical technical skill itself, surgeon feedback becomes more clinically meaningful when grounded in patient outcomes. For example, if low-skill assessments are associated with poor outcomes, then a surgeon can begin to modulate specific behaviour to improve such outcomes. To that end, we conducted a preliminary analysis regressing SAIS' surgeon skill assessments at USC onto a patient's binary recovery of urinary continence (ability to voluntarily control urination) 3 months after surgery (see details in Methods). When considering all video samples (multiple per surgical case), and controlling for surgeon caseload and patient age, we found that urinary continence recovery was 1.31 (odds ratio, CI: 1.08 – 1.58, $p = 0.005$) more likely when needle driving was assessed as high-skill than as low-skill by SAIS. When aggregating the skill assessments of video samples within a surgical case, that relationship is further strengthened (odds ratio: 1.89 CI: 0.95 – 3.76, $p = 0.071$). These preliminary findings are consistent with those based on *manual* skill assessments from recent studies^{18,19}.

POINT-BY-POINT RESPONSE

Reviewer 1

Summary

The authors propose a unified surgical AI system that can decode surgical activities. This work developed a system named SAIS which can identify the procedural steps of surgery, the actions performed by a surgeon, and the quality of such actions by utilizing two data modalities (RGB frames and optical flow maps). The proposed system can be used to decode surgical steps (phase), gestures and the surgeon's skill. Extensive experiments are conducted to show models' performance and robustness in generalizing across videos, hospitals and surgical produces in performing those tasks. The author's writing in the introduction section is commendable as it adequately stresses the need and advantage of such application. This work has merits in terms of application. However, it needs major revision in the results and methodology section to help readers understand it easily. The details on the dataset, dataset preparation and proposed model are scattered between the results and methodology section making it hard to fully comprehend. Furthermore, the proposed system lacks significant novelty and benchmarks against the existing system. The workload of this work in preparing the dataset is heavy. However, the work lacks comparative experiments and technical novelty.

R1 – Comment 1

This work showcases application novelty, where similar architectures (the last layer of the architecture changes depending on the number of classes for each task) for phase/gesture/skill recognition. While the results provided are original and interesting, the degree of advancement in terms of the model is unclear due to the following:

Based on my understanding, the proposed SAIS model is a single-task (pseudo-tri-task) model, which can be trained to recognise phase/gesture/surgeon skills. However, to perform all three tasks at once requires three SAIS models in parallel. In such a case, how different is the SAIS model compared to existing gesture recognition models, which can also be technically used for the remaining tasks since they also warrant only changing the last layer? What is the advantage of using multiple SAIS models in parallel vs using the existing state-of-the-art models for phase/gesture/skill recognition in parallel? Additional comparison of computation requirements will also provide additional insights.

Response to R1 – Comment 1

The reviewer's understanding is correct in that three unique SAIS models would be needed to decode the three elements of surgical activity that we have presented: sub-phases, gestures, and skills. We consider SAIS a *unified* architecture because it is capable of reliably decoding all of these elements. We now more explicitly outline the advantages of a *unified* architecture (see Discussion section, page 10, paragraph 1).

Compared to previous studies (see previous work section), SAIS offers a multitude of benefits. First, it is unified in that it is capable of decoding multiple elements of intraoperative surgical activity. This characteristic may reduce the cognitive burden and resources expended by machine learning practitioners in searching for and developing AI systems specialized to a specific element of surgical activity. Instead, SAIS can be the dependable core architecture upon which further modifications are made. Second, SAIS is rigorously evaluated in that it generalizes across videos, surgeons, surgical procedures, and hospitals. This highlights its translational value and is in contrast to most previous work limited to experimenting with videos captured in a controlled laboratory setting. Third, SAIS provides explainable predictions by highlighting the temporal frames it deems most important. Such explainability can be critical to gaining the trust of surgeons and to ensuring the safe deployment of AI systems for high-stakes decision making such as skill-based surgeon credentialing.

We also provide additional insight into the advantages of SAIS relative to existing state-of-the-art models like Inception3D (I3D) and 3D convolutional neural networks (3D-CNNs) more generally. In short, these advantages include the flexibility of SAIS in dealing with mini-batches of video samples of different sizes, facilitating transfer learning, and its ability to provide explanations (see Discussion section, page 10, paragraph 2).

SAIS also offers greater flexibility in how surgical videos can be processed compared to existing models, such as 3D-CNNs. This is in large part due to the transformer architecture. Specifically, previous work which leverages a pre-trained 3D-CNN is constrained to video samples with either 16 frames or multiples thereof^{9,16}. This is because such models were originally pre-trained with 16-frame video samples from the Kinetics dataset²⁰. Deviating from this design choice prevents one from exploiting the pre-trained parameters, which have proven critical to the success of previous methods. In contrast, video samples input into SAIS can consist of any number of frames. Ordinarily, such flexibility can pose a challenge for AI systems that must often be trained using a mini-batch of equally-sized samples. SAIS, however, can seamlessly process such video samples (see implementation details). This also confers benefits to transfer learning as researchers looking to leverage SAIS and its parameters are no longer constrained to a fixed-size video sample, which may be sub-optimal for achieving a particular task.

We also compare our model to existing models for the tasks of sub-phase recognition, gesture classification, and skill assessment. Importantly, we demonstrate that SAIS outperforms an existing state-of-the-model, Inception3D (I3D), for sub-phase recognition and skill assessment (see Results section, page 3 and page 8).

Activity	Hospital	I3D ⁹	SAIS (ours)
needle handling	USC	0.681 (0.07)	0.849 (0.06)
	SAH	0.730 (0.04)	0.880 (0.02)
	HMH	0.680 (0.04)	0.804 (0.03)
needle driving	USC	0.630 (0.12)	0.821 (0.05)
	SAH	0.656 (0.08)	0.797 (0.04)
	HMH	0.571 (0.07)	0.719 (0.06)

Table 3. SAIS outperforms state-of-the-art model when decoding the skill-level of surgical activity. SAIS is trained on video samples exclusively from USC. We report the average AUC (± 1 standard deviation) on the test-set of each of the 10 folds. Bold indicates the better-performing method.

R1 – Comment 2

Are different models trained for different surgical tasks (tissue dissection/suturing)? If so, it raises questions on full automation of the system as it would warrant manual video segmentation of surgical tasks before being fed to the SAIS model.

Response to R1 – Comment 2

Yes, a different SAIS model is trained to decode a different element of surgical activity. Its ability to reliably decode these various elements, though, opens the door for future research to look into decoding all elements of surgical activity with a single SAIS model, akin to a multi-task framework.

As for full automation, we do not make any claims in the manuscript that SAIS will fully automate the decoding of *all* elements of surgical activity for *all* videos. Although SAIS will require being fed videos of a particular surgical step (e.g., nerve-sparing dissection of the radical prostatectomy), identifying this video manually only requires the provision of two timestamps (the start and end time of the nerve-sparing dissection step), which is not time-consuming or laborious.

If, for some reason, providing these two time-stamps happens to be a bottleneck for researchers, then we also provide additional evidence that SAIS can comfortably distinguish between the tissue dissection and tissue suturing steps of a surgery. Specifically, we demonstrate that SAIS achieves an AUC = 1 when distinguishing between the nerve-sparing (NS) dissection step and the vesico-urethral anastomosis (VUA) step (Supplementary Note 3 → Supplementary Figure 2). We had hypothesized, prior to conducting the experiments, that such a task (phase recognition) would be relatively straightforward as the various steps of surgery are often more distinguishable based on visual cues compared to the more subtle elements of surgery (e.g., sub-phases, gestures, etc.).

Supplementary Figure 2. SAIS reliably decodes surgical phases across videos. SAIS is trained on video samples exclusively from USC and also evaluated on video samples from USC. Results are shown as an average (± 1 standard deviation) of 10 Monte-Carlo cross-validation steps.

Once the video of a surgical step is provided to SAIS, it can then automatically decode the various elements of surgical activity. We had originally presented such an example of decoding from an entire nerve-sparing video in the context of dissection gesture classification (Figure 4). We now also present such a decoding from an entire vesico-urethral anastomosis suturing video in the context of sub-phase recognition (Figure 2e). We provide details of this inference in Methods → Implementation details of inference on entire videos.

R1 – Comment 3

While results on model generalization to videos and hospital is shown, the datasets appear to be severely controlled to meet the study requirement (same surgical tools). For instance, even for the same surgical task, the number and type of tools may vary depending on the surgical system used. While an out-of-distribution approach is used to handle unknown data, the work doesn't provide insights or acknowledge its limitation in handling new surgical tools for the same tasks.

Response to R1 – Comment 3

We agree that surgery can be unpredictable and that the surgical field of view may exhibit content a model has never seen before. While our video samples from a particular task (e.g., gesture classification) are from a particular step (e.g., nerve-sparing dissection step), we do not explicitly constrain such video samples to only reflect a particular type or number of surgical tools. Although there can be exceptions, as with the clipping gesture which is often only performed with the same surgical tool, this is likely to be representative of what naturally occurs during surgery.

We acknowledge that SAIS, and many other surgical AI systems, are unlikely to be exposed to the full scope of the variability of surgical videos. To that end, when decoding elements of surgical activity from entire videos, we had adopted a state-of-the-art out-of-distribution (OOD) approach to explicitly deal with video samples that SAIS might be uncertain about. The exact details of this OOD method are provided in Methods → implementation details of inference on entire videos → abstaining from prediction. We now also acknowledge the limitations of SAIS in recognizing novel surgical activity (see Discussion section, page 10, paragraph 4).

There are important challenges our work does not yet address. First, our framework, akin to others in the field, is limited to only decoding the elements of surgical activity that have been previously outlined in some taxonomy (e.g., gestures). In other words, it cannot decode what it does not know. Although many of these taxonomies have been rigorously developed by teams of surgeons and through clinical experience, they may fail to shed light on other intricate aspects of surgical activity. This, in turn, limits the degree to which automated systems can *discover* novel activity that falls beyond the realm of existing protocol. Such discovery can lend insight into, for example, optimal but as-of-yet undiscovered surgical behaviour. In a similar vein, SAIS is currently incapable of decoding new elements of surgical activity beyond those initially presented to it. Such continual learning capabilities¹⁰ are critical to adapting to an evolving taxonomy of surgical activity over time.

R1 – Comment 4

The degree of advancement in terms of the SAIS model is unclear due to a lack of fair comparison.

The use of two non-public self-generated datasets also *raises questions on model biases*. To fairly benchmark the SAIS model, it must be evaluated against existing state-of-the-art task/gesture/surgeon skill recognition models on a public dataset.

Response to R1 – Comment 4

While evaluating SAIS on a privately-held dataset might raise questions about model biases, our rigorous evaluation of SAIS on video samples from unseen surgeons at two distinct hospitals (with a third hospital added in the revision) should help alleviate that concern. Generalizing to such video samples, which is what we have demonstrated, is indicative of a model which is less likely to be latching onto surgeon-specific or hospital-specific features in the data.

Nonetheless, we have now compared SAIS to several baseline models on both our privately-held datasets (for sub-phase recognition and skill assessment), and on two publicly-available datasets (JIGSAWS and DVC UCL). We demonstrate that SAIS performs competitively with baseline methods on these two datasets. We hope these results allay the reviewer's concerns about model biases.

R1 – Comment 5

As stated above, application merit is clear but technical novelty is unclear.

Response to R1 – Comment 5

We agree that the primary benefit of SAIS is translational in nature as it lends confidence into its use for live surgical videos stemming from distinct settings (surgeons, hospitals, and surgical procedures). This sends a signal to the broader community that AI systems can reliably cope with such videos. While the use of transformer architectures, dual-modality inputs (RGB and optical flow), and prototypes for classification are not new in and of themselves, their combination has resulted in a reliable system for decoding surgical activity, whose ramifications can be far-reaching.

R1 – Comment 6

The work presented as it has value in terms of application. Providing clear details on the dataset and SAIS model by rewriting the results and methodology section is necessary and will improve the paper. Additionally, benchmarking against state-of-the-art models on the public datasets in terms of performance, computational cost and ease of training and deployment will further validate the SAIS model and add value to the paper.

Response to R1 – Comment 6

We have taken the reviewer's suggestion of rewriting the Results and Methods sections in order to improve clarity. Please refer to the Themes section at the top of this rebuttal for details on how these sections have since been modified.

R1 – Comment 7

This work uses the same ViT model to achieve three different tasks (phase recognition, gesture recognition, and skill assessment). Simply pursuing the number of tasks does not seem to justify more novelties. In this work, is the learning among these three tasks isolated and independent of each other (The proof is that the input video segments are different for surgical phase tasks and

surgical gestures)? However, there is actually a connection between these tasks, so can the paper give insights into the progressive relationships?

Response to R1 – Comment 7

Yes, the learning of the three tasks (sub-phase recognition, gesture classification, and skill assessment) are independent of one another. As the reviewer pointed out, it is likely that videos of different surgical steps (e.g., nerve-sparing and vesico-urethral anastomosis) within the same surgery (robot-assisted radical prostatectomy) do indeed exhibit temporal and complex relationships with one another. We hypothesize that such relationships may also hold for the distinct tasks. For example, how gestures are performed during the nerve-sparing dissection step may be associated with the skill-level of the suturing activity performed by the surgeon in subsequent suturing steps. While our team has explored relationships between subsequent suturing activities, future work could begin to unearth relationships between different steps of surgery.

R1 – Comment 8

Furthermore, based on the assumption that “the tissue dissection and suturing are commonly performed within almost any surgery”, when decoding suturing gestures (78 videos from USC) and dissection gestures (86 videos from USC), why did the authors choose different surgical videos for training? The development of such a system is to be encouraged. It's good to see the integration of these tasks. The experimental design and methodology can be further improved to enrich the work.

Response to R1 – Comment 8

While nerve-sparing dissection steps and vesico-urethral suturing steps appear in almost every RARP surgical procedure, we chose to work with a different subset of videos (which partially overlap) for suturing and dissection. This is because such videos were originally annotated for the purpose of associating intraoperative activity to postoperative outcomes (e.g., erectile function and urinary continence) which our team has published on recently [1], [2]. We have now improved the description of the datasets used and the methods (please refer to Themes section at the top of the rebuttal for details on that).

R1 – Comment 9

As the information on details on details and model is scattered between results and methodology section, it's difficult to comprehend fully on how the dataset is prepared and inputted to the model. Please clarify the following:

For phase/gesture/skill recognition tasks, If the full surgical videos are auto segmented into 1 second with 2 FPS, are each classification done using two frames? How does test-time augmentation contribute here?

Response to R1 – Comment 9

First, to improve clarity, we have included the dataset description only in the Methods section. We have also now provided a better description of the datasets, method, and implementation details (see Methods section). In **Table 4**, for example, we outline the total number of video samples available for each task. Further, in **Supplementary Note 2**, we outline the average duration, in seconds, of these video samples.

As an illustrative example, given a video sample with a frame-rate of 30Hz and is 10s long, that individual video sample would consist of 300 frames. We sample a subset of these frames (see description below) and input them into SAIS. As such, test-time augmentation (TTA) can still take place by temporally offsetting the sampled frames. A description of this is now provided in Methods → implementation details and hyperparameters (page 15).

Implementation details and hyperparameters

During training and inference, we use the *start time* and *end time* of each video sample to guide the selection of video frames from that sample. For gesture classification, we select 10 equally-spaced frames from the video sample. For example, for a video sample with a frame-rate of 30Hz and that is 3s long, then from the original $30 \times 3 = 90$ frames, we would only retrieve frames $\in [0, 9, 18, \dots]$. In contrast, for sub-phase recognition and skill assessment, we select every other 10^{th} frame. For example, for the same video sample above, we would only retrieve frames $\in [0, 10, 20, \dots]$. We found that these strategies resulted in a good trade-off between computational complexity and capturing sufficiently informative signals in the video to complete the task. Similarly, optical flow maps were based on pairs of images that were 0.5 seconds apart. Shorter time-spans resulted in frames which exhibited minimal motion and thus uninformative flow maps. During training, to ensure that the RGB and optical flow maps were associated with the same time-span, we retrieved maps that overlapped in time with the RGB frames. During inference, and for test-time augmentation, we offset both RGB and optical flow frames by $K = 3$ and $K = 6$ frames.

R1 – Comment 10

It is actually unclear how many temporal frames are used for each classification task? In case more than 2 frames are used for each forward propagation, what is the computational requirement from start to end (extracting features from each frame to end classification)? Is the computational load justified compared to using existing state-of-the-art models?

Response to R1 – Comment 10

As outlined in the answer to Comment 9 above, we address this comment in Methods → implementation details and hyperparameters. In short, yes, each video sample contains more than 2 frames.

In the same section, we provide insight into how we pre-process these frames *offline* so that the training and inference process of SAIS is computationally light (e.g., by freezing the parameters of the ViT feature extractor, and extracting and storing frame representations before conducting experiments).

We conduct our experiments in PyTorch⁴⁶ using a V100 GPU on a DGX machine. Each RGB frame and optical flow map was resized to 224×224 (from 960×540 at USC and SAH and 1920×1080 at SAH) before being input into the ViT feature extractor. The ViT feature extractor pre-processed each frame into a set of square patches of dimension $H = 16$ and generated a frame representation of dimension $D = 384$. All video representations and prototypes are of dimension $E = 256$. In practice, we froze the parameters of the ViT, extracted all such representations offline (i.e., before training), and stored them as h5py files. We followed the same strategy for extracting representations of optical flow maps. This significantly reduced the typical bottleneck associated with loading videos and streamlined our training and inference process. This also facilitates inference performed on future videos. Once a new video is recorded, its features can immediately be extracted in an offline manner, and stored for future use.

R1 – Comment 11

I recommend rewriting the results and methodology. Simply results to just results and observation. Move the information in the results sections related to the dataset/methodology into the methodology section. In the methodology section, have a dataset and dataset preparation subsections. Under dataset, state the total number of videos generated and used for each task. Under dataset preparation, state how the video was segmented and augmented for each task. Under methodology, clearly define step-by-step forward propagation. How many frames were used, and how was it combined. Move the architecture figure to the methodology section for easy reference.

Response to R1 – Comment 11

We have now incorporated all of the reviewer's suggestions into the manuscript. Please refer to the Themes section at the top of the rebuttal for a detailed description of how we responded to this comment.

R1 – Comment 12

1-3 Provides state-of-the-art models and public datasets. 4 is an attention-based model that is worth benchmarking.

- 1) CholecTriplet2021: A benchmark challenge for surgical action triplet recognition
- 2) PEG TRansfer Workflow recognition challenge report: Does multi-modal data improve recognition?
- 3) Learning and Reasoning with the Graph Structure Representation in Robotic Surgery
- 4) Rendezvous: Attention mechanisms for the recognition of surgical action triplets in endoscopic videos

Many related and recent AI-engineering references on "Surgical Scenes understanding", "Surgical Interaction Recognition", "surgical report generation" (searching these keywords will yield the references) are missing.

Response to R1 – Comment 12

We thank the reviewer for bringing these studies to our attention. We have now included an additional number of relevant references to the manuscript, highlighting previous public datasets and corresponding networks. Methods → Previous work → Computational methods (page 11).

R1 – Comment 13

May use the same set of surgical videos for these tasks and consider dependencies between tasks, rather than isolating them. In addition to those mentioned in the paper, surgical evaluation can be evaluated from more perspectives. Such as: whether the task can be completed within the expected time?

Response to R1 – Comment 13

We agree that surgeons and surgical activity can be evaluated in a multitude of ways. While we have evaluated surgical activity in a handful of ways, we believe our framework can be reliably depended on by future researchers looking to evaluate the surgical activity in their videos through whatever lens they wish.

R1 – Comment 14

- (1) can add some Tables to the paper
- (2) can polish Figure 2 by filling the text in green blocks.
- (3) Formulas can be centered

Response to R1 – Comment 14

We have now included several tables in the manuscript (**Tables 1-4**) to summarize results and facilitate the comparison of methods. We have also modified Figure 1 to fill in the green blocks with text. Formulas will be centred upon publication (our latex style file overrides any centering of equations at the moment).

Reviewer 2

Summary

I appreciate the opportunity to review your manuscript. I understand this is the experimental study to develop SAIS which can automatically recognize surgical phases (dissection or suturing), dissection gestures, sub-phases, and the skill level of sub-phases. SAIS might be able to provide surgeons with optimal feedback intraoperatively, and it might contribute to improving surgical skills. Ultimately, it might contribute to improving patients' postoperative outcomes. Several comments and questions are listed below.

R2 – Comment 1

The methodology of sub-phase recognition is clear; however, **phase recognition is unclear**. How many surgical phases did you define and annotate? I understand SAIS can recognize and distinguish each sub-phase with high accuracy but how high is the accuracy of the phase recognition task and which figure does show it? According to Supplementary Table 1, each fold had approximately 3800 samples, **but what do samples mean and include?** In my understanding, samples mean the target video scene, but if so, does it mean that each intraoperative video includes approximately 50 target scenes? If each sample includes only the scene of NS, HD, and VUS with various time duration, you can extract only 3 scenes from each video, right? Besides, **when you input the entire video into SAIS, can SAIS extract only the target scene from the entire video with high accuracy?**

Response to R2 – Comment 1

To clarify, we had not initially performed *phase recognition* and had instead only performed *suturing sub-phase recognition*. We have since removed any mention of phase recognition to avoid confusion and consistently used the term sub-phase recognition throughout the manuscript. The results for sub-phase recognition can be found in **Figure 2**. However, new results to distinguish between two steps of surgery (dissection and suturing) are now provided in **Supplementary Note 3**, demonstrating strong performance (AUC = 1), indicating the relatively trivial nature of this task.

Supplementary Figure 2. SAIS reliably decodes surgical phases across videos. SAIS is trained on video samples exclusively from USC and also evaluated on video samples from USC. Results are shown as an average (± 1 standard deviation) of 10 Monte-Carlo cross-validation steps.

We have since modified the description of the datasets and methods to improve clarity. Please refer to the Themes section at the top of the rebuttal for details on how. In short, a video sample is now defined in **Methods** → **surgical video samples and annotations (page 11)**, and its average duration can be found in **Supplementary Note 2**.

Supplementary Note 2 - Duration of video samples

In this section, we present the distribution of the duration of video samples used for training and evaluating SAIS' ability to decode surgical sub-phases and the skill-level of surgeons (Fig. 1). These are shown for the three suturing sub-phases of needle handling, needle driving, and needle withdrawal (columns) for the different hospitals: USC, SAH, and HMH (rows). As we can see, the video samples can span 5 – 100 seconds.

Supplementary Figure 1. Distribution of the duration of video samples for the three sub-phases and across hospitals. Each row reflects a different hospital. Each column reflects a different suturing sub-phase: needle handling, needle driving, and needle withdrawal. We see that video samples can span 5 – 100 seconds.

With regards to extracting the “target scene”, as the reviewer has described it, SAIS is capable of decoding the element of surgical activity based on an entire video of a surgical step (e.g., nerve-sparing dissection). For example, we had initially provided evidence (**Figure 4**) that one can provide SAIS with an entire nerve-sparing video (which only requires the minimal effort of providing a start and end time-stamp by a human) and it will extract all the dissection gestures for you (precision of up to 0.80).

If, for some reason, providing these two time-stamps (start and end) happen to be a bottleneck for researchers, then we also provide additional evidence that SAIS can comfortably distinguish between the tissue dissection and tissue suturing steps of a surgery. Specifically, we demonstrate that SAIS achieves an AUC = 1 when distinguishing between the nerve-sparing (NS) dissection step and the vesico-urethral anastomosis (VUA) step (**Supplementary Note 3 → Supplementary Figure 2**). We had hypothesized, prior to conducting the experiments, that such a task (phase recognition) would be relatively straightforward as the various steps of surgery are often more distinguishable based on visual cues compared to the more subtle elements of surgery (e.g., sub-phases, gestures, etc.).

Supplementary Figure 2. SAIS reliably decodes surgical phases across videos. SAIS is trained on video samples exclusively from USC and also evaluated on video samples from USC. Results are shown as an average (± 1 standard deviation) of 10 Monte-Carlo cross-validation steps.

We have since provided additional evidence in **Figure 2e** (page 3) that one can provide SAIS with an entire vesico-urethral video and it will extract all the suturing sub-phases for you.

R2 – Comment 2

SAIS uses a ViT pre-trained in a self-supervised manner on the ImageNet dataset. In my understanding, the strong point of using a self-supervised pre-training manner is that you can use a big dataset with the same domain information for pre-training. In other words, ImageNet is just a general annotation dataset including various types of information such as foods, animals, vehicles, etc. However, by adopting a self-supervised manner for pre-training, **you can use the big dataset of robotic surgical images in a self-supervised manner. Please explain the motivation and reason why you combined self-supervised pre-training manner and ImageNet in this study.**

Response to R2 – Comment 2

We agree that self-supervised models can be used to pre-train on a large corpus of surgical videos. The motivation for leveraging a ViT pre-trained on ImageNet in a self-supervised manner in this study is twofold (**Methods → Single forward pass through SAIS → Extracting spatial features, page 13**). First, recent evidence has pointed the superior performance of *self-supervised* models pre-trained on ImageNet relative to *supervised* models pre-trained on ImageNet. Second, ease of use; pre-training on surgical videos can be a computationally expensive process that requires a large enough corpus of data in order to result in the learning of meaningful representations. As we begin to curate ever-larger datasets of surgical videos, we look forward to pre-training on such datasets in the future as a means of learning more meaningful surgery-specific representations.

Vision transformers convert each input frame into a set of square image patches of dimension $H \times H$ and introduce a self-attention mechanism that attempts to capture the relationship between image patches (i.e., spatial information). We found that this spatial attention picks up on instrument tips, needles, and anatomical edges (see Fig. 8). We chose this feature extractor based on (a) recent evidence favouring self-supervised pre-trained models relative to their supervised counterparts and (b) the desire to reduce the computational burden associated with training a feature extractor in an end-to-end manner.

R2 – Comment 3

According to Figure 2, this network architecture is **similar to I3D (3D-CNN)**. I strongly agree with your policy of selecting a spatiotemporal model for surgical video analysis tasks instead of the standard CNN model which can only analyze a static image. A previous study used I3D for surgical skill assessment (JAMA Netw Open. 2021 Aug 2;4(8):e2120786.), so you should lightly mention it in your text.

Response to R2 – Comment 3

We have since mentioned the I3D network, and due to its apparent similarity to SAIS, we have compared its performance to that of our model on multiple tasks (sub-phase recognition and skill assessment). For sub-phase recognition results, please refer to **Results → SAIS reliably decodes surgical sub-phases → Benchmarking against baseline models (page 3)**. For skill assessment results, please refer to **Results → SAIS reliably decodes surgical skills → Benchmarking against baseline models (page 7)**. In both of these settings, we demonstrate that SAIS outperforms the I3D network.

Activity	Hospital	I3D ⁹	SAIS (ours)
needle handling	USC	0.681 (0.07)	0.849 (0.06)
	SAH	0.730 (0.04)	0.880 (0.02)
	HMH	0.680 (0.04)	0.804 (0.03)
needle driving	USC	0.630 (0.12)	0.821 (0.05)
	SAH	0.656 (0.08)	0.797 (0.04)
	HMH	0.571 (0.07)	0.719 (0.06)

Table 3. SAIS outperforms state-of-the-art model when decoding the skill-level of surgical activity. SAIS is trained on video samples exclusively from USC. We report the average AUC (± 1 standard deviation) on the test-set of each of the 10 folds. Bold indicates the better-performing method.

R2 – Comment 4

In the section on Qualitative evaluation, you just introduce only one case. I wonder if this tendency you realized as the outlier can adopt and apply to the other case. You set a 60-second interval as the threshold, but this threshold should be optimized based on the result of more cases. The rationale for the number 60 is ambiguous.

Response to R2 – Comment 4

In **Methods → SAIS provides surgical gesture information otherwise unavailable to surgeons**, we provide both a quantitative and qualitative analysis. The quantitative analysis suggests that SAIS often identifies the correct dissection gesture. To clarify, for the qualitative analysis, we do not define a-priori the 60-second interval that the reviewer is mentioning here. Instead, we performed inference with SAIS on the entire nerve-sparing video using video samples 1-second in duration, and it just so happened that SAIS discovered a 60-second interval which it classified as a camera move. It

is entirely conceivable that other outliers in other videos may be shorter or longer in duration. We provide a better description of the inference process in Methods → implementation details of inference on entire videos → nerve-sparing dissection gesture classification (page 16).

R2 – Comment 5

In the section on Generalizing across videos, you mentioned that for needle handling, skill assessment is based on the number of grip reposition times, and the fewer the better. If the manual skill assessment is performed strictly following this rule, you should develop a model which can recognize the grip reposition action and count the number of target actions. I think that skill assessment based on the results is highly reliable and interpretable compared to the proposed SAIS output.

Response to R2 – Comment 5

While counting the number of repositions is a possible way to assess skill, we opted to follow a skill assessment scoring system (known as EASE) rigorously developed through a Delphi process with expert surgeons and subsequently validated. Doing so legitimizes the skill assessments that SAIS outputs and, because of the strict guidelines that the scoring system follows, these outputs remain interpretable the user on the receiving end of the AI-based skill assessments.

R2 – Comment 6

In the section on Generalizing across hospitals, you mentioned the potential source of distribution shift includes variety in the camera recording devices between the surgical robots. However, I assume all surgical robot used in this study is DaVinci with the same camera. If not, please show the list of the type of surgical robots.

Response to R2 – Comment 6

The reviewer's assumption is correct in that the surgical videos are recording daVinci machines in operation. We are now more precise in describing the potential sources of distribution shift across hospitals. These sources are less related to the camera hardware itself and more related to, for example, the behaviour of surgeons which is unique to hospitals and the content in the surgical field of view (e.g. more or less blood).

R2 – Comment 7

In the section on SAIS that can provide surgeons with actionable feedback, you mentioned that SAIS can allow surgeons to better focus on the element of intraoperative surgical activity that requires improvement. Please clarify whether your goal is intraoperative feedback or postoperative feedback. If yours is the former one, inference speed should be mentioned. I assume the network architecture which focuses on temporal information tends to take a bit longer inference time.

Response to R2 – Comment 7

One of the potential downstream use-cases of SAIS is the provision of postoperative feedback to surgeons. In other words, a surgery is performed, a video is recorded, that video is analysed after surgery, and feedback is then provided to a surgeon at a suitable time. This goal therefore reduces the need for very high inference speeds. Having said that, because of the way we pre-process the frames (e.g., by extracting representations offline), we find that we can decode the elements of

surgical activity from videos quite quickly. For example, decoding dissection gestures from 100 nerve-sparing videos can be achieved in under 6 hours. Moreover, our approach for performing inference means every time a new video is recorded, its features can simply be extracted and stored for future inference.

R2 – Comment 8

As you also mentioned in your discussion, the most crucial theme of this research field is how this contributes to improving patient outcomes, and as for this point, I strongly agree with your opinion. I would like you to show the data about the correlation between SAIS output and erectile function or urinary incontinence even a preliminary one. The correlation between them will be truly observed or not so far.

Response to R2 – Comment 8

To address the reviewer's suggestion, we have now included an additional section **Results → SAIS' skill assessments are associated with patient outcomes (page 9)**. Here, we provide preliminary evidence that demonstrated the association of SAIS' skill assessments for needle driving with urinary continence recovery 3 months after surgery. Although we found a statistically significant relationship, even after controlling for potential confounding factors, further large-scale studies would be required to cement this relationship. A description of this experiment is provided in **Methods → Association between SAIS' skill assessments and patient outcomes (page 18)**.

SAIS' skill assessments are associated with patient outcomes

While useful for mastering a surgical technical skill itself, surgeon feedback becomes more clinically meaningful when grounded in patient outcomes. For example, if low-skill assessments are associated with poor outcomes, then a surgeon can begin to modulate specific behaviour to improve such outcomes. To that end, we conducted a preliminary analysis regressing SAIS' surgeon skill assessments at USC onto a patient's binary recovery of urinary continence (ability to voluntarily control urination) 3 months after surgery (see details in Methods). When considering all video samples (multiple per surgical case), and controlling for surgeon caseload and patient age, we found that urinary continence recovery was 1.31 (odds ratio, CI: 1.08 – 1.58, $p = 0.005$) more likely when needle driving was assessed as high-skill than as low-skill by SAIS. When aggregating the skill assessments of video samples within a surgical case, that relationship is further strengthened (odds ratio: 1.89 CI: 0.95 – 3.76, $p = 0.071$). These preliminary findings are consistent with those based on *manual* skill assessments from recent studies^{18,19}.

Reviewer 3

Summary

This paper presents a framework for multi-task surgical action classification. It uses the same neural network to estimate the phase of the surgery, the gesture being performed, and the skill at which the gesture is performed. Several datasets are curated, including two types of procedures in different anatomy and data from two hospitals. The proposed network uses a pretrained feature extractor to get frame-by-frame information from both RGB images and optical flow. The features are then stacked and passed through a transformer network, which uses self-attention to selectively process the video information. Lastly, the video features are aggregated between the RGB and optical branches and the resulting feature vector is used to classify 1) the surgical gesture, 2) the surgical phase, and 3) the skill at which this gesture is performed. The authors showed that not only can the same architecture be used across tasks, but it also generalizes across datasets collected at different institutions, with different practices. This is a good step in translational advance. The thorough evaluations of the proposed model make a good case that neural networks can obtain consistent and generalizable results in clinical settings. The results from this paper could be useful in introducing more machine learning-based analysis into surgery training programs, though it would have been more convincing if there was a user study showing how the analysis actually benefitted surgeon training.

R3 – Comment 1

While there are interesting ablations of what features were important, the technical and performance-wise advance is somewhat lacking. The idea of using video analysis to assess surgical skill and gesture is not new, nor is it new to do both simultaneously, using a complex neural network followed by a simpler classifier (Khalid et al, 2020, Wu et al, 2021). The previous works were limited translationally since the publicly available datasets are on benchtop setups. Nevertheless, it would have been interesting to see a comparison to the previous methods on a consistent dataset to understand the technical contributions of this work. Comparisons with previous methods on the proposed dataset would be helpful in placing the proposed technical framework in the context of the existing body of work. Alternatively, the authors could provide results by using their method on the benchmark dataset in this field, JIGSAWS (Gao et al. 2014). Although JIGSAWS is a benchtop dataset rather than in vivo, evaluating on it would demonstrate the technical innovation better than solely evaluating on a proprietary dataset.

Response to R3 – Comment 1

We agree with the reviewer that previous work has been limited from a translational standpoint. As a community, and in particularly within healthcare, we should strive toward developing AI systems that exhibit the characteristics requisite for deployment amongst our target stakeholders. To that end, we hope SAIS contributes to this mission as evident by its reliable performance on *live* surgical videos across hospitals.

To contextualize SAIS, we now compare it to multiple baseline models across the tasks. Specifically, we incorporate both of the reviewer's suggestions into the manuscript. First, we compare SAIS to previous methods (Inception3D) on our privately-held datasets for sub-phase recognition (**Figure 2e, page 3**) and skill assessment (**Table 3, page 8**), demonstrating that SAIS outperforms I3D in both of these settings. Second, we compare SAIS to the best-performing models on two publicly-available

datasets (JIGSAWS and DVC UCL), demonstrating competitive performance on both of these datasets (Tables 1-2). We hope these findings provide further insight into the utility of our framework.

Activity	Hospital	I3D ⁹	SAIS (ours)
needle handling	USC	0.681 (0.07)	0.849 (0.06)
	SAH	0.730 (0.04)	0.880 (0.02)
	HMH	0.680 (0.04)	0.804 (0.03)
needle driving	USC	0.630 (0.12)	0.821 (0.05)
	SAH	0.656 (0.08)	0.797 (0.04)
	HMH	0.571 (0.07)	0.719 (0.06)

Table 3. SAIS outperforms state-of-the-art model when decoding the skill-level of surgical activity. SAIS is trained on video samples exclusively from USC. We report the average AUC (± 1 standard deviation) on the test-set of each of the 10 folds. Bold indicates the better-performing method.

Method	Accuracy	Modalities
Fusion-KV ¹¹	86.3	Video + Kinematics
MS-RNN ¹²	90.2	Kinematics
Sym. Dilation ¹³	90.1	Video
SAIS (ours)	87.5 (13.0)	Video

Table 1. Accuracy of gesture classification on the JIGSAWS suturing dataset. We report the accuracy of the best-performing methods⁸ evaluated using leave-one-user-out (LOUO) cross-validation and in each modality category.

Method	Accuracy (%)		
	Random	Reported	Improved
MA-TCN ¹⁵	25.9	80.9	3.1×
SAIS (ours)	14.3	59.8 (1.0)	4.2×

Table 2. Accuracy of gesture classification on the DVC UCL dataset. MA-TCN reports accuracy on a private test-set with gesture imbalance. We report the average cross-validation accuracy on the publicly-available training set with balanced categories. Bold indicates the better-performing method.

R3 – Comment 2

What is the benefit of using the same network for multiple tasks? There did not seem to be evidence suggesting that multi-task learning is actually more human-interpretable as the evaluation was mostly based on ROC curves. While it is hard to compare surgical gesture accuracy across different datasets, as different gestures are labelled at different granularity, state-of-the-art methods for surgical skill have accuracies above 90% (Funke et al. 2019, and Wang and Fey 2018, using video and robot kinematics respectively). The dataset for the previous works is simpler as it used data from phantoms rather than actual surgeries, but it does sort skills into three classes instead of two. There should be a discussion about what multi-task learning brings that is worth the trade-off for lower accuracy.

Response to R3 – Comment 2

To clarify, SAIS is *not* a multi-task learning framework, although, with minor modifications it can be trained as one. Instead, one would train 3 distinct SAIS models to decode the 3 distinct elements of surgical activity (sub-phases, gestures, skills). Figure 1b demonstrates this through the repetition of the SAIS model box 3 times. Further, we do not claim that the interpretability of the model stems from the supposed multi-task nature of SAIS. Instead, its interpretability stems from the temporal attention mechanism used as part of its network architecture.

We nonetheless provide a lengthy description of the advantages of SAIS relative to existing models in the **Discussion section (page 10)**. To enable a comparison to existing state-of-the-art models, we have demonstrated that SAIS outperforms I3D on the task of skill assessment (**Table 3, page 8**). This allows readers to have an “apples-to-apples” comparison of the performance of the various methods when trained and evaluated on exact same dataset.

Compared to previous studies (see previous work section), SAIS offers a multitude of benefits. First, it is unified in that it is capable of decoding multiple elements of intraoperative surgical activity. This characteristic may reduce the cognitive burden and resources expended by machine learning practitioners in searching for and developing AI systems specialized to a specific element of surgical activity. Instead, SAIS can be the dependable core architecture upon which further modifications are made. Second, SAIS is rigorously evaluated in that it generalizes across videos, surgeons, surgical procedures, and hospitals. This highlights its translational value and is in contrast to most previous work limited to experimenting with videos captured in a controlled laboratory setting. Third, SAIS provides explainable predictions by highlighting the temporal frames it deems most important. Such explainability can be critical to gaining the trust of surgeons and to ensuring the safe deployment of AI systems for high-stakes decision making such as skill-based surgeon credentialing.

SAIS also offers greater flexibility in how surgical videos can be processed compared to existing models, such as 3D-CNNs. This is in large part due to the transformer architecture. Specifically, previous work which leverages a pre-trained 3D-CNN is constrained to video samples with either 16 frames or multiples thereof^{6,16}. This is because such models were originally pre-trained with 16-frame video samples from the Kinetics dataset²⁰. Deviating from this design choice prevents one from exploiting the pre-trained parameters, which have proven critical to the success of previous methods. In contrast, video samples input into SAIS can consist of any number of frames. Ordinarily, such flexibility can pose a challenge for AI systems that must often be trained using a mini-batch of equally-sized samples. SAIS, however, can seamlessly process such video samples (see implementation details). This also confers benefits to transfer learning as researchers looking to leverage SAIS and its parameters are no longer constrained to a fixed-size video sample, which may be sub-optimal for achieving a particular task.

R3 – Comment 3

The paper is missing discussion on how camera views affect what the network learns. Shifting the camera should shift both the tool's position in the RGB image and the optical flow. Does the network see the same gesture in the two views as different examples of the gesture?

Response to R3 – Comment 3

While we had not explicitly discussed the effect of camera views on performance, we had provided results on the effect of the anatomical location of the nerve-sparing dissection step (left vs. right) relative to the prostate gland on dissection gesture classification (see **Figure 4a**). We outline in **Results → SAIS provides surgical gesture information otherwise unavailable to surgeons (page 6)** how videos of these different anatomical locations do indeed capture gestures from a different angle and are thus akin to different camera views. By demonstrating equivalent performance across the anatomical locations, we are in effect demonstrating SAIS' robustness to camera angles.

We further stratified this precision based on the anatomical location of the neuro-vascular bundle relative to the prostate gland. This allowed us to determine whether SAIS was (a) learning an unreliable shortcut to decoding gestures by associating anatomical landmarks with certain gestures, which is undesirable, and (b) robust to changes in the camera angle and direction of motion of the gesture. For the latter, note that operating on the *left* neuro-vascular bundle often involves using the right-hand instrument and moving it towards the left of the field of view (see Fig. 4b top row of images). The opposite is true when operating on the *right* neuro-vascular bundle.

We found that SAIS is unlikely to be learning an anatomy-specific shortcut to decoding gestures and is robust to the direction of motion of the gesture. This is evident by its similar performance when deployed on video samples of gestures performed in the left and right neuro-vascular bundles. For example, hook (h) gesture predictions exhibited precision of ≈ 0.75 in both anatomical locations. We also observed

R3 – Comment 4

Natural images often place importance on corners and edges, both of which tend to be less distinct in endoscope images so it is interesting that this works. This is particularly true for the optical flow images, which should have very different characteristics compared to ImageNet. Since the stated goal of this manuscript is to build an explainable pipeline, it seems important to know what features a network trained on natural images picks up from endoscope images and optical flow. How was the ViT architecture determined? What was the self-supervised task on ImageNet?

Response to R3 – Comment 4

We have now provided a brief description of the self-supervised task the ViT model was pre-trained on. In short, it is a contrastive learning task in which representations of augmented versions of the same image are encouraged to be similar to one another (pre-training method is called DINO). The motivation for the ViT architecture itself is provided in Methods → Single forward pass through SAIS → Extracting spatial features (page 14). We also include Figure 8 (page 15) which shows that the ViT model places high importance on instrument tips, needles, and anatomical edges.

Figure 8. Vision transformer feature extractor places the highest importance on instrument tips, needles, and anatomical edges. We present two sample RGB video frames of the needle handling activity and the corresponding spatial attention placed by ViT on patches of these frames.

R3 – Comment 5

One of the drawbacks of this method appears to be poor generalization to new anatomy (ex. from NS in RARP to HD in RAPN) despite both being dissection and sharing a common set of gestures. This

suggests that the network focuses too much on the underlying image of the anatomy. While optical flow should provide some generalization over different backgrounds so it is unclear whether the drop is caused by image features or by true differences in how gestures are performed in the different procedures. It would be interesting to see whether incorporating robot motions into the analysis would benefit generalization. Since both are robot-assisted procedures, it should be possible to capture anatomy-independent motion cues from the joint information rather than obtaining it from optical flow.

Response to R3 – Comment 6

While kinematics data can reduce a model's dependence on visual cues, potentially making it more robust to misleading and confusing content in the surgical field of view, such data face two overarching limitations. First, kinematics data are not accessible to the vast majority of researchers without a special arrangements with the robot manufacturer. Second, kinematics data present their own unique challenges in terms of the degree of noise they exhibit, the need for signal processing techniques to alleviate such noise, and so forth. Having said that, SAIS would conceivably be able to incorporate additional data modalities such as kinematics into its pipeline.

R3 – Comment 7

It is unclear from the description whether videos from one surgeon can be in both the testing and training set (if two videos were performed by the same surgeon). If it could, there might be a leak of information between the two sets that is specific to how one surgeon performs a gesture (ex. if the network sees the surgeon performing a gesture in training and saw the label for the skill level, and then it sees the same gesture performed the same way during testing, the skill level is leaked).

Response to R3 – Comment 7

Yes, distinct videos from the same surgeon *may* appear in both the training and testing set but only when we are in the *generalizing across videos* evaluation setting (see Methods → Motivation behind evaluating SAIS with Monte Carlo cross validation → Data splits, page 13). Although this could reflect the leakage of surgeon data across sets and, in turn, provide an over-estimate of model performance, we see that SAIS continues to perform well even when deployed, in the more rigorous setting, on video samples from unseen surgeons at distinct hospitals, as reflected by the *generalizing across hospitals* evaluation setup.

Data splits

For all the experiments conducted, unless otherwise noted, we split the data at the *case video level* into a training (90%) and test set (10%). We used 10% of the videos in the training set to form a validation set with which we performed hyperparameter tuning. By splitting at the video-level, whereby data from the same video do not appear across the sets, we are rigorously evaluating whether the model generalizes across unseen videos. Note that while it is possible for data from the same surgeon to appear in both the training and test sets, we also experiment with even more rigorous setups: *across hospitals* - where videos are from entirely different hospitals and surgeons, and *across surgical procedures* - where videos are from entirely different surgical procedures (e.g., nephrectomy vs. prostatectomy). While there are various ways to rigorously evaluate surgical AI systems, we do believe that demonstrating their generalizability across surgeons, hospitals, and surgical procedures, as we have done, is a step in the right direction. We report the performance of models as an

To clarify, once again, SAIS is not a multi-task learning framework and therefore will not “see” the labels from different tasks at the same time during training. For example, it will not see the label for skill-level and gesture at the same time.

R3 – Comment 8

At what time scale is skill level assessed? Gestures are decoded at 1 s intervals, at 2 fps. But from Fig. 6, more than 2 frames appear to be used for skill assessment, which makes sense since it should pick up on repetitions of a behaviour.

Response to R3 – Comment 8

We provide additional details about the video samples used in Methods → surgical video samples and annotations (page 11). We also provide information about the number of frames used in each video sample in Methods → Implementation details and hyperparameters (page 15). Lastly, we present the average duration of such video samples in Supplementary Note 2. In short, each video sample is on the order of 10-30 seconds long and can therefore span 300 frames. Details of the inference process are now provided in Methods → Implementation details of inference on entire videos (page 16).

Implementation details and hyperparameters

During training and inference, we use the *start time* and *end time* of each video sample to guide the selection of video frames from that sample. For gesture classification, we select 10 equally-spaced frames from the video sample. For example, for a video sample with a frame-rate of 30Hz and that is 3s long, then from the original $30 \times 3 = 90$ frames, we would only retrieve frames $\in [0, 9, 18, \dots]$. In contrast, for sub-phase recognition and skill assessment, we select every other 10th frame. For example, for the same video sample above, we would only retrieve frames $\in [0, 10, 20, \dots]$. We found that these strategies resulted in a good trade-off between computational complexity and capturing sufficiently informative signals in the video to complete the task. Similarly, optical flow maps were based on pairs of images that were 0.5 seconds apart. Shorter time-spans resulted in frames which exhibited minimal motion and thus uninformative flow maps. During training, to ensure that the RGB and optical flow maps were associated with the same time-span, we retrieved maps that overlapped in time with the RGB frames. During inference, and for test-time augmentation, we offset both RGB and optical flow frames by $K = 3$ and $K = 6$ frames.

Supplementary Note 2 - Duration of video samples

In this section, we present the distribution of the duration of video samples used for training and evaluating SAIS' ability to decode surgical sub-phases and the skill-level of surgeons (Fig. 1). These are shown for the three suturing sub-phases of needle handling, needle driving, and needle withdrawal (columns) for the different hospitals: USC, SAH, and HMH (rows). As we can see, the video samples can span 5 – 100 seconds.

Supplementary Figure 1. Distribution of the duration of video samples for the three sub-phases and across hospitals. Each row reflects a different hospital. Each column reflects a different suturing sub-phase: needle handling, needle driving, and needle withdrawal. We see that video samples can span 5 – 100 seconds.

R3 – Comment 9

The authors suggest that differences in performance between the two sites may be partially due to different cameras. In that case, what are the characteristics of the cameras (ex. the dimensions of the images captured could have an effect on the artifacts created by scaling)?

Response to R3 – Comment 9

We are now more precise in describing the potential sources of distribution shift across hospitals. These sources are less related to the camera hardware itself and more related to, for example, the behaviour of surgeons which is unique to hospitals and the content in the surgical field of view (e.g. more or less blood). We also include the dimensionality of the frames of videos collected from the different hospitals in Methods → implementation details and hyperparameters.

Generalizing across hospitals To measure the degree to which SAIS can generalize to unseen surgeons at a distinct hospital, we deployed it on video samples from SAH (Fig. 3c, video sample count in Table 4). We found that SAIS continues to perform well in such a setting. For example, $AUC = 0.899$ and 0.831 for the camera move (m) and clip (k) gestures, respectively. Importantly, such a finding suggests that SAIS can be reliably deployed on data with several sources of variability (surgeon, hospital, etc.). We expected, and indeed observed, a slight degradation in performance in this setting relative to when SAIS was deployed on video samples from USC. For example, $AUC = 0.823 \rightarrow 0.702$ for the cold cut (c) gesture in the USC and SAH data, respectively. This was expected due to the potential shift in the distribution of data collected across the two hospitals, which has been documented to negatively affect network performance¹⁰. Potential sources of distribution shift include variability in how surgeons perform the same set of gestures (e.g., different techniques) and in the surgical field of view (e.g., clear view with less blood). Furthermore, our hypothesis for why this degradation affects certain gestures (e.g., cold cuts) more than others (e.g., clips) is that the latter exhibits less variability than the former, and is thus easier to classify by the model.

R3 – Comment 10

How many actions were low skill level vs. high skill level and how consistent was this label for one surgeon?

Response to R3 – Comment 10

We now include the total number of video samples in **Table 4** (page 12). As mentioned in the caption of Table 4, we always train and evaluate, unless otherwise noted, on a balanced dataset. This means that for skill assessment 50% of the video samples are low-skill and the other 50% are high-skill. We also provide the motivation behind this setup in Table 4's caption.

Task	Activity	Details	Hospital	Videos	Video Samples	Surgeons	Generalizing to
Sub-phase recognition	Suturing	VUA	USC	78	4774	19	videos
			SAH	60	2115	8	hospitals
			HMH	20	1122	5	hospitals
			USC	48	inference on entire videos		
Gesture classification	Suturing	VUA	USC	78	1241	19	videos
		Laboratory	JIGSAWS	39	793	8	users
		DVC	UCL	36	1378	9	videos
	Dissection	NS	USC	86	1542	15	videos
			SAH	60	540	8	hospitals
			USC	154	inference on entire unlabelled videos		
	RAPN	USC	27	339	16	procedures	
Skill assessment	Suturing	Needle handling	USC	78	912	19	videos
			SAH	60	240	18	hospitals
			HMH	20	184	5	hospitals
		Needle driving	USC	78	530	19	videos
			SAH	60	280	18	hospitals
			HMH	20	220	5	hospitals

Table 4. Total number of videos and video samples associated with each of the hospitals and tasks. Note that we train our model, SAIS, on data exclusively shown in green following a 10-fold Monte Carlo cross-validation setup. For an exact breakdown of the number of video samples in each fold and training, validation, and test split, please refer to Supplementary Tables 1-6. The data from the remaining hospitals are exclusively used for inference. We perform inference on entire videos shown in yellow. Except for the task of sub-phase recognition, SAIS is always trained and evaluated on a class-balanced set of data whereby each category (e.g., low skill and high skill) contains the same number of samples. This prevents SAIS from being negatively affected by a sampling bias during training, and allows for a more intuitive appreciation of the evaluation results.

R3 – Comment 11

Additional citations to relevant literature:

(Funke et al. 2019) Funke, I., Mees, S.T., Weitz, J. et al. Video-based surgical skill assessment using 3D convolutional neural networks. *Int J CARS* 14, 1217–1225 (2019). <https://doi.org/10.1007/s11548-019-01995-1>

(Gao et al. 2014) Gao, Y., Vedula, S. S., Reiley, C. E., Ahmidi, N., Varadarajan, B., Lin, H. C., ... & Hager, G. D. (2014, September). Jhu-isi gesture and skill assessment working set (jigsaws): A surgical activity dataset for human motion modeling. In *MICCAI workshop: M2cai* (Vol. 3, No. 3).

(Khalid et al. 2020) Khalid S, Goldenberg M, Grantcharov T, Taati B, Rudzicz F. Evaluation of Deep Learning Models for Identifying Surgical Actions and Measuring Performance. *JAMA Netw Open*. 2020;3(3):e201664. doi:10.1001/jamanetworkopen.2020.1664

(Wang and Fey 2018) Wang, Z., Majewicz Fey, A. Deep learning with convolutional neural network for objective skill evaluation in robot-assisted surgery. *Int J CARS* 13, 1959–1970 (2018). <https://doi.org/10.1007/s11548-018-1860-1>

(Wu et al. 2021) Wu, J., Tamhane, A., Kazanzides, P., & Unberath, M. (2021). Cross-modal self-supervised representation learning for gesture and skill recognition in robotic surgery. *International Journal of Computer Assisted Radiology and Surgery*, 16(5), 779-787.

Response to R3 – Comment 11

We thank the reviewer for bringing these studies to our attention. We have included relevant references in Methods → Previous work → Computational methods (page 11).

R3 – Comment 12

It would be helpful to have a section describing the datasets.

Response to R3 – Comment 12

We now describe the datasets in Methods → Description of surgical procedures and activities and Methods → surgical video samples and annotations (page 11). The datasets are also summarized in a new **Table 4**.

Reviewer 4

Summary

This work proposes a machine learning model to classify multiple aspects of surgical activities. The authors train and test their model on the classification of surgical phases, gestures and skill level. The authors evaluate the potential generalization capability of the model on videos from different surgeons, two hospitals and two different surgical procedures.

R4 – Comment 1

The authors demonstrate a method with some generalization capability by collecting data from different surgeons, two hospitals, and two different surgical procedures with their corresponding anatomical sites. The proposed architecture is relatively simple and can perform reasonably well on multiple classification tasks on multiple frames of a surgical video. However, there is no comparison to other established machine learning models, so the difficulty of the proposed task and dataset vs. the capability of the model remain unknown. The authors provide detailed demonstrations of the clinical utility their model could bring by decoding surgical activity. While interesting from a clinical translational point of view, the work however does not aim at demonstrating high technical novelty.

Response to R4 – Comment 1

We agree with the reviewer that the primary advantage of SAIS is its translational nature. As a community, and in particularly within healthcare, we should strive toward developing AI systems that exhibit the characteristics requisite for deployment amongst our target stakeholders. To that end, we hope SAIS robustly contributes to this mission as evident by its reliable performance on live surgical videos across hospitals.

To contextualize SAIS, we now compare it to multiple baseline models across the tasks. Specifically, we incorporate both of the reviewer's suggestions into the manuscript. First, we compare SAIS to previous methods (Inception3D) on our privately-held datasets for sub-phase recognition (**Figure 2e**) and skill assessment (**Table 3, page 8**), demonstrating that SAIS outperforms I3D in both of these settings. Second, we compare SAIS to the best-performing models on two publicly-available datasets (JIGSAWS and DVC UCL), demonstrating competitive performance on both of these datasets (**Tables 1-2, page 5**). We hope these findings provide further insight into the utility of our framework.

Activity	Hospital	I3D ⁹	SAIS (ours)
needle handling	USC	0.681 (0.07)	0.849 (0.06)
	SAH	0.730 (0.04)	0.880 (0.02)
	HMH	0.680 (0.04)	0.804 (0.03)
needle driving	USC	0.630 (0.12)	0.821 (0.05)
	SAH	0.656 (0.08)	0.797 (0.04)
	HMH	0.571 (0.07)	0.719 (0.06)

Table 3. SAIS outperforms state-of-the-art model when decoding the skill-level of surgical activity. SAIS is trained on video samples exclusively from USC. We report the average AUC (± 1 standard deviation) on the test-set of each of the 10 folds. Bold indicates the better-performing method.

Method	Accuracy	Modalities
Fusion-KV ¹¹	86.3	Video + Kinematics
MS-RNN ¹²	90.2	Kinematics
Sym. Dilation ¹³	90.1	Video
SAIS (ours)	87.5 (13.0)	Video

Table 1. Accuracy of gesture classification on the JIGSAWS suturing dataset. We report the accuracy of the best-performing methods⁸ evaluated using leave-one-user-out (LOUO) cross-validation and in each modality category.

Method	Accuracy (%)		
	Random	Reported	Improved
MA-TCN ¹⁵	25.9	80.9	3.1×
SAIS (ours)	14.3	59.8 (1.0)	4.2×

Table 2. Accuracy of gesture classification on the DVC UCL dataset. MA-TCN reports accuracy on a private test-set with gesture imbalance. We report the average cross-validation accuracy on the publicly-available training set with balanced categories. Bold indicates the better-performing method.

R4 – Comment 2

The implications of this work are that the proposed machine learning model based on transformers is able to perform well and generalize on the classification of multiple classification tasks in isolated surgical activities. The performance on the proposed tasks seems satisfactory. However, the clinical applicability needs further validation.

Response to R4 – Comment 2

Indeed, SAIS opens the door to many future applications such as associating intraoperative surgical activity with postoperative patient outcomes. Based on a recommendation from **Reviewer 2 (Comment 8)**, we have now provided a preliminary analysis associating SAIS skill assessment outputs with postoperative outcomes (3-month urinary continence recovery), demonstrating a statistically significant relationship. Further studies are required on this front.

SAIS' skill assessments are associated with patient outcomes

While useful for mastering a surgical technical skill itself, surgeon feedback becomes more clinically meaningful when grounded in patient outcomes. For example, if low-skill assessments are associated with poor outcomes, then a surgeon can begin to modulate specific behaviour to improve such outcomes. To that end, we conducted a preliminary analysis regressing SAIS' surgeon skill assessments at USC onto a patient's binary recovery of urinary continence (ability to voluntarily control urination) 3 months after surgery (see details in Methods). When considering all video samples (multiple per surgical case), and controlling for surgeon caseload and patient age, we found that urinary continence recovery was 1.31 (odds ratio, CI: 1.08 – 1.58, $p = 0.005$) more likely when needle driving was assessed as high-skill than as low-skill by SAIS. When aggregating the skill assessments of video samples within a surgical case, that relationship is further strengthened (odds ratio: 1.89 CI: 0.95 – 3.76, $p = 0.071$). These preliminary findings are consistent with those based on *manual* skill assessments from recent studies^{18,19}.

R4 – Comment 3

Metrics like mAP and F1 Score are well established in the domain of surgical phase and action recognition and they should be reported here as well.

Response to R4 – Comment 3

We take the reviewer's suggestion and report the F1@10 score when both SAIS and I3D are deployed for decoding suturing sub-phases from entire vesico-urethral suturing videos (see **Figure**

2e, page 3). We adopted this metric based on a recent survey. A description of this inference process can be found in Methods → Implementation details of inference on entire videos (page 16). We also report the accuracy of gesture classification and compare that to the performance of baseline models.

R4 – Comment 4

The demonstrated model is not evaluated on other datasets. The dataset is unfortunately not public, therefore the results are not reproducible even though the model and code are to be published.

Response to R4 – Comment 4

Please refer to our response to **R4 – Comment 1** which addresses these comments.

R4 – Comment 5

The authors claim that previous work commonly allows for the leakage of data from the same video into training and test set without any specific reference or example. This is incorrect. Avoiding data leakage is standard practice in the community and most of the referenced papers are doing so. If the authors are aware of any exceptions, they need to reference them explicitly and discuss the issues. Describing not allowing data leakage in this work as especially rigorous evaluation is therefore misleading.

Response to R4 – Comment 5

We have now dedicated an entire section outlining the limitations of the evaluation setup employed by previous work. This can be found in Methods → Previous work → Evaluation setups (page 11). In short, we provide several examples in which there is either minimal information provided about the data splits or, in some cases, data splits communicated to the public which appear to contain surgeon data leakage. It is for this reason that we stress the importance of rigorously evaluating AI systems in distinct settings (e.g., across surgical procedures, surgeons, hospitals, etc.) as we have started to do in this project.

Evaluation setups Previous work often splits their data in a way that has the potential for information “leakage” across training and test sets. For example, it is believed that the commonly-adopted leave-one-user-out (LOUO) evaluation setup on the JIGSAWS dataset¹⁴ is rigorous. Although it lends insight into the generalizability of a model to a video from an unseen participant, this setup involves reporting a cross-validation score, which is often directly optimized by previous methods (e.g., through hyperparameter tuning), therefore producing an overly-optimistic estimate of performance. As another example, consider the data split used for the CholecT50 dataset³⁹. Here, there is minimal information about whether videos in the training and test sets belong to the same surgeon. Lastly, the most recent DVC UCL dataset¹⁵ consists of 36 publicly-available videos for training and 9 private videos for testing. After manual inspection, we found that these 9 videos come from 6 surgeons whose data are also in the training set. This is a concrete example of surgeon data leakage and, as such, caution the use of such datasets for benchmarking purposes. It is therefore critical to more rigorously evaluate the performance of surgical AI systems and in accordance with how they are likely to be deployed in a clinical setting.

R4 – Comment 6

Simplification of surgical events might not represent the true complexity of surgical interventions e.g. Skill level: High and Low.

Response to R4 – Comment 6

We talk about the limitations of such a simplification and those of existing taxonomies in the **Discussion section (page 10)**. Although the jury is still out, it is entirely conceivable that such a simplification may still provide sufficient insight into the variability of surgical activity, its relationship to postoperative outcomes, and so forth. By open-sourcing SAIS to the public, we can allow researchers to begin to decode their surgical videos at whatever level of detail they are interested in.

There are important challenges our work does not yet address. First, our framework, akin to others in the field, is limited to only decoding the elements of surgical activity that have been previously outlined in some taxonomy (e.g., gestures). In other words, it cannot decode what it does not know. Although many of these taxonomies have been rigorously developed by teams of surgeons and through clinical experience, they may fail to shed light on other intricate aspects of surgical activity. This, in turn, limits the degree to which automated systems can *discover* novel activity that falls beyond the realm of existing protocol. Such discovery can lend insight into, for example, optimal but as-of-yet undiscovered surgical behaviour. In a similar vein, SAIS is currently incapable of decoding new elements of surgical activity beyond those initially presented to it. Such continual learning capabilities¹⁰ are critical to adapting to an evolving taxonomy of surgical activity over time.

R4 – Comment 7

The relevance of steps, gestures and skill level on suturing and dissection activities to postoperative patient outcomes remains unknown until further studies in this direction are published. It can be

assumed that the impact of skill difference in the suturing step on patient outcome is limited. The impact of the quality of the dissection step can be assumed to be higher, but it is not analyzed here.

Response to R4 – Comment 7

We agree with the reviewer's statement about the relevance of the decoded elements and outcomes. Indeed, we have included a discussion to that end in the **Discussion section (page 10)**.

The goal of surgery is to improve patient outcomes. However, it remains an open question whether or not the decoded elements of intraoperative surgical activity: sub-phases, gestures, and skills, are the factors most predictive of postoperative patient outcomes. Although we have presented preliminary evidence in this direction for the case of surgical skills, large-scale studies are required to unearth these relationships. To further explore these relationships and more reliably inform future surgical practice, we encourage the public release of large-scale surgical video datasets from different hospitals and surgical specialties. Equipped with such videos and SAIS, researchers can begin to decode the various elements of surgery at scale.

In this version of the manuscript, however, we do provide preliminary evidence that SAIS' skill assessments are associated with postoperative outcomes (3-month urinary continence recovery) although we acknowledge the need for larger-scale studies (see **Results → SAIS' skill assessments are associated with patient outcomes, page 9**). This is consistent with prior publications by our group that surgeon skills assessment (albeit manually rated) are associated with long-term patient outcomes. By decoding surgical videos at scale from across the globe, we can begin to test hypotheses relating intraoperative surgical activity to postoperative outcomes.

SAIS' skill assessments are associated with patient outcomes

While useful for mastering a surgical technical skill itself, surgeon feedback becomes more clinically meaningful when grounded in patient outcomes. For example, if low-skill assessments are associated with poor outcomes, then a surgeon can begin to modulate specific behaviour to improve such outcomes. To that end, we conducted a preliminary analysis regressing SAIS' surgeon skill assessments at USC onto a patient's binary recovery of urinary continence (ability to voluntarily control urination) 3 months after surgery (see details in Methods). When considering all video samples (multiple per surgical case), and controlling for surgeon caseload and patient age, we found that urinary continence recovery was 1.31 (odds ratio, CI: 1.08 – 1.58, $p = 0.005$) more likely when needle driving was assessed as high-skill than as low-skill by SAIS. When aggregating the skill assessments of video samples within a surgical case, that relationship is further strengthened (odds ratio: 1.89 CI: 0.95 – 3.76, $p = 0.071$). These preliminary findings are consistent with those based on *manual* skill assessments from recent studies^{18,19}.

R4 – Comment 8

The authors proclaim that their model can already be reliably deployed on surgical videos of the nerve-sparing dissection step, but the performance on some of the classes e.g. in Fig. 5a does not seem sufficient for deployment in a clinical setting. (Recall ~ 0.5 and lower). Parts of the surgical

procedure (clips of suturing and dissection) are isolated manually, limiting automated analysis capability as proposed in this paper.

Response to R4 – Comment 8

To clarify, the results reported in Figure 4a are of Precision and *not* Recall. As for the gesture category with relatively lower performance (cold cut), we provide the reasoning behind such performance and, in fact, discover that SAIS was identifying a novel gesture that it was not explicitly trained to identify and which overlaps significantly with the cold cut gesture. Therefore, this precision score would increase when considering “cutting” gestures as a whole (see Results → SAIS provides surgical gesture information otherwise unavailable to surgeons → Quantitative evaluation, page 6)

As for full automation, we do not make any claims in the manuscript that SAIS will fully automate the decoding of *all* elements of surgical activity for *all* videos. Although SAIS will require being fed videos of a particular surgical step (e.g., nerve-sparing dissection of the radical prostatectomy), identifying this video manually only requires the provision of two timestamps (the start and end time of the nerve-sparing dissection step), which is not time-consuming or laborious.

If, for some reason, providing these two time-stamps happen to be a bottleneck for researchers, then we also provide additional evidence that SAIS can comfortably distinguish between the tissue dissection and tissue suturing steps of a surgery. Specifically, we demonstrate that SAIS achieves an AUC = 1 when distinguishing between the nerve-sparing (NS) dissection step and the vesico-urethral anastomosis (VUA) step (Supplementary Note 3 → Supplementary Figure 2). We had hypothesized, prior to conducting the experiments, that such a task (phase recognition) would be relatively straightforward as the various steps of surgery are often more distinguishable based on visual cues compared to the more subtle elements of surgery (e.g., sub-phases, gestures, etc.).

Supplementary Figure 2. SAIS reliably decodes surgical phases across videos. SAIS is trained on video samples exclusively from USC and also evaluated on video samples from USC. Results are shown as an average (± 1 standard deviation) of 10 Monte-Carlo cross-validation steps.

Once the video of a surgical step is provided to SAIS, it can then automatically decode the various elements of surgical activity. We had originally presented such an example of decoding from an entire nerve-sparing video in the context of dissection gesture classification (Figure 4). We now also present such a decoding from an entire vesico-urethral anastomosis suturing video in the context of sub-phase recognition (Figure 2e, page 3). We provide details of this inference in Methods → Implementation details of inference on entire videos (page 16).

R4 – Comment 9

In figure 7. the caption and the paragraph below state that SAIS can provide skill information that otherwise would not be available - but this is also true for all surgical assessment methods.

Response to R4 – Comment 9

While we agree with the reviewer that surgical assessment methods would also provide surgeons with otherwise unavailable information, we wanted to emphasize this point to the lay reader who may be less familiar with the potential utility of such models.

R4 – Comment 10

In Discussion the first sentence says: “Only in the past decade or so has it been demonstrated that intraoperative surgical activity can have a direct influence on postoperative patient outcomes”. This has been a known fact of surgery, in the last decades only the automatic analysis has been discussed.

Response to R4 – Comment 10

We have since modified the statement to mention that *empirical relationships* have only just been unearthed (see Discussion section, page 9).

Discussion

Only in the past decade or so has it been empirically demonstrated that intraoperative surgical activity can have a direct influence on postoperative patient outcomes. However, discovering and acting upon this relationship to improve outcomes is challenging when the details of intraoperative surgical activity remain elusive. By combining emerging technologies such as artificial intelligence with videos routinely collected as part of existing robotic surgeries, we can begin to decode multiple elements of intraoperative surgical activity.

Rebuttal 2

We thank the reviewers for taking the time and effort to provide us with additional feedback. We address your comments below. To avoid potential confusion, please note that our response to *R1 – Comment 1* is the same as that to *R3 – Comment 1*.

POINT-BY-POINT RESPONSE

Reviewer 1

R1 – Comment 1

I appreciate the efforts of the authors to revise the manuscript.

The key issues are still there regarding the lack of a significant degree of advance (fundamental, mechanistic, methodological, technological, therapeutic) in the work with respect to state of the art.

The authors themselves also agreed that “application merit is clear but technical novelty is unclear” and admitted that “the primary benefit of SAIS is translational in nature as it lends confidence into its use for live surgical videos stemming from distinct settings”. However, more novel fundamental/mechanistic/methodological/technological/therapeutic contributions are typically expected for a publication appears in the top journal of “Nature Biomedical Engineering”.

This work uses the same ViT model to achieve three different tasks (phase recognition, gesture recognition, and skill assessment). Simply pursuing the number of tasks does not seem to justify more novelties.

Response to R1 – Comment 1

Compared to previous studies, our study offers both translational and methodological contributions. We more clearly outline these contributions next and include them in the **Discussion** section (**page 10**) of the manuscript.

Translational Contribution – SAIS generalizes across distinct settings

From a translational standpoint, we demonstrated SAIS' ability to generalize across videos, surgeons, surgical procedures, and hospitals.

Why this matters

Such a finding is likely to instill surgeons with confidence in the trustworthiness of SAIS, and therefore increases their likelihood of adopting it.

How this compares to previous work

This is in contrast to previous work that has evaluated AI systems on videos captured in either a controlled laboratory environment or a single hospital, thereby demonstrating limited generalization capabilities.

Compared to previous studies (see previous work section), our study offers both translational and methodological contributions. From a translational standpoint, we demonstrated SAIS' ability to generalize across videos, surgeons, surgical procedures, and hospitals. Such a finding is likely to instill surgeons with greater confidence in the trustworthiness of SAIS, and therefore increases their likelihood of adopting it. This is in contrast to previous work that has evaluated AI systems on videos captured in either a controlled laboratory environment or a single hospital, thereby demonstrating limited generalization capabilities.

Methodological Contribution 1 – SAIS is a unified AI system

From a methodological standpoint, SAIS has much to offer compared to AI systems previously developed for decoding surgical activity. First, SAIS is *unified* in that it is capable of decoding multiple elements of intraoperative surgical activity without any changes to its underlying architecture.

Why this matters

Generally speaking, the ability of a model to perform well on Task A (e.g., sub-phase recognition) does not guarantee that it will also perform well on Task B (e.g., skill assessment), particularly if the two tasks are distinct from one another and require focusing on different aspects of the input data. As such, we do believe that our demonstrating that a single architecture (with no modifications) can perform consistently well across three distinct surgical tasks is a worthwhile contribution to the community. By acting as a dependable core architecture around which future developments are made, SAIS is likely to reduce the amount of resources and cognitive burden associated with developing AI systems to decode additional elements of surgical activity.

How this compares to previous work

This is in contrast to the status quo in which the burdensome process of developing specialized AI systems must be undertaken to decode just a single element.

Methodological Contribution 2 – SAIS provides explainable findings

Second, SAIS provides *explainable* findings in that it can highlight the relative importance of individual video frames in contributing to the decoding.

Why this matters

Such explainability is critical to gaining the trust of surgeons and ensuring the safe deployment of AI systems for high-stakes decision making such as skill-based surgeon credentialing.

How this compares to previous work

This is in contrast to previous AI systems such as MA-TCN which is only capable of highlighting the relative importance of data modalities (e.g., images vs. kinematics), and therefore lacks SAIS' finer level of explainability.

From a methodological standpoint, SAIS has much to offer compared to AI systems previously developed for decoding surgical activity. First, SAIS is *unified* in that it is capable of decoding multiple elements of intraoperative surgical activity without any changes to its underlying architecture. By acting as a dependable core architecture around which future developments are made, SAIS is likely to reduce the amount of resources and cognitive burden associated with developing AI systems to decode additional elements of surgical activity. This is in contrast to the status quo in which the burdensome process of developing specialized AI systems must be undertaken to decode just a single element. Second, SAIS provides *explainable* findings in that it can highlight the relative importance of individual video frames in contributing to the decoding. Such explainability is critical to gaining the trust of surgeons and ensuring the safe deployment of AI systems for high-stakes decision making such as skill-based surgeon credentialing. This is in contrast to previous AI systems such as MA-TCN¹⁵ which is only capable of highlighting the relative importance of data modalities (e.g., images vs. kinematics), and therefore lacks SAIS' finer level of explainability.

Methodological Contribution 3 – SAIS is a flexible AI system

SAIS is also *flexible* in that it can accept videos samples with an arbitrary number of video frames as input, primarily due to its transformer architecture.

Why this matters and how this compares to previous work

Such flexibility, which is absent from previous commonly-used models such as 3D-CNNs, confers benefits to training, fine-tuning, and performing inference with, SAIS. During training, SAIS can accept a mini-batch of videos each with a different number of frames. This can be achieved by padding videos in the mini-batch (with zeros) that have fewer frames, and appropriately masking the attention mechanism in the transformer encoder (see implementation details). This is in contrast to existing AI systems which must often be presented with a mini-batch of equally-sized videos. Similarly, during fine-tuning or inference, SAIS can be presented with an arbitrary number of video frames, and thus expanding the spectrum of videos that it can be presented with. This is in contrast to existing setups that leverage a 3D-CNN which has been pre-trained on the Kinetics dataset, whereby video samples must contain either 16 frames or multiples thereof. Abiding by this constraint can be sub-optimal for achieving certain tasks, and departing from it implies the inability to leverage the pre-trained parameters that have proven critical to the success of previous methods.

Methodological Contribution 4 – SAIS is architecturally different from baseline methods

SAIS is *architecturally* different from previous models in that it learns prototypes via supervised contrastive learning in order to decode surgical activity, an approach that has yet to be explored with surgical videos.

Why this matters

Such prototypes pave the way for multiple downstream applications from detecting out-of-distribution video samples, to identifying clusters of intraoperative activity, and retrieving samples from a large surgical database, as demonstrated by our team's previous work.

SAIS is also *flexible* in that it can accept video samples with an arbitrary number of video frames as input, primarily due to its transformer architecture. Such flexibility, which is absent from previous commonly-used models such as 3D-CNNs, confers benefits to training, fine-tuning, and performing inference with, SAIS. During training, SAIS can accept a mini-batch of videos each with a different number of frames. This can be achieved by padding videos in the mini-batch (with zeros) that have fewer frames, and appropriately masking the attention mechanism in the transformer encoder (see implementation details). This is in contrast to existing AI systems which must often be presented with a mini-batch of equally-sized videos. Similarly, during fine-tuning or inference, SAIS can be presented with an arbitrary number of video frames, and thus expanding the spectrum of videos that it can be presented with. This is in contrast to existing setups that leverage a 3D-CNN which has been pre-trained on the Kinetics dataset²⁰, whereby video samples must contain either 16 frames or multiples thereof^{6,16}. Abiding by this constraint can be sub-optimal for achieving certain tasks, and departing from it implies the inability to leverage the pre-trained parameters that have proven critical to the success of previous methods. Furthermore, SAIS is *architecturally* different from previous models in that it learns prototypes via supervised contrastive learning in order to decode surgical activity, an approach that has yet to be explored with surgical videos. Such prototypes pave the way for multiple downstream applications from detecting out-of-distribution video samples, to identifying clusters of intraoperative activity, and retrieving samples from a large surgical database²¹.

R1 – Comment 2

The learning among these three tasks (sub-phase recognition, gesture classification, and skill assessment) are independent of one another) are independent of each other. However, there are actually inherent connections between these tasks. The authors responded and would like to leave these discoveries to future work, which left many important questions unanswered.

Response to R1 – Comment 2

We appreciate the reviewer's interest in exploring the interconnections between the various surgical tasks (sub-phase recognition, gesture classification, and skill assessment). Indeed, there are various ways to go about conducting such an exploration. For example, and as we have mentioned before, one can train a multi-task network that simultaneously performs the aforementioned surgical tasks. In such a setting, positive interference between the tasks could result in even further performance improvements. As another example, one can train a network to initially perform sub-phase recognition (relatively easier task) and subsequently transfer its parameters to perform skill assessment (relatively harder task). This is akin to curriculum learning, whereby a network is presented with increasingly difficult tasks during the learning process. In light of these exciting avenues here, we now shed light on them through a dedicated paragraph in the **Discussion** section (page 11).

We also intend to explore the interdependency of the elements of intraoperative surgical activity (sub-phase recognition, gesture classification, and skill assessment). This can be achieved, for example, by training a multi-task variant of SAIS in which all elements are simultaneously decoded from a video. In such a setting, positive interference between the tasks could result in even more reliable decoding. Alternatively, SAIS can be trained to first perform sub-phase recognition (a relatively easy task) before transferring its parameters to perform skill assessment (a relatively harder task). This is akin to curriculum learning²², whereby an AI system is presented with increasingly difficult tasks during the learning process in order to improve its overall performance. We also aim to study whether SAIS exhibits algorithmic bias against various surgeon sub-cohorts²³. Such a bias analysis is particularly critical if SAIS is to be used for the provision of feedback to surgeons. For example, it may disadvantage certain surgeon sub-cohorts (e.g., novices with little experience) and thus affect their ability to develop professionally.

R1 – Comment 3

Still, many related and recent AI-engineering references on Surgical Workflow Recognition, Surgical State Estimation, Surgical Scenes understanding, Surgical Interaction Recognition are missing.

Related ideas with original methodological algorithms have been published in the literature. For example:

Qin, Y., Allan, M., Burdick, J. W., & Azizian, M. (2021). Autonomous hierarchical surgical state estimation during robot-assisted surgery through deep neural networks. *IEEE Robotics and Automation Letters*, 6(4), 6220-6227. This paper proposed Hierarchical Estimation of Surgical States to estimate the associated super- and fine-grained states concurrently.

Shi, X., Jin, Y., Dou, Q., & Heng, P. A. (2021). Semi-supervised learning with progressive unlabeled data excavation for label-efficient surgical workflow recognition. *Medical Image Analysis*, 73, 102158.

Wagner, M., Müller-Stich, B. P., Kisilenko, A., Tran, D., Heger, P., Mündermann, L., ... & Bodenstedt, S. (2021). Comparative validation of machine learning algorithms for surgical workflow and skill analysis with the HeiChole benchmark. *arXiv preprint arXiv:2109.14956*.

Soleymani, A., Asl, A. A. S., Yeganejou, M., Dick, S., Tavakoli, M., & Li, X. (2021, November). Surgical skill evaluation from robot-assisted surgery recordings. In *2021 International Symposium on Medical Robotics (ISMR)* (pp. 1-6). IEEE.

Shi, X., Jin, Y., Dou, Q., & Heng, P. A. (2020). LRTD: long-range temporal dependency based active learning for surgical workflow recognition. *International Journal of Computer Assisted Radiology and Surgery*, 15(9), 1573-1584.

Seenivasan, L., Mitheran, S., Islam, M., & Ren, H. (2022). Global-Reasoned Multi-Task Learning Model for Surgical Scene Understanding. *IEEE Robotics and Automation Letters*, 7(2), 3858-3865.

van Amsterdam, B., Clarkson, M. J., & Stoyanov, D. (2020, May). Multi-task recurrent neural network for surgical gesture recognition and progress prediction. In 2020 IEEE International Conference on Robotics and Automation (ICRA) (pp. 1380-1386). IEEE.

Zia, A., Guo, L., Zhou, L., Essa, I., & Jarc, A. (2019). Novel evaluation of surgical activity recognition models using task-based efficiency metrics. *International journal of computer assisted radiology and surgery*, 14(12), 2155-2163.

Gao, X., Jin, Y., Dou, Q., & Heng, P. A. (2020, May). Automatic gesture recognition in robot-assisted surgery with reinforcement learning and tree search. In 2020 IEEE International Conference on Robotics and Automation (ICRA) (pp. 8440-8446). IEEE.

Response to R1 – Comment 3

We thank the reviewer for bringing these references to our attention, and have now expanded our related work section accordingly (**Methods** → **Related work**, **page 11**). We do note that, since our manuscript is not a review paper, we direct readers towards more comprehensive review papers that provide a high-level overview of related methodologies.

Reviewer 2

Summary

Thank you for the opportunity to review the manuscript again.
My questions were answered clearly and appropriately by the authors.
I have no further comments.

Reviewer 3

Summary

The authors have strengthened the clinical contribution of the work by clarifying the dataset description, adding another hospital, and adding the section correlating skill to clinical outcome. This is a valuable contribution to the field since curating datasets and long term studies to track patient outcome is often not possible depending on the institution. The translational contribution is useful as it does show that AI systems can help with targeting surgeon training, and potentially improving patient outcome. The authors have done a thorough job in addressing the concerns raised in the initial review.

R3 – Comment 1

The technical contribution of the work is still not obvious though.

Response to R3 – Comment 1

Compared to previous studies, our study offers both translational and methodological contributions. We more clearly outline these contributions next and include them in the **Discussion** section (page 10) of the manuscript.

Translational Contribution – SAIS generalizes across distinct settings

From a translational standpoint, we demonstrated SAIS' ability to generalize across videos, surgeons, surgical procedures, and hospitals.

Why this matters

Such a finding is likely to instill surgeons with confidence in the trustworthiness of SAIS, and therefore increases their likelihood of adopting it.

How this compares to previous work

This is in contrast to previous work that has evaluated AI systems on videos captured in either a controlled laboratory environment or a single hospital, thereby demonstrating limited generalization capabilities.

Compared to previous studies (see previous work section), our study offers both translational and methodological contributions. From a translational standpoint, we demonstrated SAIS' ability to generalize across videos, surgeons, surgical procedures, and hospitals. Such a finding is likely to instill surgeons with greater confidence in the trustworthiness of SAIS, and therefore increases their likelihood of adopting it. This is in contrast to previous work that has evaluated AI systems on videos captured in either a controlled laboratory environment or a single hospital, thereby demonstrating limited generalization capabilities.

Methodological Contribution 1 – SAIS is a unified AI system

From a methodological standpoint, SAIS has much to offer compared to AI systems previously developed for decoding surgical activity. First, SAIS is *unified* in that it is capable of decoding multiple elements of intraoperative surgical activity without any changes to its underlying architecture.

Why this matters

Generally speaking, the ability of a model to perform well on Task A (e.g., sub-phase recognition) does not guarantee that it will also perform well on Task B (e.g., skill assessment), particularly if the two tasks are distinct from one another and require focusing on different aspects of the input data. As such, we do believe that our demonstrating that a single architecture (with no modifications) can perform consistently well across three distinct surgical tasks is a worthwhile contribution to the community. By acting as a dependable core architecture around which future developments are made, SAIS is likely to reduce the amount of resources and cognitive burden associated with developing AI systems to decode additional elements of surgical activity.

How this compares to previous work

This is in contrast to the status quo in which the burdensome process of developing specialized AI systems must be undertaken to decode just a single element.

Methodological Contribution 2 – SAIS provides explainable findings

Second, SAIS provides *explainable* findings in that it can highlight the relative importance of individual video frames in contributing to the decoding.

Why this matters

Such explainability is critical to gaining the trust of surgeons and ensuring the safe deployment of AI systems for high-stakes decision making such as skill-based surgeon credentialing.

How this compares to previous work

This is in contrast to previous AI systems such as MA-TCN which is only capable of highlighting the relative importance of data modalities (e.g., images vs. kinematics), and therefore lacks SAIS' finer level of explainability.

From a methodological standpoint, SAIS has much to offer compared to AI systems previously developed for decoding surgical activity. First, SAIS is *unified* in that it is capable of decoding multiple elements of intraoperative surgical activity without any changes to its underlying architecture. By acting as a dependable core architecture around which future developments are made, SAIS is likely to reduce the amount of resources and cognitive burden associated with developing AI systems to decode additional elements of surgical activity. This is in contrast to the status quo in which the burdensome process of developing specialized AI systems must be undertaken to decode just a single element. Second, SAIS provides *explainable* findings in that it can highlight the relative importance of individual video frames in contributing to the decoding. Such explainability is critical to gaining the trust of surgeons and ensuring the safe deployment of AI systems for high-stakes decision making such as skill-based surgeon credentialing. This is in contrast to previous AI systems such as MA-TCN¹⁵ which is only capable of highlighting the relative importance of data modalities (e.g., images vs. kinematics), and therefore lacks SAIS' finer level of explainability.

Methodological Contribution 3 – SAIS is a flexible AI system

SAIS is also *flexible* in that it can accept videos samples with an arbitrary number of video frames as input, primarily due to its transformer architecture.

Why this matters and how this compares to previous work

Such flexibility, which is absent from previous commonly-used models such as 3D-CNNs, confers benefits to training, fine-tuning, and performing inference with, SAIS. During training, SAIS can accept a mini-batch of videos each with a different number of frames. This can be achieved by padding videos in the mini-batch (with zeros) that have fewer frames, and appropriately masking the attention mechanism in the transformer encoder (see implementation details). This is in contrast to existing AI systems which must often be presented with a mini-batch of equally-sized videos. Similarly, during fine-tuning or inference, SAIS can be presented with an arbitrary number of video frames, and thus expanding the spectrum of videos that it can be presented with. This is in contrast to existing setups that leverage a 3D-CNN which has been pre-trained on the Kinetics dataset, whereby video samples must contain either 16 frames or multiples thereof. Abiding by this constraint can be sub-optimal for achieving certain tasks, and departing from it implies the inability to leverage the pre-trained parameters that have proven critical to the success of previous methods.

Methodological Contribution 4 – SAIS is architecturally different from baseline methods

SAIS is *architecturally* different from previous models in that it learns prototypes via supervised contrastive learning in order to decode surgical activity, an approach that has yet to be explored with surgical videos.

Why this matters

Such prototypes pave the way for multiple downstream applications from detecting out-of-distribution video samples, to identifying clusters of intraoperative activity, and retrieving samples from a large surgical database, as demonstrated by our team's previous work.

SAIS is also *flexible* in that it can accept video samples with an arbitrary number of video frames as input, primarily due to its transformer architecture. Such flexibility, which is absent from previous commonly-used models such as 3D-CNNs, confers benefits to training, fine-tuning, and performing inference with, SAIS. During training, SAIS can accept a mini-batch of videos each with a different number of frames. This can be achieved by padding videos in the mini-batch (with zeros) that have fewer frames, and appropriately masking the attention mechanism in the transformer encoder (see implementation details). This is in contrast to existing AI systems which must often be presented with a mini-batch of equally-sized videos. Similarly, during fine-tuning or inference, SAIS can be presented with an arbitrary number of video frames, and thus expanding the spectrum of videos that it can be presented with. This is in contrast to existing setups that leverage a 3D-CNN which has been pre-trained on the Kinetics dataset²⁰, whereby video samples must contain either 16 frames or multiples thereof^{6,16}. Abiding by this constraint can be sub-optimal for achieving certain tasks, and departing from it implies the inability to leverage the pre-trained parameters that have proven critical to the success of previous methods. Furthermore, SAIS is *architecturally* different from previous models

in that it learns prototypes via supervised contrastive learning in order to decode surgical activity, an approach that has yet to be explored with surgical videos. Such prototypes pave the way for multiple downstream applications from detecting out-of-distribution video samples, to identifying clusters of intraoperative activity, and retrieving samples from a large surgical database²¹.

R3 – Comment 2

As mentioned by multiple reviewers, there is a lot of existing work on this topic and from Tables 1 and 2, it is unclear that the proposed method gets better performance than existing methods (even with other modalities that only use video). It's not clear to me that a percentage improvement over majority class is a convincing metric since the improvement seems to mostly derive from Random doing much worse. This is especially concerning if G5 was discarded from the analysis for SAIS but included in MA-TCN.

Response to R3 – Comment 2

As it pertains to Table 2 and the MA-TCN results, we would like to clarify that the performance improvements brought about by SAIS and MA-TCN (which happens to use kinematics data as an additional modality) are *4-fold* and *3-fold*, respectively, compared to the Random setting. Therefore, the difference in performance improvement between the two methods (SAIS vs. MA-TCN) is more than just a single percentage point. We have now modified the column heading in Table 2 to avoid any confusion.

Method	Accuracy (%)		
	Random	Reported	Improved
MA-TCN ¹⁵	25.9	80.9	3.1×
SAIS (ours)	14.3	59.8 (1.0)	4.2×

Table 2. Accuracy of gesture classification on the DVC UCL dataset. MA-TCN reports accuracy on a private test-set with gesture imbalance. We report the average cross-validation accuracy on the publicly-available training set with balanced categories. Bold indicates the better-performing method.

R3 – Comment 3

The architecture proposed does not seem fundamentally different than other transformer-based architecture (such as MA-TCN), combined with established ensemble methods. The value was that it was validated on data from multiple hospitals.

Response to R3 – Comment 3

As it pertains to the architecture of SAIS compared to that of other attention-based models, there are several differences.

Architectural difference 1 – vision transformer as feature extractor

To the best of our knowledge, and at the time of submission, we are the first to use a *vision transformer* (to extract spatial features) as part of the architectural backbone in the context of surgical AI. In contrast, MA-TCN uses a convolutional neural network (CNN) feature extractor. This architectural difference also lends itself to benefits beyond performance, and

in particular, engenders greater flexibility in how video samples are fed into the AI system (see Discussion, page 10, paragraph 3).

Architectural difference 2 – attention at the level of individual video frames

As a result of using the vision transformer as a backbone for feature extraction, SAIS operates at the level of individual video frames. By subsequently using transformer encoders to capture the relationship between such frames, we can inspect the temporal attention (and therefore importance) of frames in contributing to the final prediction (e.g., skill assessment). While previous work such as MA-TCN also uses attention, it does so to combine various modalities and thus lacks our model's level of interpretability (see Discussion, page 10, paragraph 2).

Architecture difference 3 – prototypes are learned and used for classification

In contrast to previous architectures, we explicitly learn prototypes (embeddings) in an end-to-end manner via a supervised contrastive loss function, and then directly use these prototypes for classification. To the best of our knowledge, prototypes and this learning process have yet to be leveraged alongside surgical videos (see Discussion, page 10, paragraph 3, and Methodological Contribution 4 in Response to R3 – Comment 1).